# TeamCraft: A Multi-Modal Benchmark for Collaborative Agents in Minecraft

Zhi Li[1][*] Qian Long[1][*] Ran Gong[1], Ying Nian Wu[1], Demetri Terzopoulos[1], Xiaofeng Gao[2][†]
[1]University of California, Los Angeles, [2]Amazon AGI
teamcraftbench@gmail.com
https://teamcraft-bench.github.io/

## Abstract

Collaboration is a cornerstone of society. In the real world, human teammates make use of multi-sensory data to tackle challenging tasks in ever-changing environments. It is likewise essential for embodied agents collaborating in visually-rich environments replete with dynamic interactions to understand multi-modal observations and task specifications. To evaluate the performance of generalizable multi-modal collaborative agents, we present *TeamCraft*, a multi-modal multi-agent benchmark built on top of the open-world video game Minecraft. The benchmark features 55,000 task variants specified by multi-modal prompts, procedurally-generated expert demonstrations for imitation learning, and carefully designed protocols to evaluate model generalization capability. We also perform extensive analyses to better understand the limitations and strengths of existing approaches. Our results indicate that existing models continue to face significant challenges in generalizing to novel goals, scenes, and unseen numbers of agents. These findings underscore the potential for further research in this area. The *TeamCraft* platform and dataset are publicly available at https://github.com/teamcraft-bench/teamcraft.

## 1 Introduction

Developing collaborative skills is essential for embodied agents, as collaboration is a fundamental aspect of human intelligence (Smith & Gasser, 2005). In the AI community, multi-agent collaboration is frequently studied using grid-world environments (Leibo et al., 2021; Suarez et al., 2021; Stone & Veloso, 2000; Gong et al., 2023c; Dong et al., 2024; Puig et al., 2021; Park et al., 2023; Zhang et al., 2024a; Wu et al., 2021; Long et al., 2024a). However, agents in these environments lack multi-modal understanding. By contrast, learning within visually-rich environments enables agents to develop useful representations of multi-agent dynamics (Chen et al., 2020; Jaderberg et al., 2019), as vision facilitates implicit communication, coordination, and collaborative execution (Jain et al., 2020, 2019).

Learning vision-based, multi-task, multi-agent systems is a challenging objective that presents several difficulties. These systems must develop detailed scene understanding to handle the diverse visual appearances of scenes. The complexity is further heightened by the numerous combinations of task configurations, such as object spatial arrangements, goal configurations, arbitrary numbers of agents, and heterogeneous agent capabilities. Consequently, it is essential for multi-agent systems to acquire generalizable skills that can be effectively transferred across different settings.

An important step in addressing these challenges is to develop simulation systems that support multi-modal multi-agent learning. Recent advances in simulated environments have significantly facilitated progress in embodied vision-based systems (Yu et al., 2024; Jain et al., 2020; Chen et al.,

---

[*]Equal contribution.
[†]This work does not relate to the author's position at Amazon.

Submitted to 39th Conference on Neural Information Processing Systems (NeurIPS 2025). Do not distribute.

| Benchmark | MM Spec. | 3D | Observation | MA | Interaction | Tool | Generalization | # Agents | # Variants | # Demonstrations |
|---|---|---|---|---|---|---|---|---|---|---|
| ALFRED (Shridhar et al., 2020a) | ✗ | ✓ | V | ✗ | ✓ | ✓ | E | 1 | 2,600+ | 8,000+ |
| FurnMove (Jain et al., 2020) | ✗ | ✓ | V | CD | ✓ | ✗ | E | 2 | 30 | ✗ |
| Marlo (Perez-Liebana et al., 2019) | ✗ | ✓ | V | D | ✓ | ✗ | ✗ | 4+ | 14 | ✗ |
| MineDojo (Fan et al., 2022) | ✗ | ✓ | V | ✗ | ✓ | ✓ | EG | 1 | 3,000+ | 740,000+ |
| MindAgent (Gong et al., 2023c) | ✗ | ✓ | VS | C | ✓ | ✓ | ✗ | 4+ | 39 | ✗ |
| Neural MMO 2.0 (Suárez et al., 2024) | ✗ | ✗ | S | CD | ✓ | ✓ | EGA | 128+ | 25+ | ✗ |
| Overcooked-AI (Carroll et al., 2020) | ✗ | ✗ | VS | C | ✓ | ✓ | ✗ | 2 | 5 | 80 |
| PARTNR (Chang et al., 2024) | ✗ | ✓ | VS | CD | ✓ | ✓ | E | 2 | 100,000+ | 100,000+ |
| RoCoBench (Mandi et al., 2024) | ✗ | ✓ | S | CD | ✓ | ✓ | G | 2 | 6 | ✗ |
| VIMA-Bench (Jiang et al., 2022) | ✓ | ✓ | V | ✗ | ✓ | ✓ | EG | 1 | 1,000+ | 600,000+ |
| Watch&Help (Puig et al., 2021) | ✗ | ✓ | S | CD | ✓ | ✓ | EG | 2 | 1,200+ | 6,300+ |
| *TeamCraft* | ✓ | ✓ | VS | CD | ✓ | ✓ | EGA | 4+ | 55,000+ | 55,000+ |

Table 1: Comparison with other benchmarks. *TeamCraft* features visual observation for multi-agent control with widely-varied tasks specified by multi-modal prompts, targeting various types of generalization essential for multi-agent teaming. **MM Spec.**: multi-modal task specification. **Observation**: **V** for visual observation and **S** for state-based observation. **MA**: multi-agent control, **C** for centralized and **D** for decentralized. **Interaction**: object interaction. **Tool**: tool use. **Generalization**: types of generalization targeted, **E** for generalization on novel environments or scenes, **G** for novel goals, **A** for novel numbers of agents. **# Variants**: number of task variants involved.

2020; Perez-Liebana et al., 2019; Das et al., 2019). Despite notable progress, these systems have several limitations: 1) many of them target one or two-agent scenarios (Jain et al., 2019; Mandi et al., 2024; Wang et al., 2023a), 2) they are often limited to indoor settings with a narrow range of tasks (Puig et al., 2021; Zhang et al., 2024c), and 3) the task specifications are generally purely in text (Liu et al., 2022b; Mandi et al., 2024), making it hard to specify subtle task differences accurately and efficiently.

To drive progress in this area, we have developed a comprehensive benchmark, named *TeamCraft*, that features procedurally generated large-scale datasets specifically designed for multi-modal multi-agent systems. This benchmark utilizes the widely acclaimed open-world video game Minecraft as an experimental platform to engage with the complex dynamics of multi-modal multi-agent interactions. Inspired by the work of Jiang et al. (2022), we also leverage multi-modal prompts as task specifications to guide agent interactions, as language often fails to effectively convey spatial information (Cai et al., 2024). Our benchmark offers rich visual backgrounds, diverse object categories, complex crafting sequences, and varying task dynamics. These features enable systematic exploration of out-of-distribution generalization challenges for multi-modal, multi-task, multi-agent systems at scale. In particular, our benchmark evaluates a model's ability to generalize to novel goal configurations, unseen number of agents, novel agent capabilities, and new types of visual backgrounds. To evaluate existing techniques using our benchmark, we design several baseline models to work within the framework and compare their performance. Our results highlight that current approaches to vision-conditioned collaboration and task planning encounter significant challenges when tested within *TeamCraft*'s complex and dynamic environment, especially when it comes to generalizations.

In summary, the main contributions of this paper are:

- *TeamCraft*, a new multi-modal multi-agent benchmark with its associated large-scale dataset encompassing complex tasks challenging multi-agent systems in a wide variety of generalization scenarios.

- Extensive experiments and analyses on state-of-the-art multi-modal multi-agent models, uncovering their strengths and weaknesses to inform and inspire future research.

- To ensure reproducibility and encourage future work in the research community, we open source the entire platform, its training and evaluation code, and release the model checkpoints and training data at `https://github.com/teamcraft-bench/teamcraft`.

## 2 Related Work

**Embodied Language-Guided Benchmarks:** Several researchers have looked at the problem of using natural language as the interface between embodied agents, either in the form of task specifications (Shridhar et al., 2020b,a; Zheng et al., 2022; Gong et al., 2023b), question answering (Das et al., 2018; Gordon et al., 2018; Ma et al., 2023; Majumdar et al., 2024), instruction following (Anderson et al., 2018; Narayan-Chen et al., 2019; Jayannavar et al., 2020; Gao et al., 2022; Padmakumar et al., 2022; Wan et al., 2022; Gao et al., 2023), or as means of task coordination (Li et al., 2023; Mandi et al., 2024). VIMA-Bench (Jiang et al., 2022) builds on previous efforts in language-guided robotic

| | Building | Clearing | Farming | Farming | Smelting | Smelting |
|---|---|---|---|---|---|---|
| Scenes | village | snow_mountain | village | swamp | ice_on_water | desert_villge |
| Base | cyan_concrete | gold_block | hay_block | obsidian | oak_wood | glass |
| Goal | Build 1x2x4 building | Clean 3D building | Potato *3 | wheat *4 | cooked_mutton *1 | smooth_quartz *2 |
| Object | [dirt, wool, fence sandstone, sponge] | [grass_block, dirt birch_log, bookshelf,] | - | - | [birch_planks, sheep] | [oak_planks, quartz_block] |
| Agent | 3 | 3 | 2 | 2 | 3 | 2 |
| Inventory | [dirt, wool, fence, sponge,log, stone] | [stone_axe, stone_sword] | [carrot, beetroot] | [wheat_seeds, carrot, potato] | [iron_pickaxe, iron_axe, iron_sword] | [iron_pickaxe, iron_axe] |
| Demonstration |  |  |  |  |  |  |

Figure 1: We present example task configurations, as a combination of distinct biomes, playground base blocks, task goals, target blocks materials and agent counts. Agents are initialized with unique inventories, which provide them with different capabilities to complete various activities. A detailed distribution is provided in the supplementary.

manipulation (Zeng et al., 2020; Shridhar et al., 2021; Mees et al., 2022) and uses multi-modal prompts as uniform task specifications for object manipulation. *TeamCraft* extends multi-modal prompts to the multi-agent domain and uses them to specify a wide variety of collaborative tasks that require object interaction and navigation.

**Benchmarks Based on Minecraft:** Malmo (Johnson et al., 2016) marks the advent of a Gym-style platform tailored to Minecraft games. It paves the way for subsequent works such as MineRL (Guss et al., 2019), Voyager (Wang et al., 2023a), and MineDojo (Fan et al., 2022). Marlo (Perez-Liebana et al., 2019) extends Malmo to multi-agent scenarios, but the small number of task variations limit generalizations. Similar to our work, MindAgent (Gong et al., 2023c) and VillagerBench (Dong et al., 2024) focus on multi-agent collaboration in a multi-task setting. However, both of these use state-based observations, while *TeamCraft* tackles the more challenging problem of learning to collaborate from multi-modal perceptions. Table 1 compares TeamCraft with prior benchmarks.

## 3   TeamCraft Benchmark

### 3.1   Problem Formulation

Assume that an embodied multi-agent system comprised of $N$ agents needs to complete a complex task involving navigation and object manipulation. The task is specified in a multi-modal prompt $x_L = \{x_l\}_{l=1}^L$, which is a sequence of interleaved language and image tokens with length $L$. At time step $t$, each agent receives partial observation $o_n^t \in O$ from the full observation space $O$. To complete the task, each agent can choose to perform a high level action $a_t \in A$ from the full set of action $A$. The action can be further decomposed into a sequence of low level control signals.

### 3.2   Simulation Environment

*TeamCraft* utilizes Minecraft as its foundational simulation environment, offering a complex, open-world setting for multi-agent interactions. With a Gym-like environment, it facilitates the execution of intricate multi-agent commands via self-explanatory skills. Figure 3 illustrates the platform architecture. High level skills from the model can be translated into low level control signals via nested API calls through Mineflayer[3]. After execution, visual observation of each agent are rendered and provided as input to the model. Our simulation platform offers state-of-arts efficiency and scalability, detailed in the supplementary.

**Multi-Modal Prompts:** In our work, the multi-modal prompt $x_L$ consisting of a language instruction interleaved with a set of orthographic projection images (i.e. top, left, front views) for task specification. Depending on the specific task, the images can specify either the initial states, intermediate states or the goal states.

**Observation and Actions:** To mimic real world settings of embodied visual agent teaming, we use first-person view RGB image and inventory information as the observation $o_n$. The action space $A$ involves high-level self-explanatory skills such as *obtainBlock* to obtain a block and *farmWork* to farm a crop. Most actions take three input parameters, including 1) agent name such as *bot1*, as the action-executing entity, 2) item name such as *dirt*, 3) a 3D vector indicating the position of the target. There are 8 types of actions in total. A complete list of actions are described in the supplementary.

---

[3]https://github.com/PrismarineJS/mineflayer

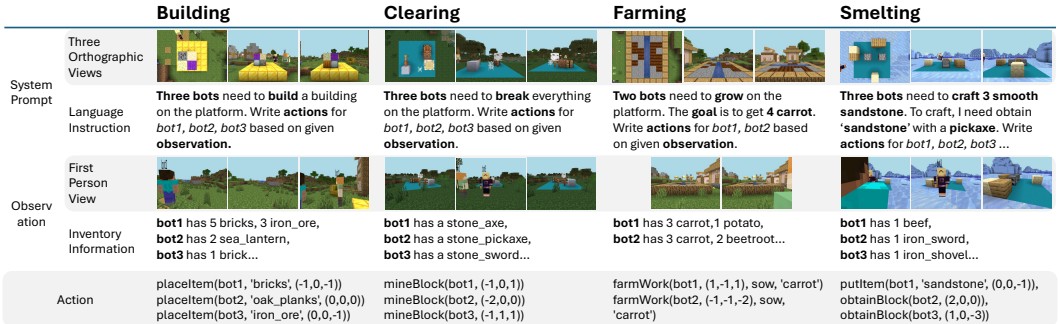

Figure 2: Multi-modal prompts are provided for all tasks. The system prompt includes both the three orthographic views and specific language instructions. Observations consist of first-person views from different agents, along with agent-specific information.

## 3.3 Task Design

*TeamCraft* introduces a variety of complex and interactive multi-agent cooperation tasks that challenge the agents' capabilities in planning, coordination, and execution within a collaborative and dynamic environment. Each task is designed to test different facets of MA interaction, including role distribution, real-time decision-making, and adaptability to changing environments. Task examples are shown in Figure 1 and the corresponding prompt examples are shown in Figure 2.

**Building:** Agents erect a structure based on a provided orthographic view blueprint. Each agent possesses a unique inventory of building blocks necessary for the construction. Successful completion requires visual cognition to associate blueprint components with inventory items, spatial reasoning to reconstruct a 3D structure from 2D images and map it to 3D coordinates for action targets, and collaborative coordination with other agents to resolve action dependencies. For example, an agent cannot place a floating block and should wait for another agent to build the supporting block first.

**Clearing:** Agents are required to remove all blocks from a specified area. Besides spatial understanding and awareness of action dependencies, agents will be given different tools and must employ appropriate tools to break blocks, which vary in durability, thereby requiring multiple interactions for complete removal. The assign correct agent with correct tools can dramatically reduce the time required to remove blocks. Thus agents must coordinate task assignments to optimize block-breaking efficiency. Strategic coordination is essential in this task as agents need to dynamically decide which blocks to target based on their assigned tools, and assist each other even without the optimal tools.

**Farming:** Agents sow and harvest crops on designated farmland plots. They must monitor crop growth stages, from newly planted to fully grown, and harvest only when crops reach maturity. Efficient task completion requires spatial reasoning to select appropriate farmland, visual cognition to assess crop maturity, and continuous updating of farmland states based on other agents' actions. As the available farmland exceeds what is needed, understanding other agents' actions to avoid redundancy, and dynamically allocating sub-tasks based on positions, available seeds, and crop maturity are essential. For example, some agents can sow while others are harvesting, stop when the total yield meets the goal.

**Smelting:** Agents obtain processed items using furnaces by gathering materials and coordinating actions. They collect resources from the environment, by harvesting blocks or killing mobs, or use existing inventory items to produce goal items like cooked food or refined materials. Agents also need to gather fuel before they can make use of furnaces. Efficient task completion requires spatial understanding to locate furnaces and resources, coordinating actions with inter-agent dependencies, and assigning task to agent who has appropriate inventory and tools. For instance, if one agent equipped sword is collecting beef, others without sword should focus on gathering fuel rather than duplicating efforts. Working as a team to use limited furnaces efficiently is crucial, rather than each agent independently smelting their own goal item.

## 3.4 Centralized and Decentralized Agents

**Centralized Agents:** The centralized model is given the observational data of all agents, including the first-person view, action history, and inventory information. Based on these comprehensive data,

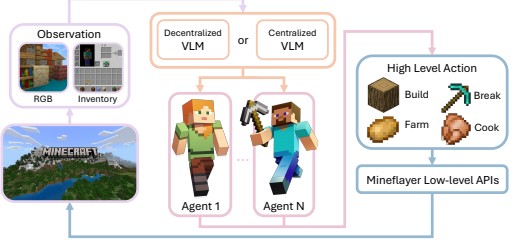

| | Building | Clearing | Farming | Smelting |
|---|---|---|---|---|
| # Action Sequences | 2 − 6 | 2 − 9 | 2 − 7 | 2 − 8 |
| # Agents | 2 − 3 | 2 − 3 | 2 − 3 | 2 − 3 |
| # Tools | – | 1 − 4 | – | 1 − 4 |
| # Scenes | 6 | 5 | 4 | 5 |
| # Base Types | 10 | 11 | 9 | 11 |
| # Furnaces | – | – | – | 1 − 2 |
| # Target Block Types | 19 | 16 | 3 | 13 |
| # Target Block Counts | 5 − 12 | 4 − 9 | 2 − 14 | 1 − 4 |
| # Fuel Types | – | – | – | 12 |
| # Resource Types | – | – | – | 20 |
| # Dimensional Shapes | 2 | 2 | 2 | 1 |
| # Placement Shapes | 7715 | 12724 | 13188 | 8885 |
| # Total Demonstrations | 14998 | 14641 | 14815 | 10803 |
| # Test Set | 50 | 50 | 50 | 50 |
| # Generalization Set | 200 | 200 | 150 | 200 |
| # Generalization Conditions | 4 | 4 | 3 | 4 |

Figure 3: (Left) The *TeamCraft* platform consists of three main components: 1) a Minecraft server that hosts online game, 2) Mineflayer as the interface for controlling agents in the server, and 3) a Gym-like environment that provides RGB and inventory observations to the models, allowing control of multiple agents through high-level actions. (Right) Task variants and dataset statistics

the model generates the actions for all agents simultaneously. This approach leverages the full scope of information in the environment to coordinate and optimize the actions of all agents collectively.

**Decentralized Agents:** The decentralized models do not receive information about other agents except for the initial inventory of the team. Each model generates actions solely for the individual agent based on its limited view. This setting simulates a more realistic scenario where agents operate independently with restricted information, focusing on their actions absent centralized coordination.

### 3.5 Diversity

The tasks are complex and challenging, testing multi-agent systems in diverse settings. Figure 3 provides task statistics and variants, with visual diversity detailed in supplementary.

**Object Diversity:** More than 30 target object or resource are used. Objects, such as fences, anvils, or stone, have different shapes and textures. Farm crops have different visual appearances during growth stage. Resources like chickens or rabbits have different appearances.

**Inventory Diversity:** Agent's inventory include essential items mixed with non-essential ones (i.e., distractors), realistically simulating scenarios where agents must choose the right materials for specific tasks while managing inventory constraints. Agents are provided with random tools for each task. Appropriate tools significantly enhances efficiency in tasks like clearing. For smelting, some resources must be collected by agent with specific tools.

**Scene Diversity:** More than 10 scenes are included in the tasks, covering biomes such as village, mountain, forest, swamp, desert, etc. Tasks take place on grounds with diverse textured bases such as glass, concrete, and quartz. Certain tasks may involve additional complexity, including farmland which are intermixed with non-plantable blocks.

**Goal Diversity:** Each task requires achieving a varying number of goal targets. Building requires different blocks placed into various shapes, categorized based on dimensionalities, e.g., 2D (all blocks are at the same level) or 3D (some blocks are on top of others). Farming requires various target crops and yields. For the smelting task, the target object is sampled from various food or processed items.

### 3.6 Tasks and Expert Demonstrations Generation

To create a rich learning environment and effective imitation learning dataset, systematic scenario design and data collection methods are employed, as follows:

**Task Generation:** Variables from a diversity pool, such as agent counts, scenes, and goals, are sampled to establish task configurations. Specifically, a solvable task is formulated by rejection sampling of the essential task variables. "Solvable" implies that the task can be completed within the Minecraft world rules and is within the agents' capabilities. For example, in smelting tasks, fuel must either be available to collect in the scene or directly accessible in the inventory.

**Planner-Based Demonstrations Generation:** Given the task specifications, a planner assigns actions to agents at every time step, utilizing privileged information of the environment. Assume agent $i$ performing action $j$, the planner optimizes a cost function designed to minimize total task completion time $T$, idle actions $E_i$, action dependencies $D$, redundant actions $U$, and the cost $c_{ij}$ for agent

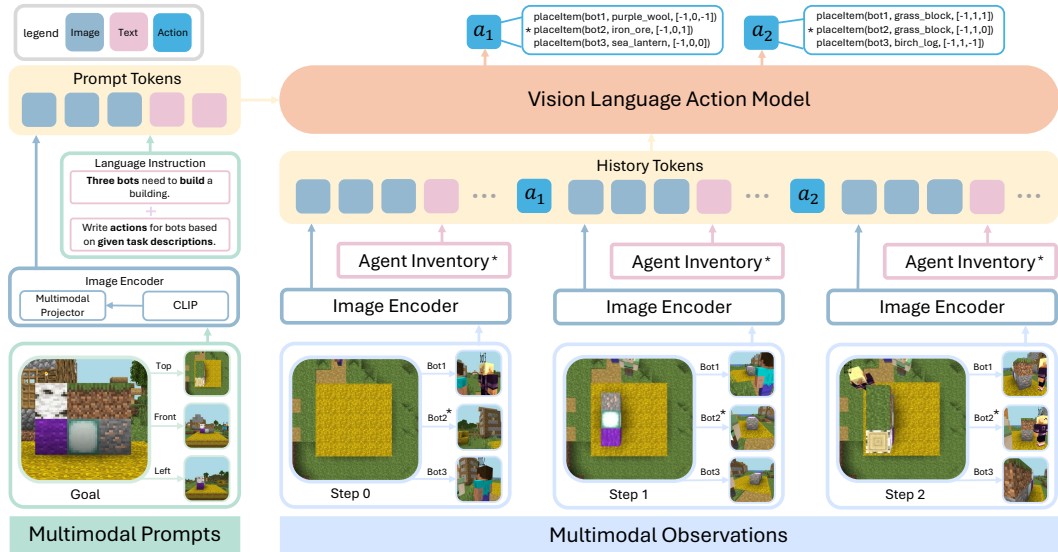

Figure 4: The architecture of the TeamCraft-VLA model. Multi-modal task specifications combining three orthographic views images of the task goal states and corresponding language instructions are encoded as initial input to the model. Agents inventories and visual observations are further encoded in each step to generate actions. For decentralized setting, the model only has access to one agent's information, exampled by Bot2: items associated with a * represent the fact that only the data associated with agent 2 are available.

$i$ performing action $j$: $C = w_1 T + w_2 \sum_{i=1}^{N} E_i + w_3 D + w_4 \sum_{i=1}^{N} \sum_{j \in A_i} c_{ij} + w_5 U$ , where $w_1, w_2, w_3, w_4, w_5$ are weighting coefficients. Details of the weights are in the supplementary.

As shown in Figure 3, we generated 55,000 unique task variants, each with one demonstration. A demonstration consists of a multi-modal prompt as task specification, including three orthographic view images representing task initial states or goal states and the corresponding language instructions. At each time step, agent inventories, first-person RGB observations and actions are recorded.

## 3.7 Test Set and Generalization Set

*TeamCraft* features a test set, where tasks and agents variables follow the same distribution as training. To evaluate the model generalization, we further designed a generalization set with hold-out elements excluded from training data. In general, we withheld test cases involving *four agents*, whereas the training data include only two or three agents. We also introduced unseen *scenes* not present during training. In addition to these general hold-outs, we implemented task-specific exclusions as following: 1) Building: novel *shapes* and *materials* to build. We exclude 8 block placement *shapes*, defining how target blocks are arranged on the ground. These shapes varied in complexity, containing 5 to 12 blocks in both 2D and 3D configurations. Additionally, we omitted 3 block *materials* appeared in clearing but not in building. 2) Clearing: novel *shapes* and *materials* to clear. We held out 6 block placement *shapes* with block counts ranging from 4 to 9. We also excluded 3 block *materials* present in building but absent in clearing. 3) Farming: novel *crops* to farm and collect. 4) Smelting: novel number of *furnaces* and *goal* objects. We excluded 4 unseen *goal* objects and introduced scenarios with novel number of *furnaces* in the scene. As shown in Figure 3, with 50 samples per task per each generalization condition, *TeamCraft* contains a total of 950 test cases.

## 4 Experiments

### 4.1 Baselines and Ablations

**TeamCraft-VLA:** We introduce TeamCraft-VLA, a multi-modal Vision-Language-Action (VLA) model designed for multi-agent collaborations. As shown in Figure 4, the model first encodes multi-modal prompts specifying the task, then encodes the visual observations and inventory information from agents during each time step to generate actions. Following Liu et al. (2024), the VLA model architecture consists of a CLIP encoder for images and a projector to align the image features with

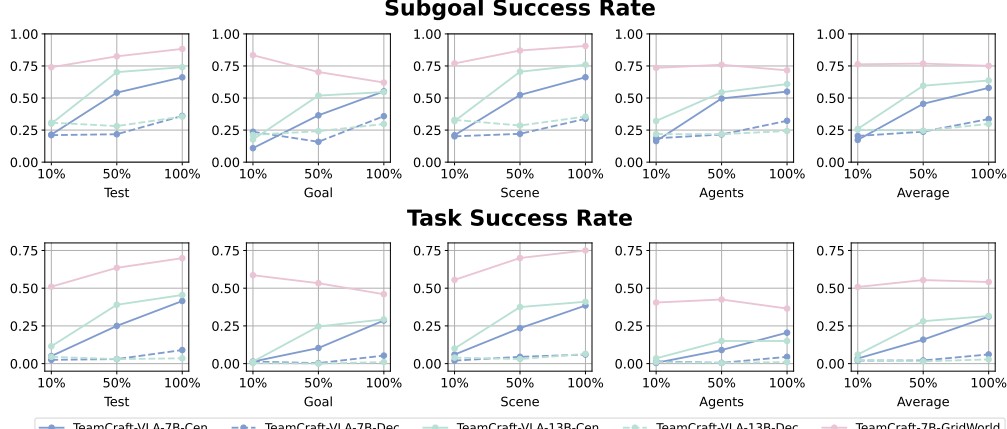

Figure 5: Subgoal success rate and task success rate across centralized, decentralized and grid-world settings. The leftmost column displays the *Test* category, which shares similar data distribution as training. The *Goal*, *Scene* and *Agents* categories represent generalization tasks involving unseen goals, scenes, and tasks involving four agents, respectively. Average performance is presented in the rightmost column.

the language model. We use CLIP ViT-L/14 as the visual encoder and a linear projector for modality alignment. The model is trained on demonstration data for three epochs before convergence.

**Grid-World Settings:** To understand the impact of learning in multi-modal environments as opposed to purely text-based or state-based environments, we perform an ablation study by translating the *TeamCraft* environments into a 3D grid-world. We retain the same prompt structure of the training data used in the TeamCraft-VLA models, with the main difference being that environmental information (i.e., visual observations and three orthographic view images) is now represented in text, describing the voxel coordinate of each block; e.g., "brick is at (2,3,0), stone is at (2,3,1)...". We fine-tuned an LLM in a centralized setting with variance in the dataset size (10%, 50%, and 100% of the total data) for three epochs before convergence.

**Proprietary VLA:** We evaluated four proprietary VLA models: GPT-4o, o4-mini, Claude 3.7, and Gemini-2.5-Pro, under two prompting settings: the *Vanilla* setting, which uses prompt structures similar to the centralized finetuned TeamCraft-VLA model, with additional information in the initial system prompt to supply sufficient task context, and the *Grid* setting, which is the same as the Grid World setting described above. This additional grounding compensates for the models' extremely limited 3D spatial reasoning capabilities, enabling them to skip orthographic projection images, and generate plans consistent with the true scene context. See the supplementary for detailed prompts.

**Ablations:** We performed a total of 15 ablation studies, varying in dataset sizes (10%, 50%, and 100% of the total data), control settings (centralized and decentralized), experiment settings (Multi-modal and Grid-World), and sizes of the VLA model (7B and 13B).

## 4.2 Evaluation Metrics

We evaluated the performance of the methods based on two key metrics: task success rate and subgoal success rate. With supplemental metrics: redundancy rate, action sequence length, and multi-agent effectiveness detailed in supplementary.

**Subgoal Success Rate:** This metric evaluates the effectiveness of agents in completing tasks. Given $M$ test cases, each test case $m$ has $s_m^g$ subgoals, and agents complete $s_m^d$ subgoals. The subgoal success rate $SGS$ is defined as $SGS = \frac{1}{M} \sum_{m=1}^{M} \frac{s_m^d}{s_m^g}$. Specifically, subgoals are designed based on the task requirements, i.e. the number of blocks to be built for building and the number of target objects to be created for smelting.

**Task Success Rate:** This metric indicates the proportion of test cases that the model can successfully complete from start to finish. Specifically, the task success rate $TS$ is defined as $TS = \frac{1}{M} \sum_{m=1}^{M} \mathbb{1}\left[s_m^d = s_m^g\right]$. A higher success rate reflects the model's ability to consistently achieve the desired outcomes in various scenarios.

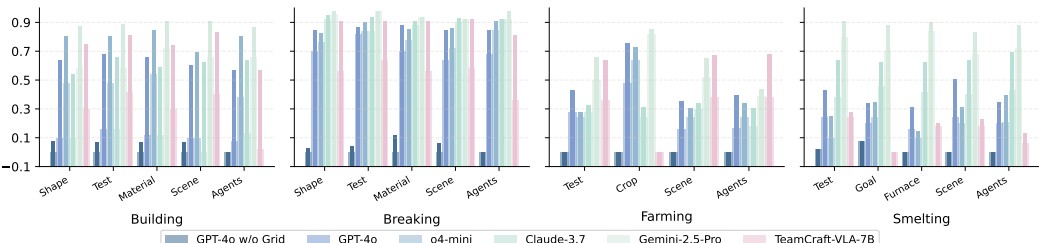

Figure 6: Task success rates (wide translucent bars) and subgoal success rates (narrow opaque bars within) for centralized models. Each clustered bar encodes two metrics for the same model. TeamCraft-VLA-7B and GPT-4o w/ Grid are tested under the Vanilla setting; other models are tested under the Grid setting (extra voxel info). Only TeamCraft-VLA is trained on the full dataset.

## 4.3 Evaluation Results

We evaluated the subgoal and task success rate of the models. As illustrated in Figure 5 and Figure 6.

**Success Rate**: For both the 7B and 13B models, the subgoal success rate and task success rate fall short of optimal performance. This is particularly evident in challenging tasks such as smelting, with both subgoal and task success rates below $40\%$. This highlights inherent difficulty of the designed tasks and current limitations of VLA models in handling multi-step, sequentially dependent processes.

**Across Model Size**: In Figure 5, we observe that as training data increases, the performance of the 7B model approaches that of the 13B model, especially when generalizing to novel goals and agents counts, so blindly scaling up model sizes does not guarantee success.

**Multi-Modal Environment vs. Grid-World**: The performance of the language model in the text-based Grid-World significantly surpasses VLA models in multi-modal settings. This suggests that state descriptions in text format are less challenging for models than multi-modal inputs, underscoring a notable gap in current VLA models' ability to effectively interpret visual information. For the language model, we observe a surprising trend in the *Goal* and *Agents* splits: training with more data lower the success rate. This decline suggests that the generalization capacity for certain task categories actually diminishes as training goes on. One possible cause is when exposed to more data, the model relies more heavily on patterns specific to the training examples, limiting its ability to adapt to unseen scenarios.

**On Generalization Splits**: For VLA models, performance generally drops when models transfer to novel generalization splits, especially in the *Goal* and *Agents* categories. The *Scene* split primarily tests image understanding, while the *Goal* and *Agents* splits emphasize task planning and allocation, critical factors in multi-agent systems. This indicates VLA models struggle with planning for unseen goals and adapting to variable numbers of agents.

**Scaling Law**: As training data increases, both subgoal and task success rates across centralized and decentralized settings significant improvements, underscoring the importance of dataset size for achieving better performance. The improvement is particularly pronounced when the training data increases from $10\%$ to $50\%$ in centralized settings. This suggests that while more data generally leads to better performance, gains diminish beyond a certain point, especially in the decentralized setting.

**Centralized vs. Decentralized**: Figure 5 compares centralized and decentralized settings in terms of subgoal and task success rates across all task variants. Centralized tasks outperform nearly all variants, highlighting the challenge of effective planning with partial information. This finding also demonstrates that multi-agent systems cannot be simplistically modeled as single agents interacting with environments containing other agents. In decentralized settings, the absence of agent modeling is particularly impactful, especially for cooperation-intensive tasks like "Farming" or "Building".

**Proprietary VLA — Vanilla**: Figure 6 shows GPT-4o in the Vanilla setting, which failed on almost all the test cases. While GPT-4o is able to associate blocks with their name, it struggles with mapping block coordinates based on visual inputs, demonstrating a lack of the 3D spatial reasoning necessary for accurate task execution. A block recognition test across GPT-4o, GPT-4.1, o1, o3, and o4-mini, along with a case analysis of o1 and Claude 3.7, confirmed that other proprietary models exhibit the same behavior, as detailed in the supplementary. This shortcoming severely impacts performance, since most of our tasks require precise spatial orientation and alignment. For example, in the building task, a brick should be placed at (1,1,1), while the output of the model is "placeItem(bot1, 'bricks', (1,0,1))", which results in wrong execution, and consequently affects the subsequent actions.

**Proprietary VLA — Grid**: As shown in Figure 6, all the models achieved strong performance, often matching or surpassing the trained TeamCraft-VLA. Among the proprietary models, Gemini 2.5 Pro led in both subgoal and task success rates, closely matching that of the trained TeamCraft-VLA in the grid world environment. The high success rate of the grid setting compared to the vanilla setting indicates the difficulty of orthographic projection from images. In particular, for the smelting task, which is characterized by higher action complexity and dependency, proprietary models with more parameters significantly outperformed TeamCraft-VLA, exhibiting greater ability in such complex, long-horizon tasks.

## 4.4 Qualitative Analysis

We performed a qualitative analysis across three generalization splits, examining how models handle novel goals, new scenes, and novel numbers of agents:

**Goals**: When faced with novel goals, the models struggle to generalize beyond familiar items and fail to adapt to unseen objectives. For example, in the "farming" task, if instructed to farm beetroot—a crop not encountered in training—the model generates a command "farm_work(bot1, (9,3,3), 'sow', 'beef')," causing Bot1 to sow "beef", which appears in the training data for "smelting". This behavior reflects the model's reliance on similar, seen items in the training data and reveals limited ability to infer new tasks based on similarity.

**Object State Recognition**: VLA models show strong generalization to new scenes, performing comparably to the *Test* set. However, errors often arise in recognizing object states. For example, in "farming" tasks, agents may harvest crops before they are fully grown due to challenges in identifying crop states, especially in new scenes. This highlights limitations in precise object state recognition when operating within unseen environments.

**Agents:** For generalization to four agents, models often ignore the fourth agent and assign inefficiently only to two or three agents. For example, in "building", the model output actions {"placeItem(bot1, ...)", "placeItem(bot2, ...)", "placeItem(bot3, ...)"} with the fourth agent overlooked, reducing productivity and preventing timely task completion. This limitation becomes especially evident in tasks requiring full coordination, such as "Building", where each of the four agents holds unique blocks in their inventory, and to complete the structure all agents must contribute some specific block that only they hold. The model's inability to command all agents leads to incomplete structures or outright task failure, highlighting limitations in coordination and workload distribution for collaboration.

## 5 Conclusions

We have presented *TeamCraft*, a benchmark for multi-modal multi-agent collaborative task planning in Minecraft. The benchmark consists of challenging collaborative tasks and evaluation splits designed to systematically test multi-modal agents across novel goal configurations, unseen numbers of agents, and unseen scenes.

## 5.1 Limitations and Future Work

We have conducted extensive experiments and analyses to pinpoint the limitations of the current models and identified promising research directions for collaborative multi-modal agents. (1) Given the limited capacity of existing multi-agent VLA models, *TeamCraft* relies on MineFlayer as an oracle controller to execute skills predicted by the models. Enabling VLA models to directly control multiple agents via low-level control (Wang et al., 2023c,b) would be important future research. (2) We have trained the models using procedurally generated multi-agent demonstration data. Learning from noisy but more diverse real-world demonstrations of human players can potentially further strengthen model generalization (Baker et al., 2022; Fan et al., 2022). (3) Currently, decentralized *TeamCraft* agents rely solely on implicit communication (Jain et al., 2019); i.e., by passively perceiving other agents and the environment, to gather information and to collaborate. Enabling agents to communicate explicitly via natural language (Narayan-Chen et al., 2019; Jayannavar et al., 2020; Mandi et al., 2024) has great potential in avoiding redundant actions and increasing efficiency. (4) Multi-player video games have been widely used as testbeds for human-AI collaboration (Carroll et al., 2020; Gao et al., 2020; Amresh et al., 2023). Extending *TeamCraft* with human players is a promising direction.

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

# Appendix

## A    Additional Related Work

**Platforms for Multi-Agent Systems:** The recent success of multi-agent reinforcement learning (MARL) methods (Lowe et al., 2020; Yu et al., 2021; Long et al., 2020, 2024b) has attracted attention, as these methods explore cooperation and competence behaviors among agents. However, many of the methods are evaluated in simplified 2D environments (Leibo et al., 2021; Suarez et al., 2021; Mordatch & Abbeel, 2017; Vinyals et al., 2019; Carroll et al., 2020). Recent work on embodied multi-agent benchmarks has considered more realistic tasks and environments (Liu et al., 2022a,b; Gong et al., 2023a; Park et al., 2023; Chang et al., 2024), but it often relies on certain privileged sensor information of the environment (Zhang et al., 2024b; Puig et al., 2021, 2023). Additionally, subject to environmental constraints, these works often have limited set of tasks (Jain et al., 2019; Tan et al., 2020) related to navigation and simple interactions such as object rearrangement (Szot et al., 2021). By comparison, TeamCraft is based on Minecraft, a three-dimensional, visually rich open-world realm characterized by procedurally generated landscapes and versatile game mechanics supporting an extensive spectrum of object interactions, providing rich activities ripe for intricate collaborations.

## B    High Level Skills

The action space of agents mainly involves high-level self-explanatory skills such as *obtainBlock* and *farmWork*. We provided 8 such skills. Most skills take three input parameters, including 1) agent name such as *bot1*, as the action executing entity, 2) item name such as *dirt*, which strongly associated with task goal or agent's inventory, 3) a vector indicating the position of the target on the test field.

For example, `obtainBlock(bot1, new Vec3(1, 0, 1))` takes the agent name `bot1` and a 3D vector `(1, 0, 1)` as its arguments. It directs `bot1` to perform multiple actions in Minecraft via APIs provided by Mineflayer. First, it controls `bot1` to `goto` a diggable position for block `(1, 0, 1)`, then has `bot1`'s vision ray cast to the block at `(1, 0, 1)` using the `lookAt` action. Next, it commands `bot1` to `equip` a proper tool that can dig the block at `(1, 0, 1)` most efficiently, and then instructs `bot1` to dig the target block. Once the target block has been mined, `bot1` will `goto` the position where the block item dropped and collect it.

Similarly, `farmWork(bot2, "sow", "potato", new Vec3(2, 0, 4))` takes the agent name `bot2`, action type `"sow"` (as opposed to `"harvest"`), crop seed item `"potato"`, and a 3D vector `(2, 0, 4)` as its arguments. It directs `bot2` to `goto` a placeable position for farmland at `(2, 0, 4)`, then `check` if the seed is a valid item—that is, a crop seed available within `bot2`'s inventory. It then `checks` if the farmland at `(2, 0, 4)` is plantable. Finally, it instructs `bot2` to `lookAt` the farmland and `sow` it with the seed `"potato"`.

Table 2 documents all the skills, which are implemented in JavaScript code with Mineflayer APIs.

| Type | Arguments | Description |
|---|---|---|
| placeItem | BotID, ItemType, Location | BotID places an item of ItemType at the specified 3D Location. |
| mineBlock | BotID, Location | BotID mines a block at the specified 3D Location. |
| farmWork | BotID, Location, Action, ItemType | BotID performs an Action (sow or harvest) on ItemType at the specified 3D Location. |
| obtainBlock | BotID, Location | BotID obtains a block from the specified 3D Location. |
| putFuelFurnace | BotID, ItemType, Location | BotID places an ItemType as fuel into a furnace at the specified 3D Location. |
| putItemFurnace | BotID, ItemType, Location | BotID inserts an ItemType into a furnace at the specified 3D Location. |
| takeOutFurnace | BotID, ItemType, Location | BotID removes an ItemType from a furnace at the specified 3D Location. |
| killMob | BotID, Location | BotID engages and eliminates a mob at the specified 3D Location. |

Table 2: Action space within the *TeamCraft*.

## C    Low Level Atomic Actions

High level skills are processed through multiple stages before reaching the final execution APIs. At each time step, *TeamCraft* accepts a list of skills as input, with a maximum length equal to the number

of agents involved in the current task and a minimum length of zero. Each agent can perform at most one skill per time step. The updated list of skills is then passed into the JavaScript environment along with the predefined atomic actions. Each atomic action is processed simultaneously, meaning that agents' actions are executed concurrently rather than sequentially. This avoid the dependency issue that might occur in sequential execution. For example, if one agent's action is executed ahead of another's, the first agent may block the location where the next agent intends to place a block. The agent whose atomic action is executed first will have a higher chance of success, potentially altering the dynamics of the multi-agent setting. Executing actions concurrently ensures fairness among agents and maintains the equivalence of the multi-agent environment.

## D    Simulator Efficiency, Scalability and Flexibility

*TeamCraft* offers dedicated simulation architecture and optimized infrastructure components achieve highly efficient and scalable simulation. Other Minecraft simulation platforms, suffer either deprecated support for multi-agent (MineRL) or architectural inefficiencies in agent scaling (MindAgent). Shown in Table 3, our codebase achieves a $13.6\%$ performance improvement over MindAgent, a multi-agent extension of Voyager. Overall, our system supports over 10 agents, 800 additional block types, 1,500 extra items, and 50 additional biomes, enabling researchers to design millions of new tasks.

| Average/Var | 2 Agents & 2 Actions | 3 Agents & 2 Actions | 4 Agents & 2 Actions |
|---|---|---|---|
| *TeamCraft* | 38.67 / 0.259 | 40.7 / 0.058 | 42.99 / 0.094 |
| MindAgent | 44.19 / 0.230 | 46.25 / 0.066 | 48.22 / 0.090 |
| Average/Var | 2 Agents & 10 Actions | 3 Agents & 10 Actions | 4 Agents & 10 Actions |
| *TeamCraft* | 40.95 / 0.456 | 42.33 / 0.102 | 44.43 / 0.055 |
| MindAgent | 45.41 / 0.061 | 47.77 / 0.160 | 49.33 / 0.076 |

Table 3: Simulation Speed comparison. Units in seconds. Naive movement action. Ubuntu 22.04 with 1x Intel i9-14900KF

## E    Visual Diversity

*TeamCraft* uses a set of visual variate to provide a visual rich environment. Each task is constructed from a random number of agents, in a randomly selected scene, achieving different goal on playground built by different base block.

### E.1    Shared Elements

Each task begins with a basic setting involving multiple agents on a playground. Each agent has a unique skin, as illustrated in Figure 17, and is rendered as a two-block-high character. The playground combines a base platform spawned within a Minecraft biome. The base block is also randomly selected from a pool,shown in Figure 17. Each biome offers variations in special surrounding blocks, designs, and environments.

For example, the `seaside village` biome is a village near the sea with houses made of oak wood and cobblestone, decorated with flowers and cow sheds, as shown in Figure 22. It also features a nearby farm surrounded by oak logs (Figure 23). Another variation of village is the `desert village` biome, built from acacia planks, acacia logs, and sandstone, blending seamlessly with the desert's arid terrain, shown in Figure 24. Figure 25 illustrates a biome that is located on half of the mountains, where a small flat land protruding from a cliff. Additional examples of biomes used are shown in Figure 26, Figure 27, and Figure 28.

### E.2    Task Specific Diversity

**Clearing** task uses a random set of blocks as its targets, illustrated in Figure 18. **Building** task also uses a random set of blocks as its target, with some blocks shared with clearing task, as illustrated

in Figure 19. Unlike other tasks, the **Farming** task does not use a regular base. The playground is constructed from a combination of farmland for planting crops, water blocks, and randomly selected unfarmable blockers from the base that replace some of the farmland. An example is shown in Figure 33. Each corps used in farming task has its own grown stage with different appearances, shown in Figure 20. **Smelting** task requires a wide varieties of resources to achieve its goal. Resources could be either entity, block, or item. Shown in Figure 21. Detailed statistics of each task is presented in Table 14, Table 15, Table 16 and Table 17.

# F Planner for Expert Demonstration

*TeamCraft* employed a planner to assign actions to each agent at every time step, utilizing perfect knowledge of the task including goal object positions, agents' inventories, and each agent's efficiency in performing actions. The planner optimizes actions using a cost function designed to minimize the total time to complete the task, reduce idle times for agents, minimize action dependencies to prevent agents from waiting on others, maximize parallelism of actions, assign tasks to the most efficient agents, and eliminate redundant or unnecessary actions. The cost function considers the following components:

**Minimize Total Task Completion Time** $T$: Denoted by $\min T$, our primary objective is to reduce the overall time required to complete the task, measured in time steps until the last agent finishes their final action.

**Minimize Idle Actions for Each Agent** $E$: Denoted by $\min \sum_{i=1}^{N} E_i$, we minimize the total empty actions, the sum of empty action $E_i$ preformed by agent $i$.

**Minimize Action Dependencies Across Agents** $D$: Denoted by $\min D$, we minimize dependencies cause agents to wait for others to complete certain actions.

**Minimize Redundant or Useless Actions** $U$: Denoted by $\min U$, we minimize the total number of redundant or unnecessary actions performed by all agents.

**Maximize Action Efficiency**: Denoted by $\min \sum_{i=1}^{N} \sum_{j \in A_i} c_{ij}$, we assign actions to agents with higher capabilities to reduce the overall cost, where $c_{ij}$ be the cost (inverse of efficiency) for agent $i$ to perform action $j$.

We assign each component a weight:

$$C = w_1 T + w_2 \sum_{i=1}^{N} E_i + w_3 D + w_4 \sum_{i=1}^{N} \sum_{j \in A_i} c_{ij} + w_5 U$$

where $w_1, w_2, w_3, w_4, w_5$ are weighting coefficients, and adjusted for each tasks.

**Building**: In the building task, where dependencies are moderate and parallelization is preferred, we place greater emphasis on minimizing idle actions by setting $w_2 = 1.4$ and assign a weight of 0.9 to the other components. This encourages agents to remain active and reduces idle time, enhancing overall efficiency.

**Clearing**: In the clearing task, using the correct tools can significantly speed up block removal (up to a threefold increase). Therefore, we assign a higher weight of $w_4 = 1.8$ to maximize action efficiency by assigning tasks to the most capable agents. The other weights are set to 0.8 to maintain overall performance while focusing on efficient tool usage.

**Farming**: Farming task is not heavily constrained by action dependencies, we assign equal weights of 1 to all components, ensuring a balanced consideration of time minimization, idle actions, action dependencies, action efficiency, and redundancy elimination.

**Smelting**: In the smelting task, which involves comparatively long and highly dependent action sequences, we prioritize minimizing action dependencies by setting $w_3 = 1.8$. The other weights are assigned a value of 0.8 to support this focus, facilitating smoother coordination among agents and reducing waiting times.

### F.1 Example Expert Demonstrations

*TeamCraft* will provide a multi-modal prompt alone with a three orthographical view for each variant, each task. A prompt contains information of the task goal, inventory of each agent, and necessary information to complete the task. For each time step, only textual inventory information, first perspective view image for each bot, and a three orthographical view will be provided. Sky view image shown in each step is just for clarity and will not be provided to any of the tested models.

Figure 29 and Figure 30 show a classic example of the building task, which involves three agents building a 2x3 building on the mountain half. Each of the agents has some of the needed blocks in their inventory to build the building. For every time step after step 0, each of the three agents build one block, from bottom level to the second level.

Figure 31 shows an example of the clearing task. Two agents are assigned to clean the blocks on a 6x6 platform. Each of them has a stone pickaxe in their inventory, which is the efficient tool to break "stone-like" blocks. In this case, they are able to break brick and sandstone in just one time step with pickaxe but requires two time step to break "wood-made" blocks like bookshelf and crafting table. This resulted time step 2 and 3 has exactly same visual observation, shown in Figure 32.

Figure 33 and Figure 34 shows an example of two agents farming on a snow mountain for two extra carrots. In step 1, agent1 and agent2 both sow the carrots on the open ground. In step 2 they saw that the carrots are ready to collect and they both collect one carrot in step 3 and eventually they collected two carrots.

Figure 35 and Figure 36 shows an example of smelting task where two agents need to get two cooked porkchops. In step 1, one agent is in charge of adding the fuel to the furnace and the other agent tries to kill the pork to get the raw porkchop. Since bot2 already has one porkchop, it only requires one additional porkchop. In step 2, both agents put the porkchop to the furnace and in step 3, they got 2 cooked porkcops.

## G  Grid-World Settings

Under the grid-world setting, we replace the three orthographic view images and first person view images with text descriptions of the task goal and current environment states, and provide them as input to the model. Here we show one example of the prompt construction in each task.

**Building**: As shown in Figure 13, the system prompt consists of both task description and the target building coordination of each block. The user prompt consists of the built blocks and the inventories of the agents.

**Clearing**: As shown in Figure 14, the system prompt consists of both task description and the blocks that appeared on the platform initially. The user prompt consists of the blocks that appeared on the platform at current time step and the inventories of the agents.

**Farming**: As shown in Figure 15, the system prompt consists of both task description and the blocks in the farmland. The user prompt consists of the blocks in the farmland and crops information at current time step and the inventories of the agents.

**Smelting**: As shown in Figure 16, the system prompt consists of both task description, instructions to craft different items and the blocks in the field. The user prompt consists of the blocks locations at current time step and the inventories of the agents.

## H  TeamCraft-VLA Implementation Details

We use Vicuna-v1.5 as the LLM backbone. For the visual encoder, we employ CLIP ViT-L/14 to process all input images, including three orthogonal views and the first-person view of the agents. The image embeddings are then projected into the LLM space with a linear projection layer and concatenated with the text embeddings. The combined embeddings are fed into the LLM, which outputs the final action. During training, we froze the visual encoder and projector and only finetune the LLM. All image embeddings are positioned before the text embeddings, separated by "image start" and "image end" tokens. In centralized settings, where the number of images varies depending on the number of agents, we pad a dummy image at the end for training stability if the task involves

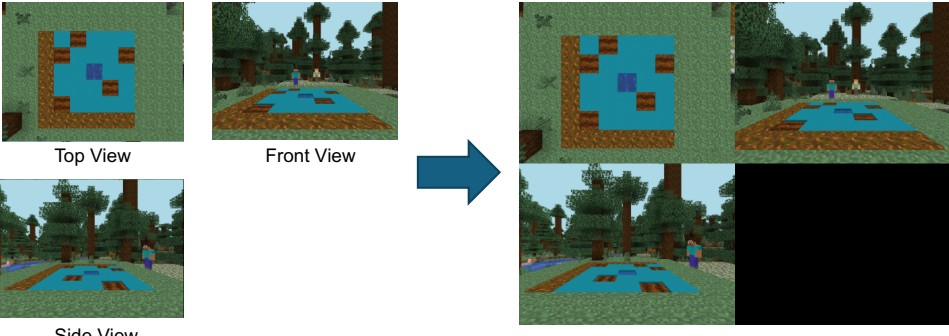

Figure 7: Combining three orthogonal view images into a single composite image as model input.

only two agents. In decentralized settings, the number of image inputs remains unaffected, as the model processes only the first-person view of the current agent, excluding views from others.

We train each model for 3 epochs using the training split, leveraging 8 A100 GPUs with a global batch size of 16. In the centralized setting, training the 7B model takes 36 hours, while the 13B model requires 72 hours. In the decentralized setting, the training duration doubles, with the 7B model requiring 72 hours and the 13B model taking 144 hours. In the grid-world setting, training the 7B model takes 20 hours.

### H.1 Arrangement of Three Orthogonal Views

For training and evaluation, we combine the three orthogonal view images into a single composite image by arranging them to the upper-left top-left corner, top-right corner, and the lower-left corner of the composite image. An example of this arrangement is shown below Figure 7. This process is to reduce the number of images provided to the model to conform with the 4096 context length limit.

### H.2 Hyperparameters

We present the hyperparameters for VLA training in Table 4.

| lr | model max length | vision tower | patch size | resolution | language model | optimizer | lr scheduler type | warmup ratio |
|---|---|---|---|---|---|---|---|---|
| 2e-5 | 4096 | openai-clip-vit-large | 14 | 336*336 | Vicuna-v1.5 | AdamW | constant_with_warmup | 0.03 |

Table 4: Hyperparameters for TeamCraft-VLA

### H.3 Model Output Parsing

The output of the model is a string which will be parsed into the pre-defined high level skills. The string will be first processed by removing special sentence begin token, ``, and ending token ``. It will then be split into a list, where each item is parsed as the skill of one agent.

## I  Additional Results of TeamCraft-VLA

### I.1 Task Success Rate and Subgoal Success Rate

We show task success rate and subgoal success rate of centralized and decentralized 7B models with different data scales in Table 10, and those of 13B models in Table 11. We compare among different centralized models in Table 12.

### I.2 Redundancy Rate

This metric assesses whether multiple agents are performing the same action at the same time, which would lead to conflicts. Assume $p_m$ is the total number of actions for test case $m$ and $q_m$ the number of conflicts between agents, the redundancy rate $RR$ is defined as:

|  | Test | Goal | Scene | Agents | Average |
|---|---|---|---|---|---|
| TeamCraft-VLA-7B-Cen | 0.01 | 0.02 | 0.01 | 0.01 | 0.01 |
| TeamCraft-VLA-13B-Cen | 0.01 | 0.00 | 0.01 | 0.01 | 0.01 |
| TeamCraft-VLA-7B-Dec | 0.13 | 0.12 | 0.13 | 0.24 | 0.15 |
| TeamCraft-VLA-13B-Dec | 0.11 | 0.11 | 0.12 | 0.22 | 0.14 |

Table 5: Comparison of TeamCraft-VLA redundancy rates.

|  | Test | Goal | Scene | Agents | Average |
|---|---|---|---|---|---|
| TeamCraft-VLA-7B-Cen | 6.62 | 7.63 | 5.93 | 6.35 | 6.63 |
| TeamCraft-VLA-13B-Cen | 6.25 | 7.44 | 6.46 | 6.47 | 6.65 |
| TeamCraft-VLA-7B-Dec | 8.42 | 8.53 | 8.06 | 7.38 | 8.1 |
| TeamCraft-VLA-13B-Dec | 8.62 | 8.46 | 8.41 | 6.71 | 8.04 |

Table 6: Comparison of TeamCraft-VLA action sequence length.

$$RR = \frac{1}{M} \sum_{m=1}^{M} \frac{q_m}{p_m}$$

A lower redundancy rate indicates better task allocation among agents and a higher level of cooperative efficiency.

Table 5 compares redundancy rates between centralized and decentralized settings. Our results show that decentralized agents exhibit significantly higher redundancy rates than centralized agents, indicating reduced efficiency in task planning and allocation. This inefficiency becomes even more pronounced as the number of agents increases, creating greater challenges for effective task allocation. In decentralized settings, the absence of centralized control complicates the avoidance of redundant work, as each agent must independently infer the intentions of others to prevent duplication. By contrast, a centralized controller can efficiently assign distinct tasks to each agent, minimizing overlap and enhancing overall efficiency. Both the 7B and 13B models exhibit redundancy issues in decentralized settings. Increasing model size alone does not resolve the redundancy problem in such scenarios. These findings suggest that VLA models lack explicit mechanisms to understand or infer the actions of other agents, highlighting a critical need for improved inter-agent communication and awareness within decentralized systems.

### I.3 Action Sequence Length

We compared the average action lengths across different splits between the 7B and 13B models under both centralized and decentralized settings, as shown in Table 6. In general, decentralized settings require longer action sequences to complete tasks. Among the splits, the *Goal* split is the most challenging, as it demands more actions to accomplish the tasks.

### I.4 Multi-Agent Effectiveness

Table 7 comparing success rates and average steps under centralized and decentralized settings. Overall, efficiency improves with more agents, which result a fewer steps to complete the task. In the centralized setting, 3 agents exhibited better task completion. In the decentralized one, 2 agents achieved a higher success rate, indicating the growing difficulty and complexity of coordination in decentralized settings with more agents.

### I.5 Case Study

We present a detailed failure cases analysis by categories.

**Object Mismatching**: As an example (Figure 8), in the farming tasks two agents need to get 10 beetroot. In step 0, the actions involve a mismatch in the objects; the agents mistakenly sow "beet_seeds" instead of "beetroot_seeds." Consequently, in step 1, due to the object mismatch, no crops grow on the farmland. As another example, two agents need to get 2 dried kelp in the smelting

| | Agent # | Task Success Rate | Subgoal Success Rate | Average Steps |
|---|---|---|---|---|
| TeamCraft-VLA-7B-Cen | 2 Agents | 0.506 | 0.722 | 7.487 |
| TeamCraft-VLA-7B-Cen | 3 Agents | 0.540 | 0.755 | 5.744 |
| TeamCraft-VLA-7B-Dec | 2 Agents | 0.075 | 0.427 | 9.764 |
| TeamCraft-VLA-7B-Dec | 3 Agents | 0.074 | 0.403 | 7.536 |

Table 7: Comparison on the number of agents.

task (Figure 9). The task requires one bot to put the kelp and the other put the fuel. However, in this example bot1 mistake the object "kelt" to "cobbl1".

**Task Allocation Failure**: This occurs when a task requires four agents. As two examples, four agents must break everything on the platform in the clearing task (Figure 10), and construct on the platform in the building task (Figure 11). Only three agents are assigned distinct actions, leaving the fourth agent idle.

**Object State Recognition Failure**: As an example (Figure 12), a farming task requires two agents to collect four additional carrots. In step 0, *bot1* and *bot2* both sow carrots and attempt to harvest them in step 2. However, at that time, the carrots are still immature and not ready for collection. The mature state of the carrot is shown in Figure 20.

| Company | Model | Name |
|---|---|---|
| OpenAI | GPT-4o | gpt-4o-2024-08-06 |
| OpenAI | o4-mini | o4-mini-2025-04-16 |
| Anthropic | Claude 3.7 | claude-3-7-sonnet-20250219 |
| Google | Gemini 2.5 Pro | gemini-2.5-pro-preview-05-06 |

Table 8: Proprietary VLA Models

# J   Proprietary VLA Model Implementations

We use GPT-4o, o4-mini, Claude 3.7, and Gemini 2.5 Pro as the proprietary VLA models. Details are shown in Table 8. We test proprietary VLA models in two prompting settings: basic setting and grid setting. Table 13 shows the detailed results.

## J.1   Vanilla Setting

The Vanilla setting uses similar prompt structures as the centralized finetuned TeamCraft-VLA model, with additional information in the initial system prompt to supply sufficient task context.

Specifically, all proprietary models share the same system prompt that (i) specifies the task objective, (ii) enumerates observation inputs, (iii) stipulates the action and response syntax, (iv) lists all possible blocks/items names that will be used in current task, but may not being used for a task variance, (v) details workspace constraints together with coordinate-axis and orientation conventions, and (vi) provides a fully worked roll-out for a closely related task. Figure 37 shows the system prompts for the building task; Figure 40 shows the system prompts for the clearing task; Figure 42 shows the system prompts for the farming task; Figure 45 and Figure 46 show the system prompts for the smelting task.

Based on the system prompts and user prompts, the model predicts the actions. As the interaction progresses with subsequent prompts, the context is maintained and expanded with the addition of prior responses and updated visual data. In the first step, we additionally provide the first user prompt, where the model is given a specific multi-modal task specification accompanied by initial visual observations and inventory details of the agents. Based on the system prompts and user prompts, the model predicts the actions. As the interaction progresses with subsequent prompts, the context is maintained and expanded with the addition of prior responses and updated visual data, as shown in Figure 38 and Figure 39.

Under the Vanilla setting, results from GPT-4o (as shown in Table 13), along with case studies on GPT-4o, Claude 3.7, and o1, as well as block recognition tests involving GPT-4o, GPT-4.1, o1, o3, and o4-mini, collectively indicate that the proprietary VLA models suffer a lack of 3D spatial reasoning necessary for accurate task execution.

### J.1.1 GPT-4o Error Analysis

Some failure cases are visualized in Figure 38, Figure 39, Figure 41, Figure 43, and Figure 44. In general, GPT4-o fails to understand spatial relations and often chooses the incorrect coordinates as the locations for placing and mining actions. In addition, the model sometimes fails to follow instructions and does not harvest the crop in the farming task, as shown in Figure 43 and Figure 44.

### J.1.2 o1 Error Analysis

We evaluated a subset of our test cases using o1-2024-12-17, a reasoning model that produces a chain-of-thought at inference time. To accommodate its extended context length allowance, we provided o1 with a more comprehensive system prompt, as illustrated in Figure 47. We also included a one-shot example of a question and answer sequence, which is placed between the system prompt and the actual test tasks, to clarify our specialized coordinate system (Figure 48).

Despite its ability to identify blocks accurately from the goal image, as is shown by Figure 49), o1 exhibits significant deficiencies in adhering to task-specific constraints and performing reliable 3D spatial reasoning. Notably, it even violates the system-level instruction to generate exactly one action per agent, instead producing two actions, which leads to partial or unintended command execution. Additionally, o1's misinterpretation of absolute coordinates results in incorrect placements; for example, positioning a `sea_lantern` at $(0, 0, 0)$ rather than one block below the origin. Figure 49 and Figure 50 further demonstrate o1's struggle to reconcile multiple orthographic views to infer vertical stacking, causing it to consistently place blocks at ground level instead of at the correct elevated layer. Although o1 occasionally succeeds at recognizing lateral orientation from a first-person viewpoint (e.g., determining the "right side" of a reference point), its broader challenge in synthesizing 2D and 3D cues compromises the precise placement of blocks. Consequently, these spatial miscalculations substantially diminish o1's efficacy in tasks demanding fine-grained alignment, thus revealing the model's limitations in translating visual information into spatially coherent actions.

### J.1.3 Claude 3.7 Error Analysis

We evaluated claude-3-7-sonnet-20250219 under the same conditions as the o1 model, employing a comparable prompt configuration and environment constraints, as shown in Figure 51 and Figure 52. As depicted in Figure 55 and Figure 56, Claude 3.7 accurately identifies blocks from the goal image and associates them with their corresponding names. Moreover, it adheres to the system prompt's requirements by generating exactly one well-formed action per agent, thus meeting all specified directives.

In Figure 53 and Figure 54, Claude 3.7 further demonstrates an ability to parse multiple orthographic views, correctly placing blocks on the first layer prior to stacking additional blocks on top. However, Figure 53 shows that the model's spatial reasoning weakens when interpreting first-person orientation cues: rather than arranging blocks horizontally, it produces a vertical configuration. This shortfall becomes more evident in Figure 54, where Claude 3.7 consistently positions a `sea_lantern` on the right side of the existing blocks instead of at the top. Similarly, when more blocks are placed, Claude 3.7 starts to struggle in tracking the special relationship of the blocks on the playground, and begins placing block at ground level instead of in the intended second layer.

As illustrated by Figure 55, Figure 56, and Figure 57, Claude 3.7 eventually begins to place surplus blocks from its inventory in a random manner, ultimately filling a substantial portion of the play area with extraneous structures.

### J.1.4 Block Recognition Test

We designed a dedicated test to evaluate proprietary VLA models on their 3D spatial reasoning abilities. The models are provided with orthographic projection images and are asked to identify all visible blocks along with their coordinates. As in the Vanilla setting, we provide detailed task

| Model | Avg. Accuracy (↑) | Block Type Accuracy (↑) | False Positive Rate (↓) |
|---|---|---|---|
| GPT–4o | $0.0507 \pm 0.0121$ | $0.4315 \pm 0.0551$ | $0.2363 \pm 0.0764$ |
| GPT–4.1 | $0.1175 \pm 0.0618$ | $0.6519 \pm 0.0423$ | $0.2734 \pm 0.0653$ |
| o1 | $0.0220 \pm 0.0156$ | $0.1296 \pm 0.0805$ | $0.0523 \pm 0.0336$ |
| o3 | $0.0703 \pm 0.0018$ | $0.6659 \pm 0.0896$ | $0.1788 \pm 0.0180$ |
| o4-mini | $0.0124 \pm 0.0098$ | $0.1350 \pm 0.0689$ | $0.4893 \pm 0.1169$ |

Table 9: Overall comparison of model performance. Higher values are better for Accuracy and Block Recognition Accuracy, while lower values are better for False Positive Rate.

instructions, including the list of all possible block/item types, the coordinate system and axis orientation, and a sample input-output pair to enable single-shot learning.

We define three evaluation metrics:

**Average Accuracy ($A_{\text{pos}}$):** Measures the fraction of blocks correctly identified by both name and position:

$$A_{\text{pos}} = \frac{N_{\text{correct\_name\_and\_position}}}{N_{\text{total}}},$$

where $N_{\text{correct\_name\_and\_position}}$ is the number of blocks correctly predicted in both type and 3D location, and $N_{\text{total}}$ is the total number of blocks in the image.

**Block Type Accuracy ($A_{\text{type}}$):** Evaluates how well the model identifies block types and their counts, regardless of position:

$$A_{\text{type}} = \frac{N_{\text{correct\_types}}}{N_{\text{total}}},$$

where $N_{\text{correct\_types}}$ is the number of block types correctly identified with the correct count. Over-reporting a block type (e.g., reporting two gold blocks when only one exists) results in only the first being counted as correct.

**False Positive Rate ($FPR$):** Measures the proportion of reported blocks that do not exist in the scene, based on type:

$$FPR = \frac{N_{\text{false\_positives}}}{N_{\text{total}}},$$

where $N_{\text{false\_positives}}$ is the number of reported blocks of types not present in the image.

As shown in Table 9, all models achieve an average accuracy $A_{\text{pos}} < 0.1$, revealing poor performance in 3D spatial reasoning and localization. However, their low false positive rate ($FPR$), attributed to texture-based type identification, suggests that the models can still recognize block types with high visual fidelity due to their training on Minecraft data.

## J.2 Grid Setting

The Grid setting retains the same structures as the basic setting but supplies extra grid information that translates the orthographic projection images to text, by describing the voxel coordinate of each block; e.g., "brick is at (2,3,0)". The grid information text is similar to the user prompt in Figure 16, but only appears once at the very first user prompt to describe orthographic projection images. This additional grounding compensates for the models' extremely limited 3D spatial reasoning abilities, enabling them to skip orthographic projection images and generate plans consistent with the true scene context.

| Tasks | Condition | Centralized | | | Decentralized | | |
|-------|-----------|-------------|---|---|---------------|---|---|
| | | 10% | 50% | 100% | 10% | 50% | 100% |
| Building | Test | 0.00 (0.12) | 0.38 (0.76) | 0.42 (0.81) | 0.00 (0.18) | 0.00 (0.28) | 0.00 (0.38) |
| | Shape | 0.00 (0.12) | 0.20 (0.67) | 0.30 (0.75) | 0.00 (0.15) | 0.00 (0.25) | 0.00 (0.40) |
| | Material | 0.00 (0.13) | 0.18 (0.64) | 0.30 (0.74) | 0.00 (0.13) | 0.00 (0.20) | 0.00 (0.34) |
| | Scene | 0.00 (0.15) | 0.36 (0.73) | 0.40 (0.83) | 0.00 (0.16) | 0.00 (0.21) | 0.00 (0.36) |
| | Agents | 0.00 (0.18) | 0.02 (0.50) | 0.02 (0.57) | 0.00 (0.12) | 0.00 (0.20) | 0.00 (0.14) |
| Clearing | Test | 0.00 (0.13) | 0.08 (0.43) | 0.64 (0.91) | 0.00 (0.45) | 0.02 (0.35) | 0.20 (0.68) |
| | Shape | 0.00 (0.09) | 0.08 (0.34) | 0.56 (0.91) | 0.00 (0.47) | 0.02 (0.27) | 0.16 (0.74) |
| | Material | 0.00 (0.10) | 0.12 (0.45) | 0.56 (0.90) | 0.00 (0.48) | 0.00 (0.22) | 0.16 (0.67) |
| | Scene | 0.00 (0.11) | 0.10 (0.44) | 0.58 (0.92) | 0.00 (0.41) | 0.04 (0.37) | 0.10 (0.64) |
| | Agents | 0.00 (0.16) | 0.14 (0.64) | 0.36 (0.81) | 0.02 (0.50) | 0.02 (0.54) | 0.12 (0.60) |
| Farming | Test | 0.14 (0.43) | 0.34 (0.60) | 0.36 (0.63) | 0.02 (0.07) | 0.02 (0.14) | 0.00 (0.09) |
| | Crop | 0.00 (0.00) | 0.00 (0.00) | 0.00 (0.00) | 0.00 (0.00) | 0.00 (0.00) | 0.00 (0.00) |
| | Scene | 0.16 (0.39) | 0.34 (0.65) | 0.38 (0.67) | 0.00 (0.05) | 0.00 (0.11) | 0.02 (0.07) |
| | Agents | 0.02 (0.18) | 0.18 (0.61) | 0.38 (0.68) | 0.00 (0.08) | 0.00 (0.11) | 0.04 (0.27) |
| Smelting | Test | 0.06 (0.17) | 0.20 (0.36) | 0.24 (0.28) | 0.08 (0.13) | 0.08 (0.09) | 0.16 (0.29) |
| | Goal | 0.08 (0.21) | 0.04 (0.07) | 0.00 (0.00) | 0.08 (0.17) | 0.00 (0.00) | 0.00 (0.00) |
| | Furnace | 0.10 (0.28) | 0.10 (0.20) | 0.18 (0.20) | 0.06 (0.07) | 0.06 (0.06) | 0.06 (0.16) |
| | Scene | 0.08 (0.19) | 0.14 (0.28) | 0.18 (0.23) | 0.08 (0.19) | 0.14 (0.19) | 0.12 (0.28) |
| | Agents | 0.00 (0.15) | 0.02 (0.24) | 0.06 (0.13) | 0.04 (0.05) | 0.00 (0.02) | 0.02 (0.28) |

Table 10: Task success rates and subgoal success rates of the TeamCraft-VLA-7B-Cen and TeamCraft-VLA-7B-Dec models. Subgoal success rates are given in parentheses.

| Tasks | Condition | Centralized | | | Decentralized | | |
|-------|-----------|-------------|---|---|---------------|---|---|
| | | 10% | 50% | 100% | 10% | 50% | 100% |
| Building | Test | 0.00 (0.18) | 0.46 (0.80) | 0.48 (0.79) | 0.00 (0.13) | 0.00 (0.18) | 0.00 (0.31) |
| | Shape | 0.00 (0.16) | 0.30 (0.73) | 0.26 (0.69) | 0.00 (0.15) | 0.00 (0.15) | 0.00 (0.32) |
| | Material | 0.00 (0.15) | 0.24 (0.65) | 0.08 (0.63) | 0.00 (0.14) | 0.00 (0.14) | 0.00 (0.31) |
| | Scene | 0.00 (0.16) | 0.38 (0.75) | 0.48 (0.83) | 0.00 (0.17) | 0.00 (0.17) | 0.00 (0.28) |
| | Agents | 0.00 (0.16) | 0.00 (0.49) | 0.04 (0.59) | 0.00 (0.14) | 0.00 (0.16) | 0.00 (0.23) |
| Clearing | Test | 0.04 (0.37) | 0.42 (0.83) | 0.64 (0.94) | 0.00 (0.46) | 0.02 (0.62) | 0.02 (0.60) |
| | Shape | 0.00 (0.26) | 0.42 (0.85) | 0.78 (0.96) | 0.00 (0.47) | 0.00 (0.57) | 0.04 (0.58) |
| | Material | 0.04 (0.36) | 0.36 (0.83) | 0.56 (0.92) | 0.02 (0.53) | 0.00 (0.60) | 0.02 (0.58) |
| | Scene | 0.06 (0.35) | 0.44 (0.88) | 0.48 (0.90) | 0.00 (0.55) | 0.02 (0.59) | 0.08 (0.64) |
| | Agents | 0.02 (0.55) | 0.16 (0.65) | 0.16 (0.77) | 0.02 (0.50) | 0.02 (0.52) | 0.02 (0.50) |
| Farming | Test | 0.4 (0.72) | 0.62 (0.79) | 0.46 (0.73) | 0.08 (0.39) | 0.04 (0.23) | 0.02 (0.33) |
| | Crop | 0.00 (0.00) | 0.00 (0.00) | 0.00 (0.00) | 0.00 (0.00) | 0.00 (0.00) | 0.00 (0.00) |
| | Scene | 0.30 (0.69) | 0.52 (0.76) | 0.44 (0.75) | 0.04 (0.32) | 0.06 (0.29) | 0.10 (0.33) |
| | Agents | 0.12 (0.54) | 0.44 (0.79) | 0.36 (0.72) | 0.02 (0.22) | 0.00 (0.19) | 0.02 (0.23) |
| Smelting | Test | 0.06 (0.08) | 0.22 (0.44) | 0.32 (0.59) | 0.10 (0.25) | 0.06 (0.09) | 0.10 (0.19) |
| | Goal | 0.00 (0.00) | 0.00 (0.00) | 0.00 (0.00) | 0.00 (0.10) | 0.00 (0.00) | 0.00 (0.00) |
| | Furnace | 0.06 (0.08) | 0.20 (0.40) | 0.18 (0.38) | 0.06 (0.12) | 0.04 (0.08) | 0.04 (0.07) |
| | Scene | 0.04 (0.08) | 0.16 (0.43) | 0.24 (0.56) | 0.12 (0.28) | 0.04 (0.09) | 0.08 (0.18) |
| | Agents | 0.00 (0.03) | 0.00 (0.26) | 0.04 (0.37) | 0.00 (0.02) | 0.00 (0.01) | 0.00 (0.00) |

Table 11: Task success rates and subgoal success rates of the TeamCraft-VLA-13B-Cen and TeamCraft-VLA-13B-Dec models. Subgoal success rates are given in parentheses.

| Tasks | Condition | TeamCraft-VLA-7B | TeamCraft-VLA-13B | GPT-4o | TeamCraft-7B-GridWorld |
|---|---|---|---|---|---|
| Building | Test | 0.42 (0.81) | 0.48 (0.79) | 0.00 (0.07) | 0.42 (0.88) |
| | Shape | 0.30 (0.75) | 0.26 (0.69) | 0.00 (0.08) | 0.50 (0.90) |
| | Material | 0.30 (0.74) | 0.08 (0.63) | 0.00 (0.07) | 0.26 (0.82) |
| | Scene | 0.40 (0.83) | 0.48 (0.83) | 0.00 (0.07) | 0.48 (0.89) |
| | Agents | 0.02 (0.57) | 0.04 (0.59) | 0.00 (0.00) | 0.12 (0.71) |
| Clearing | Test | 0.64 (0.91) | 0.64 (0.94) | 0.00 (0.03) | 1.00 (1.00) |
| | Shape | 0.56 (0.91) | 0.78 (0.96) | 0.00 (0.04) | 1.00 (1.00) |
| | Material | 0.56 (0.91) | 0.56 (0.92) | 0.00 (0.12) | 1.00 (1.00) |
| | Scene | 0.58 (0.92) | 0.48 (0.90) | 0.00 (0.06) | 1.00 (1.00) |
| | Agents | 0.36 (0.81) | 0.16 (0.77) | 0.00 (0.00) | 0.84 (0.97) |
| Farming | Test | 0.36 (0.64) | 0.46 (0.73) | 0.00 (0.00) | 0.78 (0.86) |
| | Crop | 0.00 (0.00) | 0.00 (0.00) | 0.00 (0.00) | 0.00 (0.00) |
| | Scene | 0.38 (0.67) | 0.44 (0.75) | 0.00 (0.00) | 0.90 (0.96) |
| | Agents | 0.38 (0.68) | 0.36 (0.72) | 0.00 (0.00) | 0.40 (0.73) |
| Smelting | Test | 0.24 (0.28) | 0.32 (0.59) | 0.02 (0.02) | 0.24 (0.51) |
| | Goal | 0.00 (0.00) | 0.00 (0.00) | 0.08 (0.08) | 0.00 (0.00) |
| | Furnace | 0.18 (0.20) | 0.18 (0.38) | 0.00 (0.00) | 0.24 (0.39) |
| | Scene | 0.18 (0.23) | 0.24 (0.56) | 0.00 (0.00) | 0.36 (0.58) |
| | Agents | 0.06 (0.13) | 0.04 (0.37) | 0.00 (0.00) | 0.00 (0.31) |

Table 12: Task success rates and subgoal success rates of various centralized models. Subgoal success rates are given in parentheses. All models are trained with the full training data except GPT-4o.

| Tasks | Condition | TeamCraft-VLA-7B | GPT–4o w/o Grid | GPT–4o | o4-mini | Claude-3.7 | Gemini-2.5-Pro |
|---|---|---|---|---|---|---|---|
| Building | Test | 0.42 (0.81) | 0.07 (0.00) | 0.68 (0.16) | 0.80 (0.48) | 0.66 (0.16) | 0.89 (0.58) |
| | Shape | 0.30 (0.75) | 0.08 (0.00) | 0.64 (0.10) | 0.81 (0.48) | 0.54 (0.10) | 0.88 (0.58) |
| | Material | 0.30 (0.74) | 0.07 (0.00) | 0.66 (0.12) | 0.85 (0.54) | 0.59 (0.12) | 0.91 (0.72) |
| | Scene | 0.40 (0.83) | 0.07 (0.00) | 0.60 (0.10) | 0.70 (0.10) | 0.62 (0.00) | 0.91 (0.66) |
| | Agents | 0.02 (0.57) | 0.00 (0.00) | 0.57 (0.08) | 0.80 (0.38) | 0.64 (0.13) | 0.87 (0.66) |
| Clearing | Test | 0.64 (0.91) | 0.04 (0.00) | 0.87 (0.82) | 0.90 (0.84) | 0.94 (0.84) | 0.98 (0.98) |
| | Shape | 0.56 (0.91) | 0.03 (0.00) | 0.85 (0.70) | 0.83 (0.76) | 0.95 (0.92) | 0.98 (0.96) |
| | Material | 0.56 (0.91) | 0.12 (0.00) | 0.88 (0.70) | 0.85 (0.78) | 0.91 (0.88) | 0.94 (0.94) |
| | Scene | 0.58 (0.92) | 0.06 (0.00) | 0.85 (0.64) | 0.86 (0.72) | 0.93 (0.90) | 0.92 (0.92) |
| | Agents | 0.36 (0.81) | 0.00 (0.00) | 0.85 (0.68) | 0.91 (0.85) | 0.93 (0.92) | 0.98 (0.92) |
| Farming | Test | 0.36 (0.64) | 0.00 (0.00) | 0.43 (0.28) | 0.28 (0.24) | 0.33 (0.28) | 0.66 (0.50) |
| | Crop | 0.00 (0.00) | 0.00 (0.00) | 0.76 (0.48) | 0.73 (0.64) | 0.31 (0.24) | 0.86 (0.82) |
| | Scene | 0.38 (0.67) | 0.00 (0.00) | 0.35 (0.16) | 0.31 (0.24) | 0.34 (0.30) | 0.65 (0.52) |
| | Agents | 0.38 (0.68) | 0.00 (0.00) | 0.40 (0.17) | 0.34 (0.24) | 0.31 (0.18) | 0.44 (0.39) |
| Smelting | Test | 0.24 (0.28) | 0.02 (0.02) | 0.43 (0.24) | 0.25 (0.10) | 0.64 (0.38) | 0.91 (0.80) |
| | Goal | 0.00 (0.00) | 0.08 (0.08) | 0.34 (0.20) | 0.35 (0.24) | 0.63 (0.46) | 0.88 (0.70) |
| | Furnace | 0.18 (0.20) | 0.00 (0.00) | 0.31 (0.16) | 0.15 (0.10) | 0.62 (0.42) | 0.90 (0.84) |
| | Scene | 0.18 (0.23) | 0.00 (0.00) | 0.51 (0.24) | 0.31 (0.20) | 0.64 (0.40) | 0.83 (0.68) |
| | Agents | 0.06 (0.13) | 0.00 (0.00) | 0.35 (0.20) | 0.39 (0.21) | 0.69 (0.43) | 0.88 (0.72) |

Table 13: Task success rates and subgoal success rates of various centralized models. Subgoal success rates are given in parentheses. TeamCraft-VLA-7B is trained with the full training data. GPT-4o w/o Grid is tested under Vanilla setting with no grid information provided. GPT-4o, o4-mini, Claude-3.7 and Gemini-2.5-Pro are tested under Grid setting with grid information provided.

**System Prompt**
Two bots need to grow on the platform. The goal is to get 10 beetroot. Write the actions for bot1, bot2 based on this given observation."

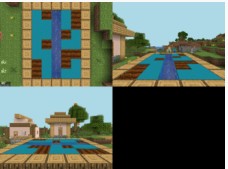

(a) orthographic views
of the task

**Step 0:**

**User Prompt**
bot1 has 7 beetroot_seeds. bot1 has 1 carrot. bot1 has 3 potato. bot2 has 5 beetroot_seeds. bot2 has 2 wheat_seeds. Write the actions for bot1, bot2 based on this given observation.

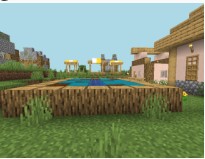 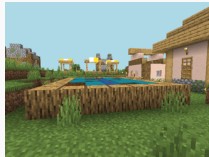

(b) Agent1 observation

(c) Agent2 observation

**Actions**
["farm_work(bot1, new Vec3(-1,-1,2), 'sow', 'beat_seeds')",
"farm_work(bot2, new Vec3(-2,-1,-2), , 'sow', 'beat_seeds')"]

**Step 1:**

**User Prompt**
bot1 has 7 beetroot_seeds. bot1 has 1 carrot. bot1 has 3 potato. bot2 has 5 beetroot_seeds. bot2 has 2 wheat_seeds. Write the actions for bot1, bot2 based on this given observation.

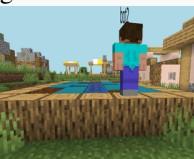 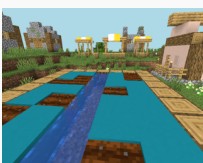

(d) Agent1 observation

(e) Agent2 observation

**Actions**
["farm_work(bot1, new Vec3(3,-1,-1), 'sow', 'beat_seeds')",
"farm_work(bot2, new Vec3(0,-1,-1), 'sow', 'beat_seeds')"]

Figure 8: Object mismatching example 1.

**Step 0:**

**User Prompt**
bot1 has 1 iron_pickaxe. bot1 has 1 iron_axe. bot2 has 2 charcoal. bot2 has 1
iron_shovel. bot2 has 1 iron_sword. Write the actions for bot1, bot2 based on this
given observation.

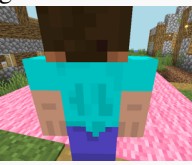 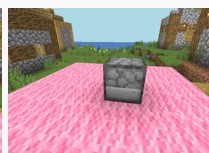

(b) Agent1 observa-  (c) Agent2 observa-
tion                  tion

**Actions**
[putItemFurnace(bot1, 'cobbl1', new Vec3(0,0,-1)),
putFuelFurnace(bot2, 'charcoal', new Vec3(0,0,-1))]

**Step 1:**

**User Prompt**
bot1 has 2 kelp. bot1 has 1 iron_pickaxe. bot1 has 1 iron_axe. bot2 has 1 charcoal.
bot2 has 1 iron_shovel. bot2 has 1 iron_sword. Write the actions for bot1, bot2 based
on this given observation.

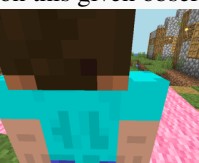 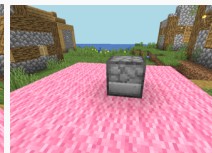

(d) Agent1 observa-  (e) Agent2 observa-
tion                  tion

**Actions**
[putItemFurnace(bot1, 'cobbl1', new Vec3(0,0,-1)),
putFuelFurnace(bot2, 'charcoal', new Vec3(0,0,-1))]

Figure 9: Object mismatching example 2.

**System Prompt**
"Four bots need to break everything on the platform." ",

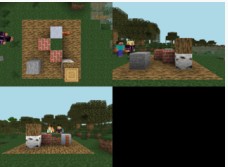

(a) orthographic views
of the task

**Step 0:**

**User Prompt**
bot0 has 1 stone_axe. bot1 has 1 stone_shovel. bot2 has 1 stone_pickaxe. bot3 has 1 stone_sword. Write the actions for bot0, bot1, bot2, bot3 based on this given observation.

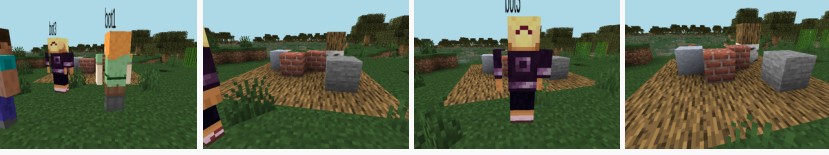

(b) Agent0 observa-(c) Agent1 observa-(d) Agent2 observa-(e) Agent3 observa-
tion                tion                tion                tion

**Actions**
[" mineBlock(bot1, new Vec3(-2,0,1)),
mineBlock(bot2, new Vec3(-1,0,0)),
mineBlock(bot3, new Vec3(1,0,1))", ]

**Step 1:**

**User Prompt**
Write the actions for bot0, bot1, bot2, bot3 based on this given observation."

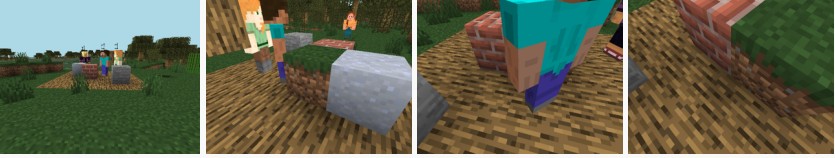

(f) Agent0 observa-(g) Agent1 observa-(h) Agent2 observa-(i) Agent3 observa-
tion                tion                tion                tion

**Actions**
[" mineBlock(bot1, new Vec3(1,0,0)),
mineBlock(bot2, new Vec3(-2,0,-2)),
mineBlock(bot3, new Vec3(1,0,1))", ]

Figure 10: Task allocation failure example 3.

**System Prompt**
"Four bots need to build a building on the platform. "

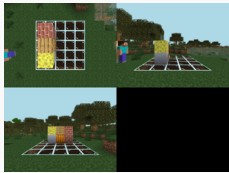

(a) orthographic views
of the task

**Step 0:**

**User Prompt**
bot0 has 4 bricks. bot0 has 5 sponge. bot0 has 2 oak_planks...bot1 has 1 pumpkin...
bot2 has 3 bricks... bot3 has 1 clay... Write the actions for bot0, bot1, bot2, bot3
based on this given observation.

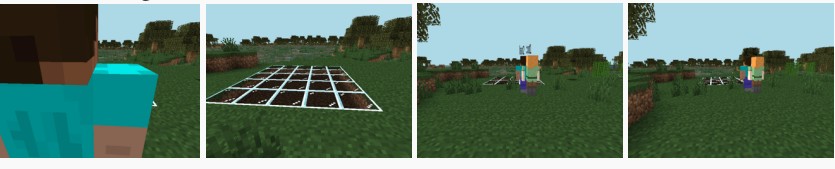

(b) Agent0 observa-(c) Agent1 observa-(d) Agent2 observa-(e) Agent3 observa-
tion                 tion                 tion                 tion

**Actions**
[ " placeItem(bot1, 'bricks', new Vec3(0,0,-1)),
placeItem(bot2, 'clay', new Vec3(-1,0,-1)),
placeItem(bot3, 'clay', new Vec3(1,0,-1))" ]

**Step 1:**

**User Prompt**
Write the actions for bot0, bot1, bot2, bot3 based on this given observation.

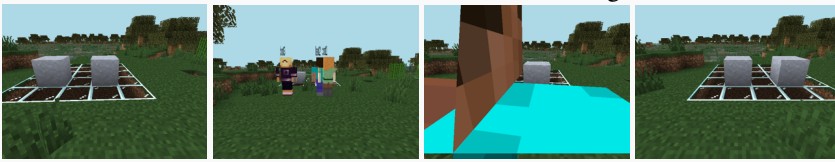

(f) Agent0 observa-(g) Agent1 observa-(h) Agent2 observa-(i) Agent3 observa-
tion                 tion                 tion                 tion

**Actions**
["placeItem(bot1, 'sponge', new Vec3(-1,1,-1)),
placeItem(bot2, 'bricks', new Vec3(1,1,-1)),
placeItem(bot3, 'sponge', new Vec3(0,1,-1))", ]

Figure 11: Task allocation failure example 4.

**System Prompt**
Two bots need to grow on the platform. The goal is to get 4 carrot.

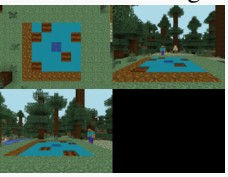

(a) orthographic views
of the task

**Step 0:**

**User Prompt**
"bot1 has 4 carrot. bot1 has 1 potato. bot1 has 1 beetroot. bot2 has 5 carrot. bot2 has 2 beetroot. bot2 has 3 wheat_seeds. bot2 has 2 potato. Write the actions for bot1, bot2 based on this given observation."

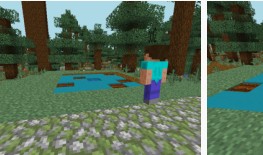

(b) Agent1 observa-(c) Agent2 observa-
tion                tion

**Actions**
[" farm_work(bot1, new Vec3(1,-1,-2), 'sow', 'carrot'),
farm_work(bot2, new Vec3(2,-1,-2), 'sow', 'carrot')", ]

**Step 2:**

**User Prompt**
"bot1 has 3 carrot. bot1 has 1 potato. bot1 has 1 beetroot. bot2 has 5 carrot. bot2 has 2 beetroot. bot2 has 3 wheat_seeds. bot2 has 2 potato. Write the actions for bot1, bot2 based on this given observation."

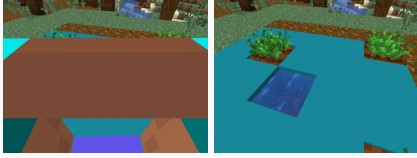

(d) Agent1 observa-(e) Agent2 observation
tion                tion

**Actions**
[" farm_work(bot1, new Vec3(-2,-1,-2), 'sow', 'carrot'),
farm_work(bot2, new Vec3(-1,-1,1), 'sow', 'carrot')", ]

Figure 12: Object state recognition failure example 5.

**System Prompt** Three bots need to build a building on the platform. Target building is: Put sea_lantern on [0 ,1 ,0]. Put oak_fence on [-1 ,1 ,0]. Put sponge on [0 ,1 ,-1]. Put emerald_block on [-1 ,1 ,-1]. Put dirt on [0 ,0 ,0]. Put bricks on [-1 ,0 ,0]. Put emerald_block on [0 ,0 ,-1]. Put clay on [-1 ,0 ,-1]. .

**User Prompt**
bot1 has 4 dirt. bot1 has 3 clay. bot1 has 7 emerald_block. bot1 has 1 oak_fence. bot1 has 3 sponge. bot1 has 1 bricks. bot1 has 3 sea_lantern. bot2 has 4 bricks. bot2 has 2 sponge. bot2 has 6 sea_lantern. bot2 has 2 oak_fence. bot2 has 4 emerald_block. bot2 has 1 dirt. bot2 has 3 clay. bot3 has 6 emerald_block. bot3 has 4 oak_fence. bot3 has 2 dirt. bot3 has 2 sponge. bot3 has 3 clay. bot3 has 2 sea_lantern. bricks is on [-1 ,0 ,0]. dirt is on [0 ,0 ,0]. Write the actions for bot1, bot2 and bot3 based on this given observation.

Figure 13: Prompt example for Building task under the grid-world setting.

**System Prompt** Three bots need to break everything on the platform. clay is on [-2 ,0 ,-2]. birch_log is on [-2 ,0 ,0]. dirt is on [-1 ,0 ,-2]. crafting_table is on [-1 ,0 ,1]. anvil is on [-1 ,1 ,1]. anvil is on [0 ,0 ,-2]. iron_ore is on [0 ,0 ,1]. cobweb is on [1 ,0 ,1].

**User Prompt** bot1 has 1 stone_pickaxe. bot1 has 1 anvil. bot2 has 1 stone_axe. bot2 has 1 crafting_table. bot3 has 1 stone_pickaxe. bot3 has 1 dirt. clay is on [-2 ,0 ,-2]. birch_log is on [-2 ,0 ,0]. iron_ore is on [0 ,0 ,1]. cobweb is on [1 ,0 ,1]. Write the actions for bot1, bot2 and bot3 based on this given observation.

Figure 14: Prompt example for Clearing task under the grid-world setting.

**System Prompt** Two bots need to grow on the platform. The goal is to get 5 carrot. farmland is on [-3 ,-1 ,-2] with value of 7. cyan_concrete is on [-3 ,-1 ,-1]. water is on [-3 ,-1 ,0]. cyan_concrete is on [-3 ,-1 ,1]. cyan_concrete is on [-3 ,-1 ,2]. farmland is on [-2 ,-1 ,-2] with value of 7. cyan_concrete is on [-2 ,-1 ,-1]. water is on [-2 ,-1 ,0]. farmland is on [-2 ,-1 ,1] with value of 7. cyan_concrete is on [-2 ,-1 ,2]. cyan_concrete is on [-1 ,-1 ,-2]. cyan_concrete is on [-1 ,-1 ,-1]. water is on [-1 ,-1 ,0]. farmland is on [-1 ,-1 ,1] with value of 7. farmland is on [-1 ,-1 ,2] with value of 7. cyan_concrete is on [0 ,-1 ,-2]. farmland is on [0 ,-1 ,-1] with value of 7. water is on [0 ,-1 ,0]. cyan_concrete is on [0 ,-1 ,1]. cyan_concrete is on [0 ,-1 ,2]. cyan_concrete is on [1 ,-1 ,-2]. cyan_concrete is on [1 ,-1 ,-1]. water is on [1 ,-1 ,0]. farmland is on [1 ,-1 ,1] with value of 7. cyan_concrete is on [1 ,-1 ,2]. cyan_concrete is on [2 ,-1 ,-2]. cyan_concrete is on [2 ,-1 ,-1]. water is on [2 ,-1 ,0]. cyan_concrete is on [2 ,-1 ,1]. farmland is on [2 ,-1 ,2] with value of 7. cyan_concrete is on [3 ,-1 ,-2]. farmland is on [3 ,-1 ,-1] with value of 7. water is on [3 ,-1 ,0]. cyan_concrete is on [3 ,-1 ,1]. farmland is on [3 ,-1 ,2] with value of 7.

**User Prompt**
bot1 has 5 carrot. bot1 has 2 beetroot. bot1 has 3 potato. bot2 has 2 carrot. bot2 has 2 beetroot. bot2 has 2 wheat_seeds. farmland is on [-3 ,-1 ,-2] with value of 7. cyan_concrete is on [-3 ,-1 ,-1]. water is on [-3 ,-1 ,0]. cyan_concrete is on [-3 ,-1 ,1]. cyan_concrete is on [-3 ,-1 ,2]. farmland is on [-2 ,-1 ,-2] with value of 7. cyan_concrete is on [-2 ,-1 ,-1]. water is on [-2 ,-1 ,0]. farmland is on [-2 ,-1 ,1] with value of 7. cyan_concrete is on [-2 ,-1 ,2]. cyan_concrete is on [-1 ,-1 ,-2]. cyan_concrete is on [-1 ,-1 ,-1]. water is on [-1 ,-1 ,0]. farmland is on [-1 ,-1 ,1] with value of 7. farmland is on [-1 ,-1 ,2] with value of 7. cyan_concrete is on [0 ,-1 ,-2]. farmland is on [0 ,-1 ,-1] with value of 7. water is on [0 ,-1 ,0]. cyan_concrete is on [0 ,-1 ,1]. cyan_concrete is on [0 ,-1 ,2]. cyan_concrete is on [1 ,-1 ,-2]. cyan_concrete is on [1 ,-1 ,-1]. water is on [1 ,-1 ,0]. farmland is on [1 ,-1 ,1] with value of 7. cyan_concrete is on [1 ,-1 ,2]. cyan_concrete is on [2 ,-1 ,-2]. cyan_concrete is on [2 ,-1 ,-1]. water is on [2 ,-1 ,0]. cyan_concrete is on [2 ,-1 ,1]. farmland is on [2 ,-1 ,2] with value of 7. cyan_concrete is on [3 ,-1 ,-2]. farmland is on [3 ,-1 ,-1] with value of 7. water is on [3 ,-1 ,0]. cyan_concrete is on [3 ,-1 ,1]. farmland is on [3 ,-1 ,2] with value of 7. carrots is on [3 ,0 ,-1] with value of 0. carrots is on [3 ,0 ,2] with value of 0. Write the actions for bot1, bot2 based on this given observation.

Figure 15: Prompt example for Farming task under the grid-world setting.

**System Prompt** Two bots need to craft 2 stone. here are the instructions: Cooking Food: 1. To cook a 'cooked_beef'... cobblestone is on [-2 ,0 ,2]. furnace is on [0 ,0 ,1]. spruce_planks is on [2 ,0 ,-3]. cobblestone is on [2 ,0 ,-1].

**User Prompt**
bot1 has 1 iron_sword. bot1 has 1 iron_shovel. bot1 has 1 iron_pickaxe. bot1 has 1 cobblestone. bot1 has 1 spruce_planks. bot2 has 1 spruce_planks. bot2 has 1 iron_shovel. bot2 has 2 iron_pickaxe. cobblestone is on [-2 ,0 ,2]. furnace is on [0 ,0 ,1]. spruce_planks is on [2 ,0 ,-3]. cobblestone is on [2 ,0 ,-1]. Write the actions for bot1, bot2 based on this given observation.

Figure 16: Prompt example for Smelting task under the grid-world setting.

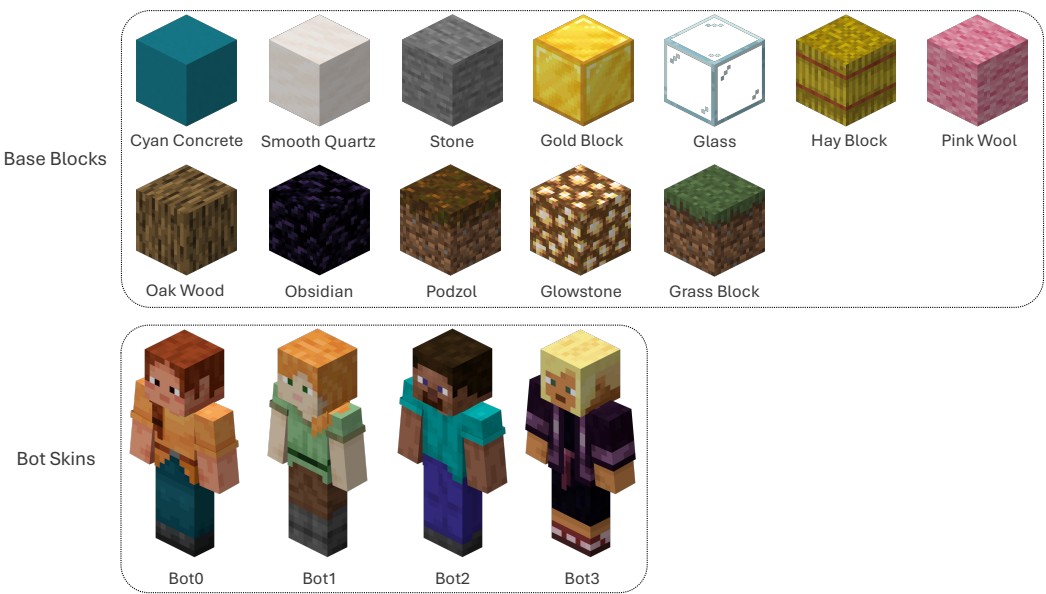

Figure 17: A close-up view of the shared visual diversity in every tasks.

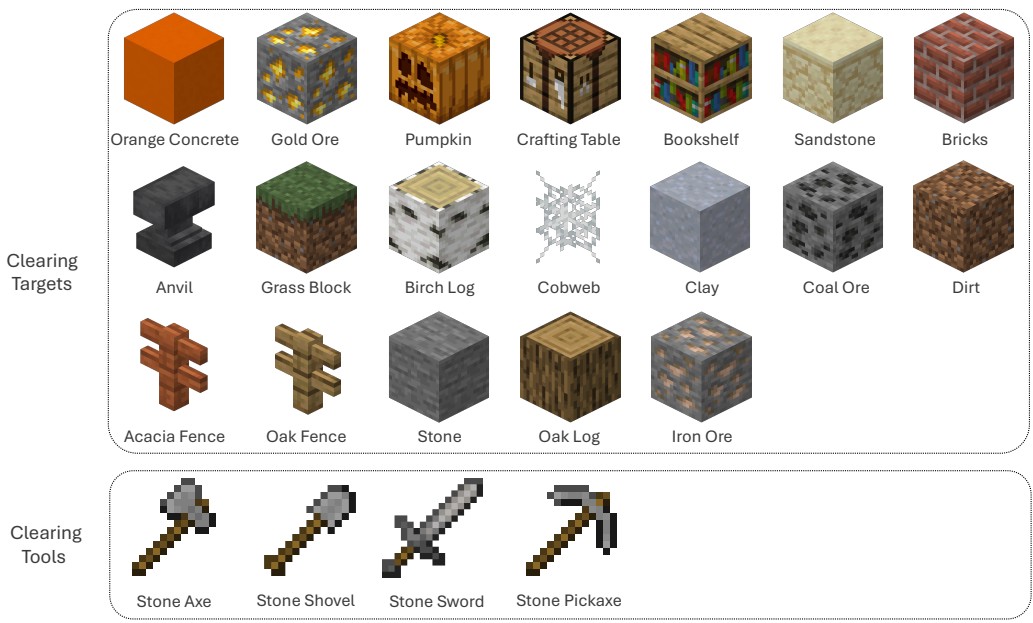

Figure 18: A close-up view of the visual diversity in clearing tasks.

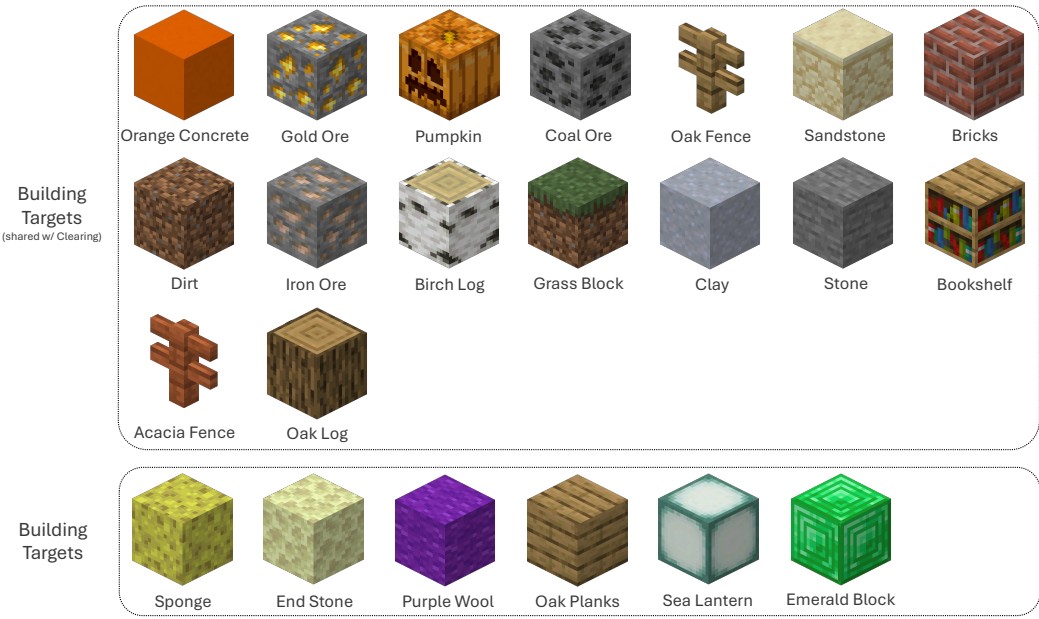

Figure 19: A close-up view of the visual diversity in building tasks.

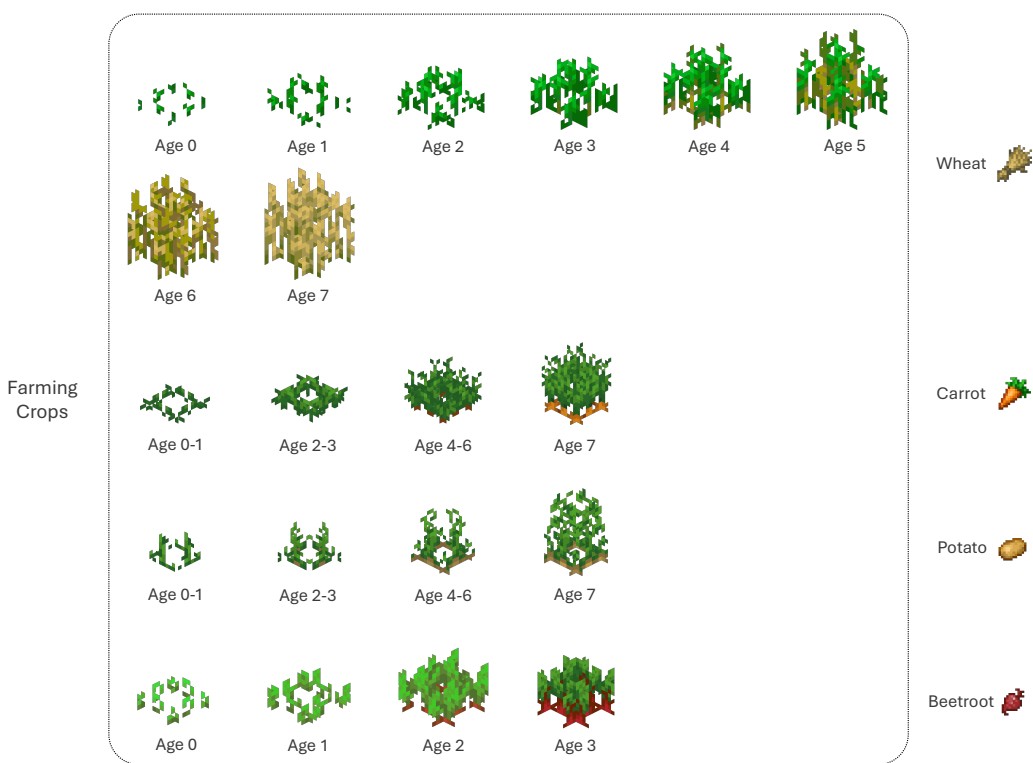

Figure 20: A close-up view of crops appearances across various growing stages in farming tasks.

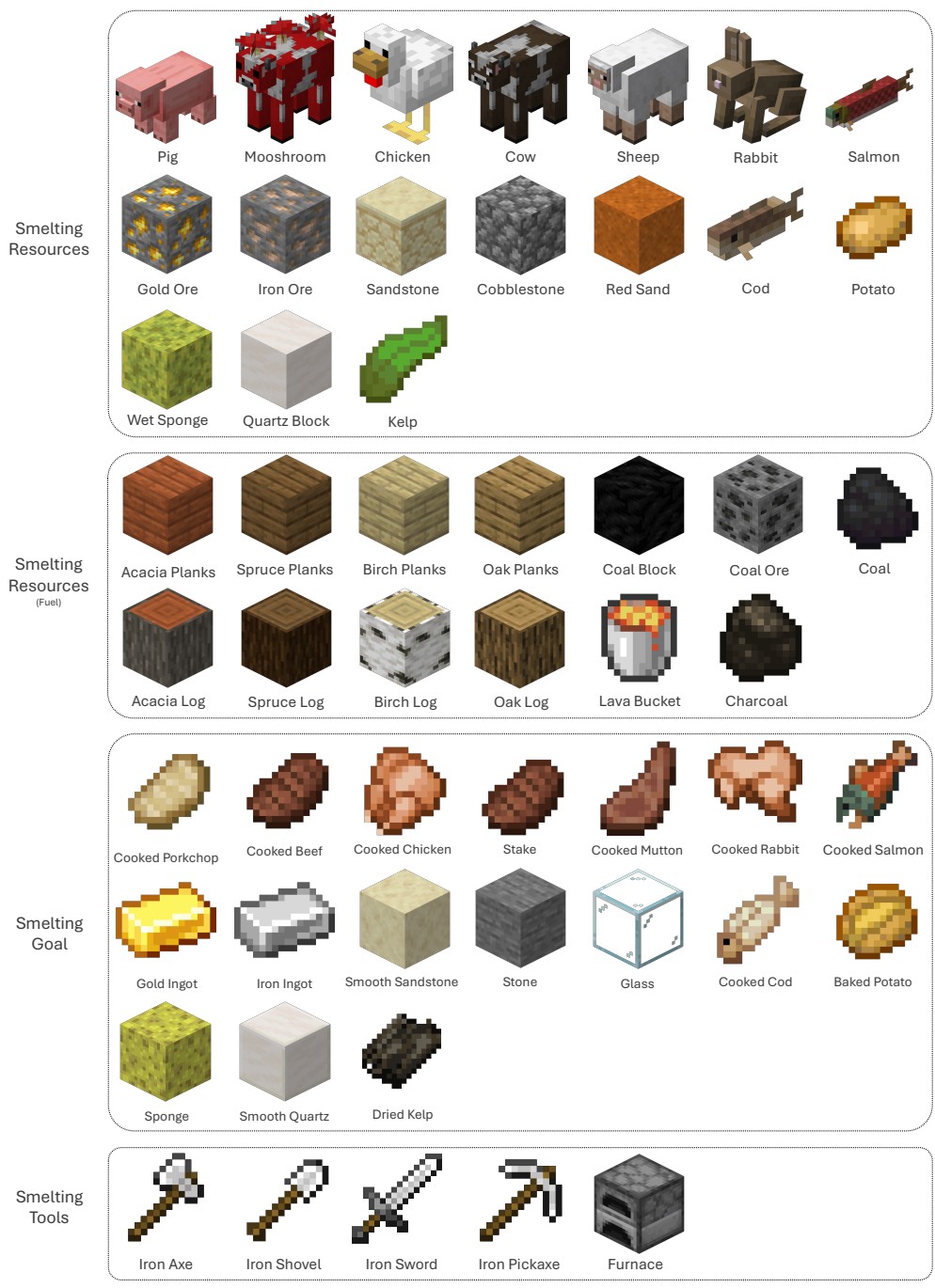

Figure 21: A close-up view of the visual diversity in smelting tasks.

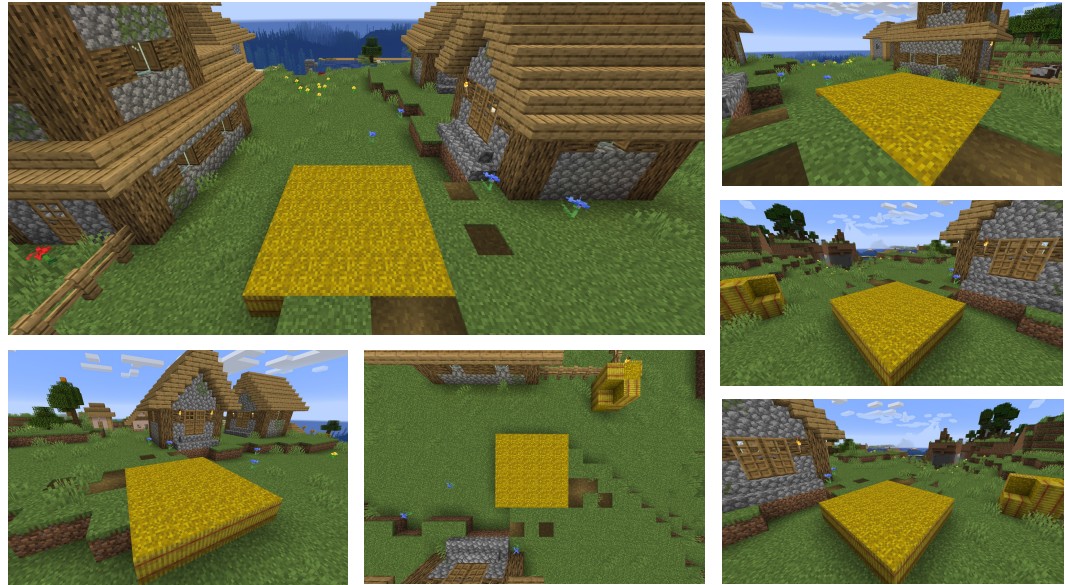

Figure 22: One example scene in the seaside village biome.

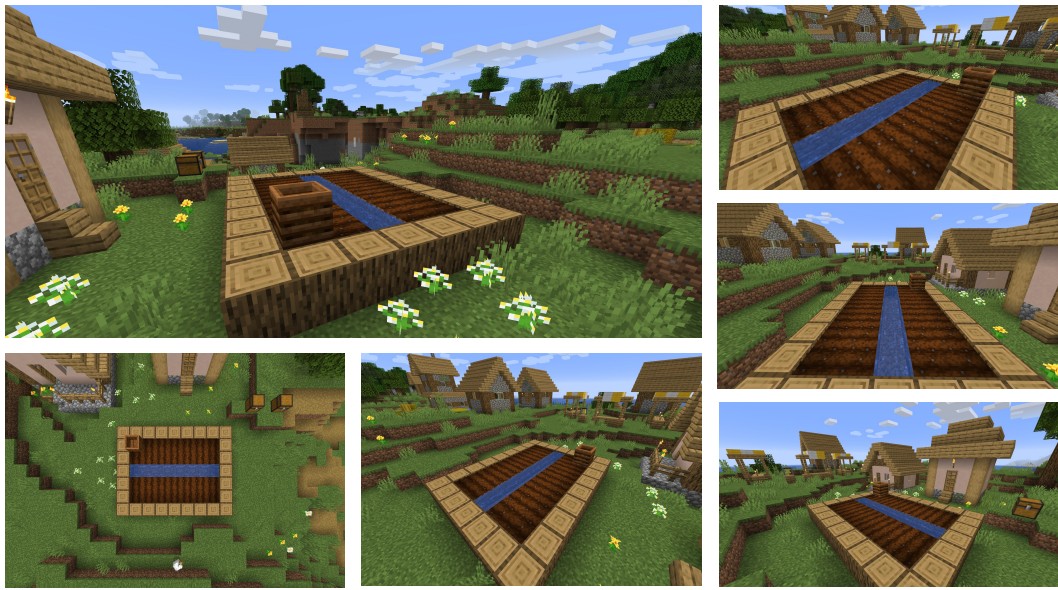

Figure 23: One example scene in the grass village biome.

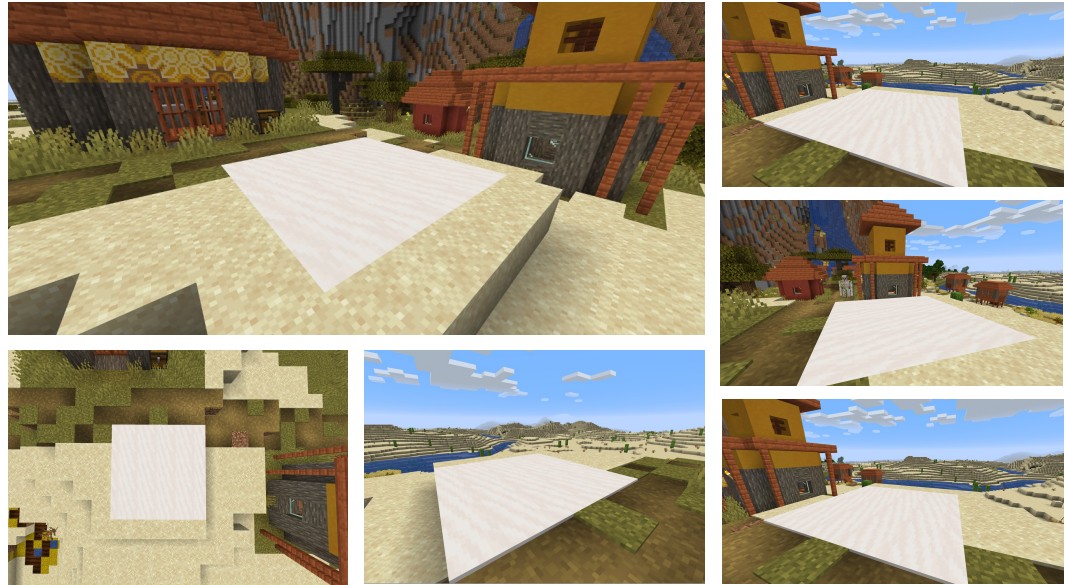

Figure 24: One example scene in the dessert village biome.

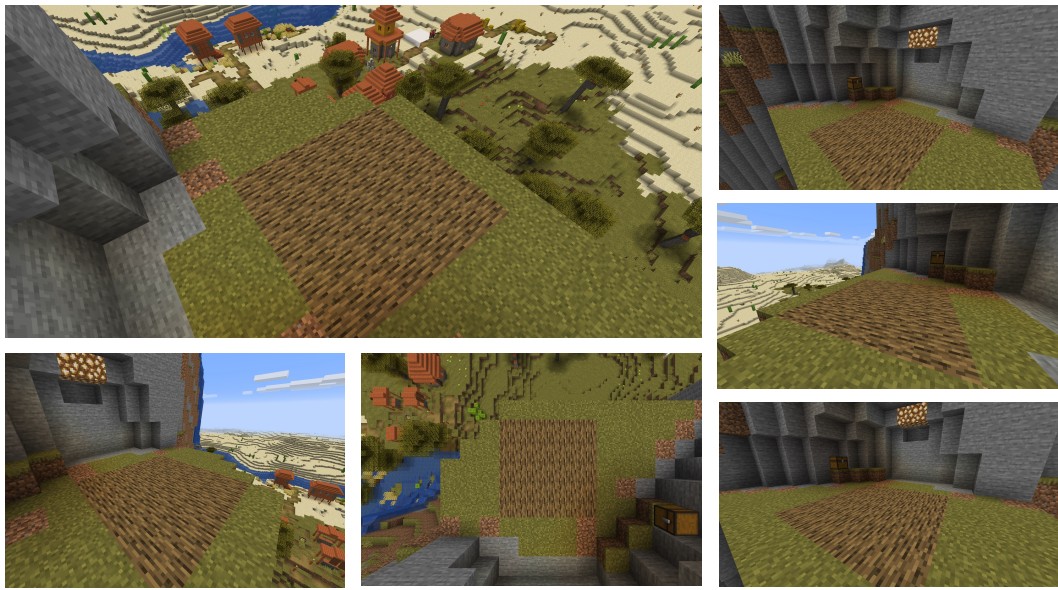

Figure 25: One example scene in the half mountain biome.

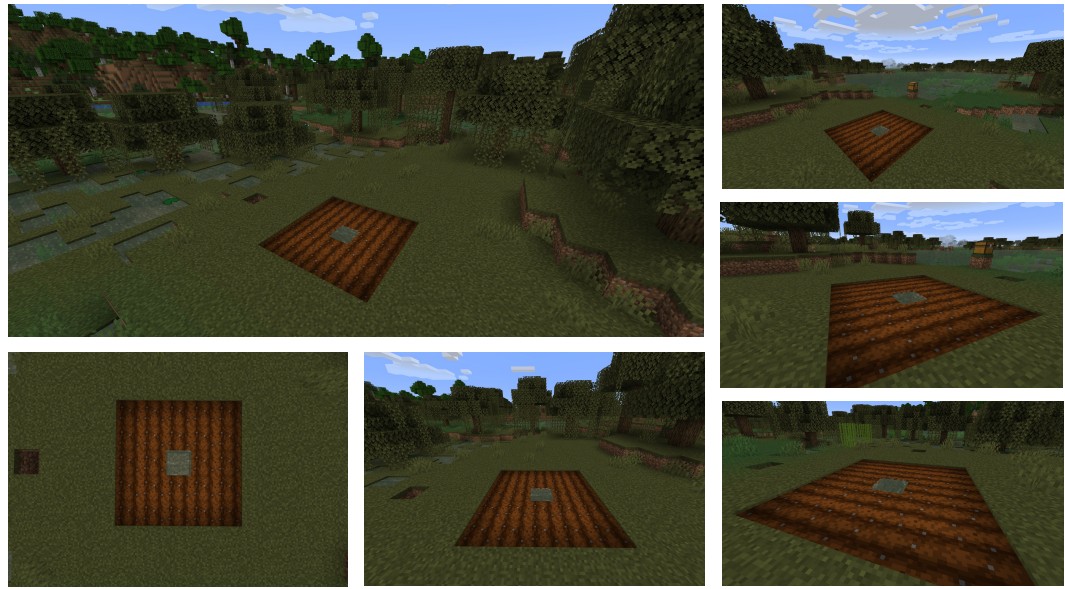

Figure 26: One example scene in the swamp biome.

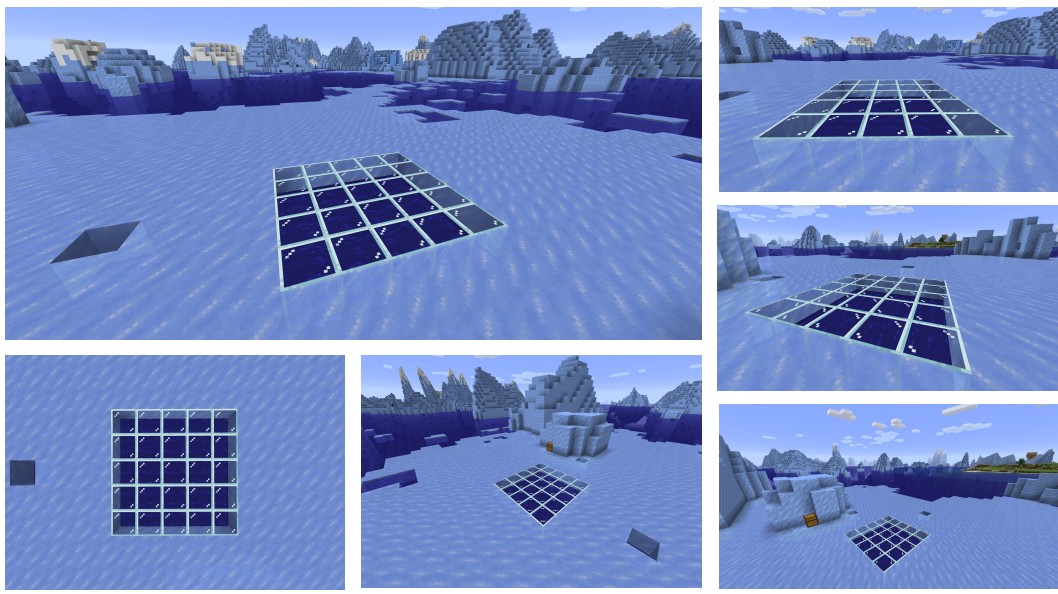

Figure 27: One example scene in the iceberg biome.

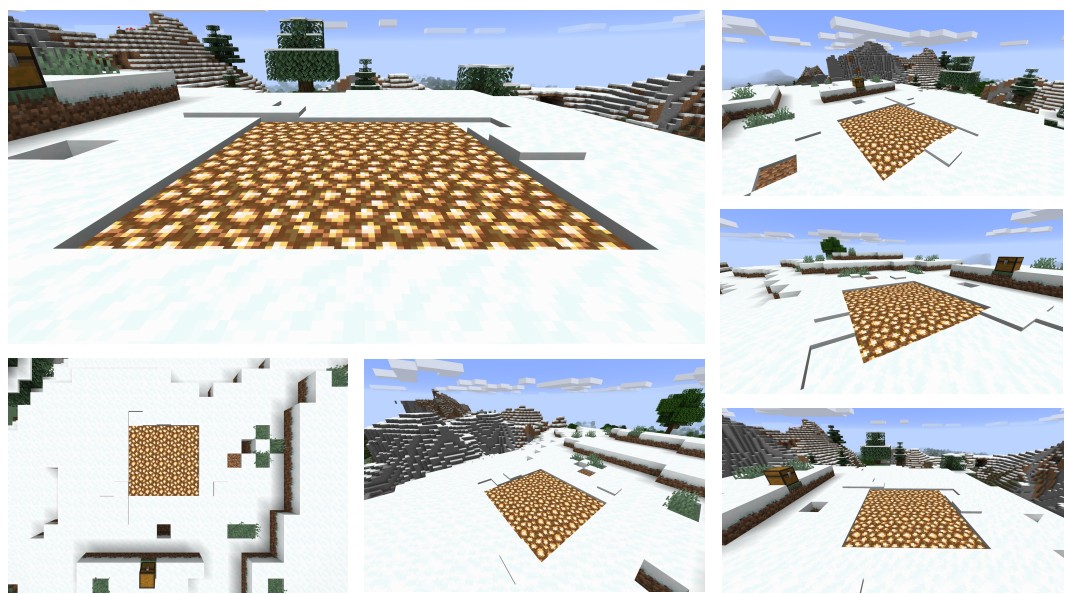

Figure 28: One example scene in the snow mountain biome.

Building 01:

build a building on the playground base halfway up the mountain, following the blueprint given below. Bot1 has 5 Pumpkin, 2 Sea Lantern, 1 Coa Ore. Bot2 has 3 Coa Ore, 2 Orange Concrete, 2 Oak Planks. Bot3 has 3 Bricks, 4 Oak Planks. The blueprint looks like:

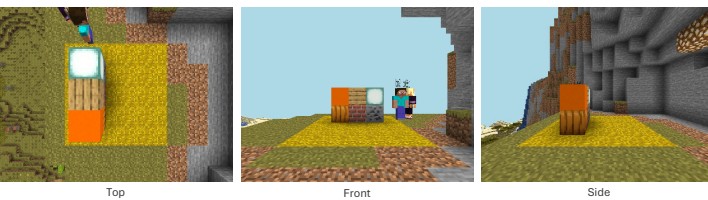

Step 0:

**Actions:** None

**Reward:** 0

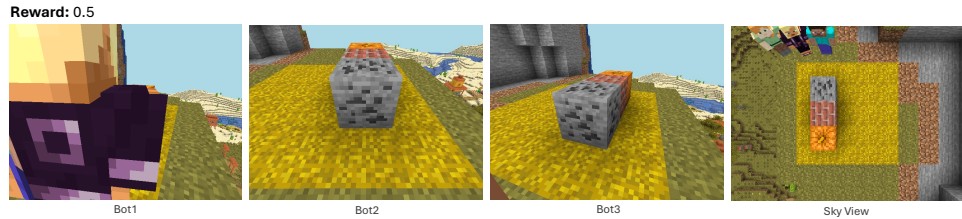

Step 1:

**Actions:** placeItem(bot1, 'pumpkin', new Vec3(-1,0,-1)), placeItem(bot2, 'coal_ore', new Vec3(1,0,-1)), placeItem(bot3, 'bricks', new Vec3(0,0,-1))

**Reward:** 0.5

Figure 29: An example demonstration in the building task, part I.

Step 2:

**Actions:** placeItem(bot1, 'sea_lantern', new Vec3(1,1,-1)), placeItem(bot2, 'orange_concrete', new Vec3(-1,1,-1)), placeItem(bot3, 'oak_planks', new Vec3(0,1,-1))

**Reward:** 1.0

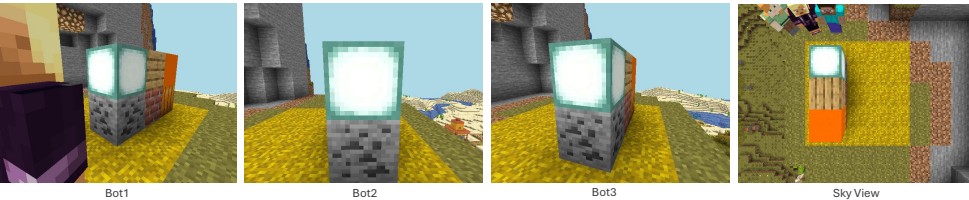

Figure 30: An example demonstration in the building task, part II.

Clearing 01:

 remove  blocks on the playground base  in a dissert village. 

Bot1  Bot2          Crafting Table  Bookshelf  Sandstone  Bricks                              Gold Block                          Bot1

has 1 ,  has 1 .  requires only one timestep to break , but two timesteps

Stone Pickaxe  Bot2  Stone Pickaxe  Stone Pickaxe                              Sandstone  Bricks

to break  or . The initial playground looks like:

Bookshelf  Crafting Table

Top                          Front                          Side

Step 0:

**Actions:** None

**Reward:** 0

Bot1                          Bot2                          Sky View

Figure 31: An example demonstration in the clearing task, part I.

Step 1:

**Actions:** mineBlock(bot1, new Vec3(1,0,0)), mineBlock(bot2, new Vec3(0,0,-1))

**Reward:** 0.5

Bot1                          Bot2                          Sky View

Step 2:

**Actions:** mineBlock(bot1, new Vec3(-1,0,1)), mineBlock(bot2, new Vec3(-2,0,1))

**Reward:** 0.5

Same visual observation as it requires two timesteps to break.

Step 3:

**Actions:** mineBlock(bot1, new Vec3(-1,0,1)), mineBlock(bot2, new Vec3(-2,0,1))

**Reward:** 1.0

Bot1                          Bot2                          Sky View

Figure 32: An example demonstration in the clearing task, part II.

Farming 01:

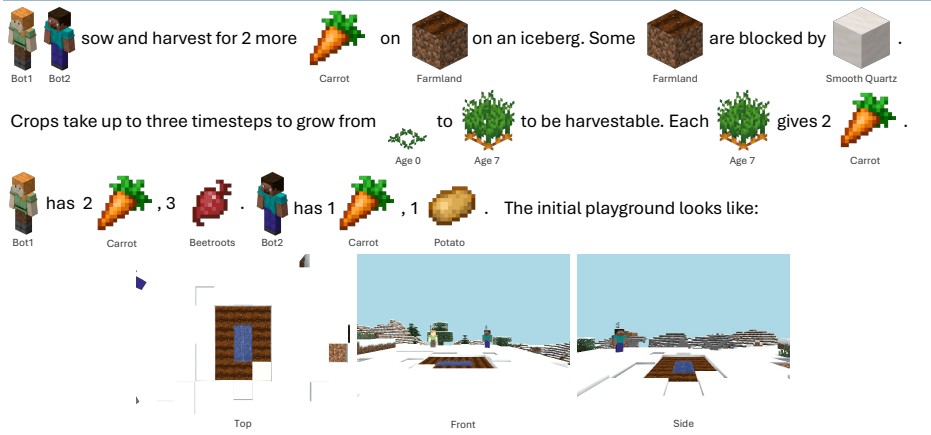

Figure 33: An example demonstration in the farming task, part I.

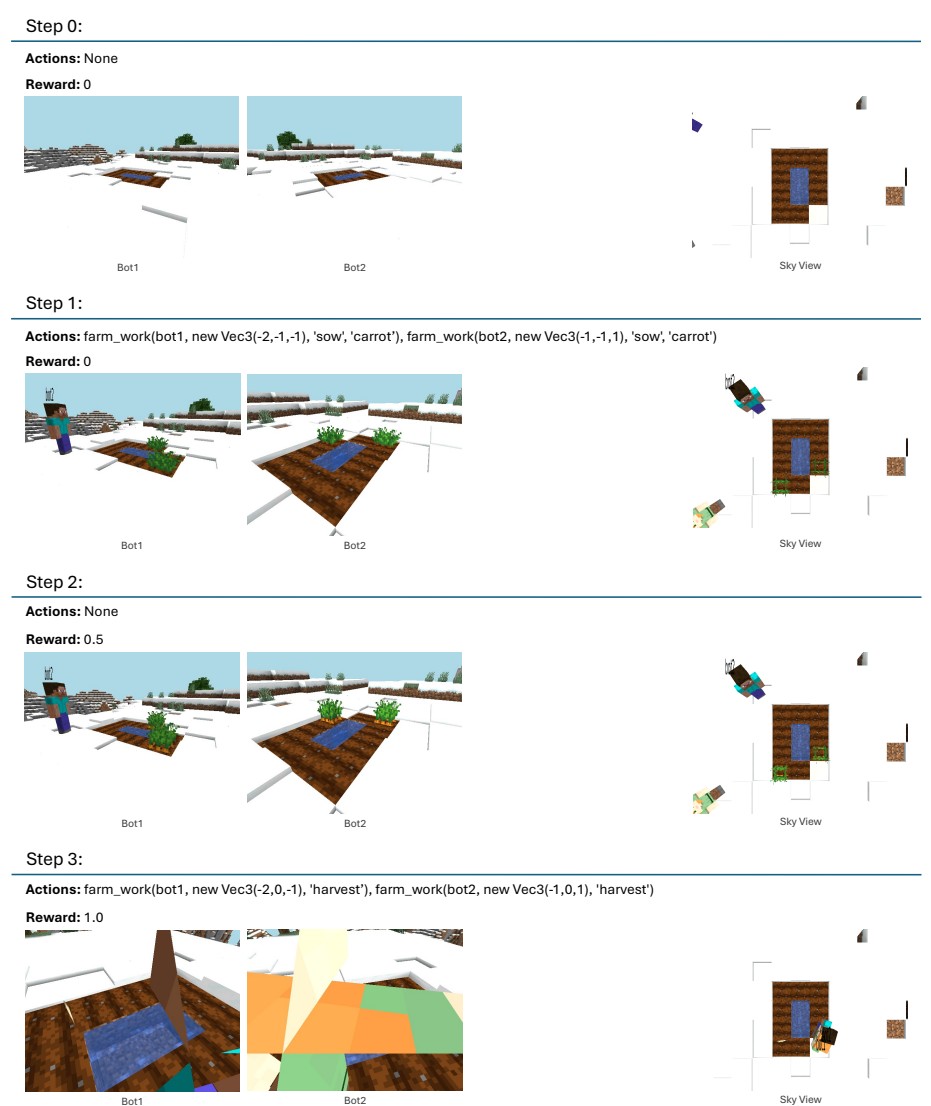

Figure 34: An example demonstration in the farming task, part II.

Smelting 01:

[Bot1] [Bot2] cook for 2 [Cooked Porkchop] in a dissert village. To get [Cooked Porkchop], smelt [Porkchop] with [Birch Log] as fuel in [Furnace].

Resources are on base [Hay Block] or in inventory. To get [Porkchop], kill [Pig] with [Iron Sword]. To get [Birch Log], collect with

[Iron Axe]. [Bot1] has 1 [Iron Sword]. [Bot2] has 1 [Iron Axe], 1 [Porkchop], and [Birch Log]. The initial playground looks like:

Top      Front      Side

Step 0:

**Actions:** None

**Reward:** 0

Bot1      Bot2      Sky View

Step 1:

**Actions:** killMob(bot1, new Vec3(2,0,-3)), putFuelFurnace(bot2, 'birch_log', new Vec3(0,0,0))

**Reward:** 0

Bot1      Bot2      Sky View

Figure 35: An example demonstration in the smelting task, part I.

**Step 2:**

**Actions:** putItemFurnace(bot1, 'porkchop', new Vec3(0,0,0)), putItemFurnace(bot2, 'porkchop', new Vec3(0,0,0))

**Reward:** 0.5

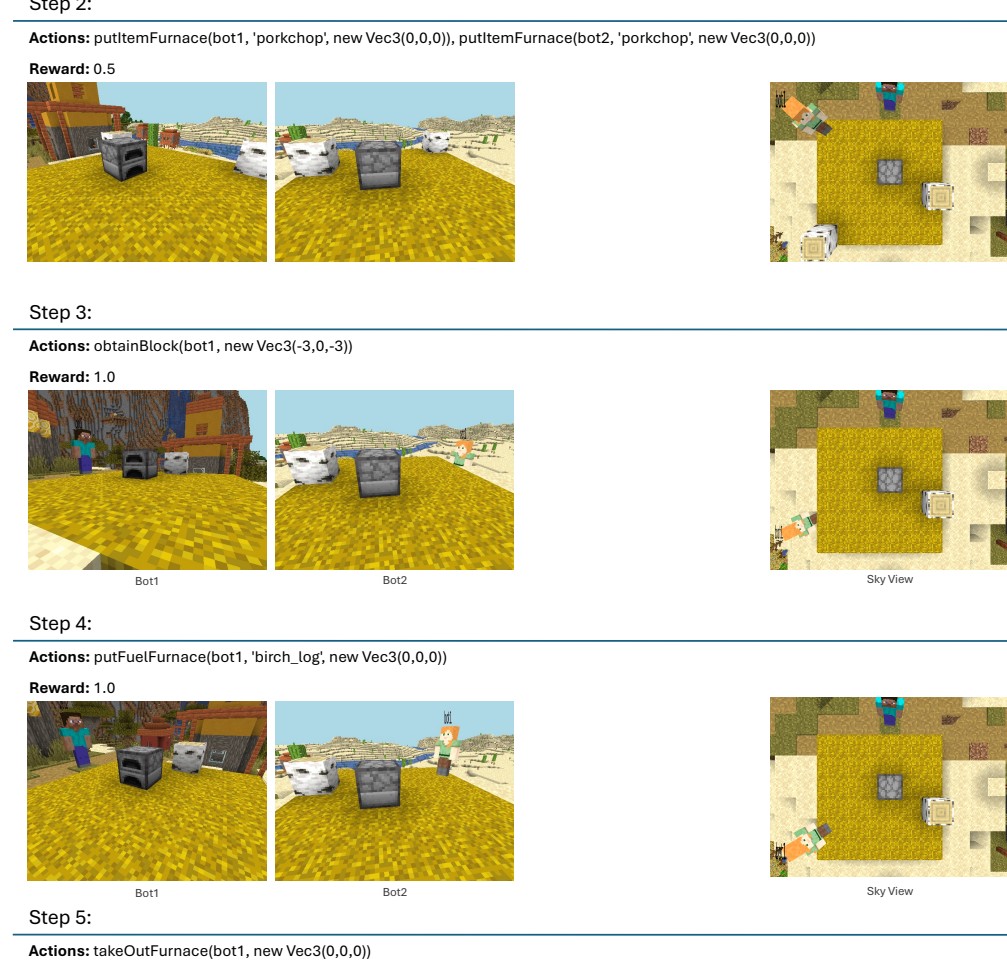

**Step 3:**

**Actions:** obtainBlock(bot1, new Vec3(-3,0,-3))

**Reward:** 1.0

Bot1                    Bot2                    Sky View

**Step 4:**

**Actions:** putFuelFurnace(bot1, 'birch_log', new Vec3(0,0,0))

**Reward:** 1.0

Bot1                    Bot2                    Sky View

**Step 5:**

**Actions:** takeOutFurnace(bot1, new Vec3(0,0,0))

**Reward:** 1.0

Same visual observation as step 4.

Figure 36: An example demonstration in the smelting task, part II.

**System Prompt**
You are controlling up to 4 bots in a Minecraft world. Your mission is to build a specific structure on a platform by coordinating the bots' actions across multiple timesteps with observation feedback.
**Observations:** - **Images:** - Up to 4 first-person perspective images (one from each bot), all agent's view will reset to look at the middle of the platform for each timestep - The goal structure (last image): Contains three orthographic views of the target structure. * Top-left: Top view (looking down from above, shows x-z plane, x increasing to the right, z increasing to the top) Note if the structure has two layers, the top view will show the top layer. * Top-right: Front view (looking from front, shows x-y plane, x increasing to the right, y increasing to the top) * Bottom-left: Right side view (looking from right side, shows z-y plane, z increasing to the right, y increasing to the top) * Bottom-right: Pure black image (no information) - **Current State:** - A text description of inventory information for each bot
**Action Format:** - Use the placeItem() function with these parameters: - botID: 'bot1', 'bot2', 'bot3' (for agent number 1, 2, 3, depending on number of bots available); 'bot0', 'bot1', 'bot2', 'bot3' (if 4 bots are available) - block: The type of block to place (must be in bot's inventory) - position: new Vec3(x,y,z) where (based on the top view image): * x: vertical axis (-2 to 2) * y: height axis (0 to 1) * z: horizontal axis (-2 to 2) - No space in coordinates x,y,z **Available Blocks:**

```
'oak_fence', 'birch_log', 'coal_ore', 'bricks', 'sandstone', 'stone',
'iron_ore', 'gold_ore', 'sponge', 'sea_lantern', 'dirt', 'grass_block',
'clay', 'oak_planks', 'emerald_block', 'pumpkin', 'orange_concrete',
'purple_wool', 'end_stone', 'bookshelf', 'acacia_fence', 'oak_log'
```

**Constraints:** - Each bot can only place blocks from its own inventory - Inventory may contain unneeded blocks - No overlapping blocks at the same position - No floating blocks (must build on existing structures) - Each bot can place at most one block per timestep
**Workspace Coordinates:** - Center: (0,0,0) - Platform size: 5x5 units (x and z axes) - Platform corners (**based on the top view image**): * Top-left: (2,0,-2) * Top-right: (2,0,2) * Bottom-right: (-2,0,2) * Bottom-left: (-2,0,-2) - Height: * 0: Ground level (one block above the workspace platform, lowest height to place blocks) * 1: Above ground level
**Orientation Strategy:** - Bot positions and orientations vary at each timestep, meaning bot's left might not be the same as the previous timestep, or the same as the other bots, or the same as the goal image - Recommended approach: First place blocks at known coordinates (like center 0,0,0 and one-block-to-the-left 0,0,-1) to establish reference points in bot's view - Use placed blocks and its positions in your action output to confirm your understanding of the coordinate system
**Response Format:** - Provide only action commands, each on a new line using new line as separator - Do not include explanations, markdown, or code blocks - A bot can be idle (no action) for a timestep, simply not output such bot's action - Use "NULL" if all bots are idle
**Example Response:**

```
placeItem(bot1, 'pumpkin', new Vec3(-1,0,-1))
placeItem(bot2, 'purple_wool', new Vec3(1,0,-1))
placeItem(bot3, 'coal_ore', new Vec3(0,0,-1))
```

Figure 37: Sample rollouts of GPT-4o in a building task, part I.

**User Prompt**

Here are the observations at the current time step for each agent. The final image is the goal image. bot1 has 5 dirt. bot1 has 3 clay. bot1 has 7 emerald block. bot1 has 1 oak fence. bot1 has 3 sponge. bot1 has 1 bricks. bot1 has 3 sea lantern. bot2 has 5 bricks. bot2 has 2 sponge. bot2 has 6 sea lantern. bot2 has 2 oak fence. bot2 has 4 emerald block. bot2 has 1 dirt. bot2 has 3 clay. bot3 has 6 emerald block. bot3 has 4 oak fence. bot3 has 2 dirt. bot3 has 2 sponge. bot3 has 3 clay. bot3 has 2 sea lantern.

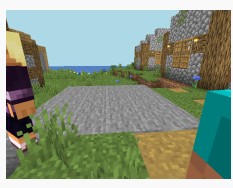 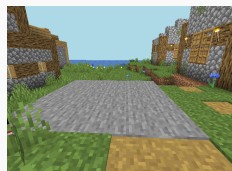 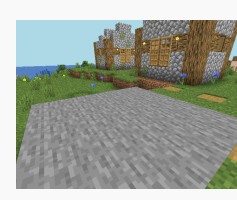 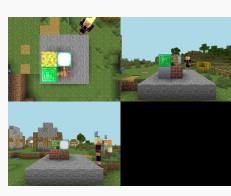

(a) Agent1 observation    (b) Agent2 observation    (c) Agent3 observation    (d) orthographic views of the goal

**Assistant**

[placeItem(bot1, emerald block, new Vec3(-1,0,-1)), placeItem(bot2, bricks, new Vec3(0,0,0)),
placeItem(bot3, sea lantern, new Vec3(-1,0,0))]

Figure 38: Sample rollouts of GPT-4o in a building task, part II.

**User Prompt**

Here are the observations at the current time step for each agent. The final image is the goal image. bot1 has ...

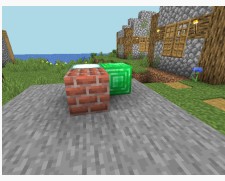

(a) Agent1 observation

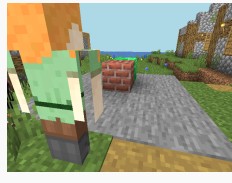

(b) Agent2 observation

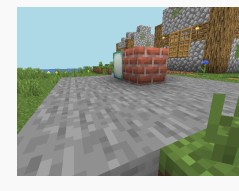

(c) Agent3 observation

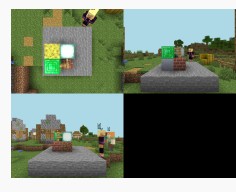

(d) orthographic views of the goal

**Assistant**

[placeItem(bot1, sponge, new Vec3(1,0,-1)), placeItem(bot2, oak fence, new Vec3(0,0,1))]

**User Prompt**

Here are the observations at the current time step for each agent. The final image is the goal image. bot1 has ...

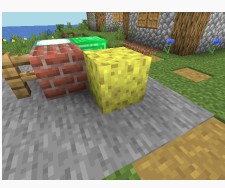

(e) Agent1 observation

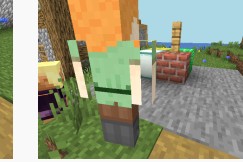

(f) Agent2 observation

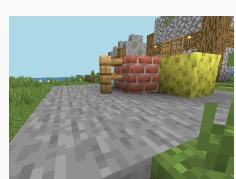

(g) Agent3 observation

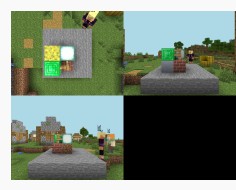

(h) orthographic views of the goal

**Assistant**

[placeItem(bot3, sea lantern , new Vec3(0,0,-1))]

**Comment**

GPT-4o failed to place the blocks in the correct location. The agents placed an additional fence that is not required by the goal. The sponge (yellow) block should be placed next to a sea lattern (white) block.

Figure 39: Sample rollouts of GPT-4o in a building task, part III.

**System Prompt**
You are controlling up to 4 bots in a Minecraft world. Your mission is to clear all the blocks on the platform, by coordinating the bots' actions across multiple timesteps with observation feedback.
**Observations:** - **Images:** - Up to 4 first-person perspective images (one from each bot), all agent's view will reset to look at the middle of the platform for each timestep - The goal structure (last image): Contains three orthographic views of the target structure. * Top-left: Top view (looking down from above, shows x-z plane, x increasing to the right, z increasing to the top) Note if the structure has two layers, the top view will show the top layer. * Top-right: Front view (looking from front, shows x-y plane, x increasing to the right, y increasing to the top) * Bottom-left: Right side view (looking from right side, shows z-y plane, z increasing to the right, y increasing to the top) * Bottom-right: Pure black image (no information) - **Current State:** - A text description of inventory information for each bot **Action Format:** - Use the mineBlock() function with these parameters: - botID: 'bot1', 'bot2', 'bot3' (for agent number 1, 2, 3, depending on number of bots available); 'bot0', 'bot1', 'bot2', 'bot3' (if 4 bots are available) - position: new Vec3(x,y,z) where (based on the top view image): * x: vertical axis (-3 to 2) * y: height axis (0 to 1) * z: horizontal axis (-3 to 2) - No space in coordinates x,y,z
**Available Blocks:**

'oak_fence', 'birch_log', 'coal_ore', 'bricks', 'sandstone', 'stone', 'iron_ore','dirt', 'grass_block', 'clay', 'bookshelf', 'acacia_fence', 'oak_log', 'crafting_table', 'anvil', 'cobweb', 'pumpkin', 'orange_concrete', 'gold_ore'

**Constraints:** - Each bot can only mine one block per timestep - Each bot has tools in their inventory to break blocks. Plan the mining sequence based on the tools in the inventory - You must mine blocks that is exposed to the air, you cannot mine blocks that are inside the block **Workspace Coordinates:** - Center: (0,0,0) - Platform size: 6x6 units (x and z axes) - Platform corners (**based on the top view image**): * Top-left: (2,0,-3) * Top-right: (2,0,2) * Bottom-right: (-3,0,2) * Bottom-left: (-3,0,-3) - Height: * 0: Ground level (one block above the workspace platform, lowest height to place blocks) * 1: Above ground level
**Orientation Strategy:** - Bot positions and orientations vary at each timestep, meaing bot's left might not be the same as the previous timestep, or the same as the other bots, or the same as the goal image - Recommended approach: First place blocks at known coordinates (like center 0,0,0 and one-block-to-the-left 0,0,-1) to establish reference points in bot's view - Use placed blocks and its positions in your action output to confirm your understanding of the coordinate system
**Response Format:** - Provide only action commands, each on a new line using new line as separator - Do not include explanations, markdown, or code blocks - A bot can be idle (no action) for a timestep, simply not output such bot's action - Use "NULL" if all bots are idle
**Example Response:**

mineBlock(bot1, new Vec3(-1,0,-1))
mineBlock(bot2, new Vec3(1,0,-1))
mineBlock(bot3, new Vec3(0,0,-1))

Figure 40: Sample rollouts of GPT-4o in a clearing task, part I.

**User Prompt**
Here are the observations at the current time step for each agent.

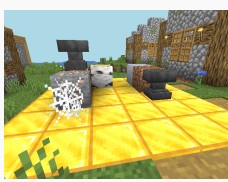 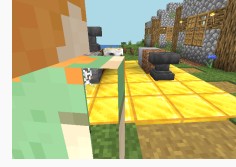 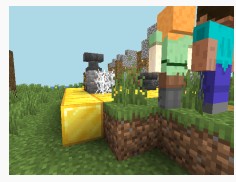 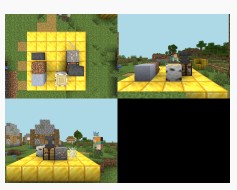

(a) Agent1 observation    (b) Agent2 observation    (c) Agent3 observation    (d) orthographic views

**Assistant**
[mineBlock(bot1, new Vec3(3,0,2)), mineBlock(bot2, new Vec3(2,0,1))]

**User Prompt**
Here are the observations at the current time step for each agent. The final image is the goal image. bot1 has ...

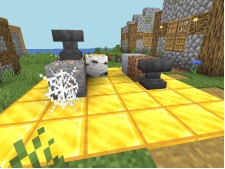 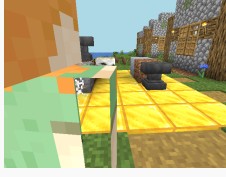 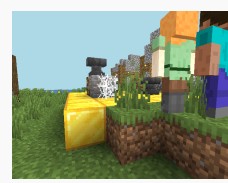 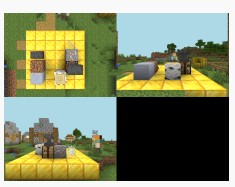

(e) Agent1 observation    (f) Agent2 observation    (g) Agent3 observation    (h) orthographic views

**Assistant**
[mineBlock(bot1, new Vec3(3,0,2)), mineBlock(bot2, new Vec3(2,0,1))]

**User Prompt**
Here are the observations at the current time step for each agent. The final image is the goal image. bot1 has ...

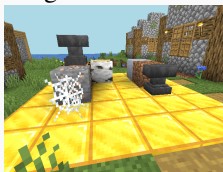 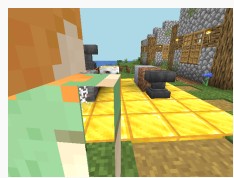 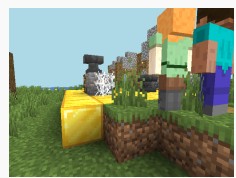 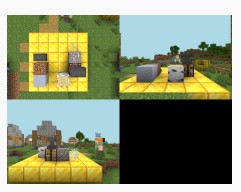

(i) Agent1 observation    (j) Agent2 observation    (k) Agent3 observation    (l) orthographic views

**Assistant**
[mineBlock(bot1, new Vec3(1,0,2)), mineBlock(bot2, new Vec3(2,0,1))]

**Comment**
The model did not predict correct block locations and failed to mine any blocks.

Figure 41: Sample rollouts of GPT-4o in a clearing task, part II.

**System Prompt**

You are controlling up to 4 bots in a Minecraft world. Your mission is control bots to sow seeds, wait until crops grow, and harvest them, in a designated farming area. The goal is to get a certain number of crops to any bots' inventory. Coordinate the bots' actions across multiple timesteps with observation feedback.

**Observations:** - **Images:** - Up to 4 first-person perspective images (one from each bot), all agent's view will reset to look at the middle of the platform for each timestep - The goal structure (last image): Contains three orthographic views of the target structure. * Top-left: Top view (looking down from above, shows x-z plane, x increasing to the right, z increasing to the top) Note if the structure has two layers, the top view will show the top layer. * Top-right: Front view (looking from front, shows x-y plane, x increasing to the right, y increasing to the top) * Bottom-left: Right side view (looking from right side, shows z-y plane, z increasing to the right, y increasing to the top) * Bottom-right: Pure black image (no information) - **Current State:** - A text description of inventory information for each bot **Action Format:** - Use the farm_work() function with these parameters: - botID: 'bot1', 'bot2', 'bot3' (for agent number 1, 2, 3, depending on number of bots available); 'bot0', 'bot1', 'bot2', 'bot3' (if 4 bots are available) - position: new Vec3(x,y,z) where (based on the top view image): * x: vertical axis (-3 to 3) * y: height axis (-1 to 0) * z: horizontal axis (-3 to 3) - action: 'sow', 'harvest' - crop: (optional for harvest action) A string, surround by single quote, the type of crop to sow (must be in bot's inventory) - No space in coordinates x,y,z - one action per bot per timestep

**Available Blocks:**

```
'water', 'farmland', 'cyan_concrete', 'stone', 'oak_wood',
'hay_block', 'glass','dirt', 'pink_wool', 'obsidian',
'smooth_quartz'
```

**Available Seeds**

```
'wheat_seeds', 'carrot', 'potato', 'beetroot_seeds'
```

**Constraints:** - Each bot can only act one action per timestep, either sow or harvest one crop. If you have two bots, you can only output two actions per timestep. - Observe the crop growth, a crop may take up to 3 timesteps to grow, you can only harvest when crops is full grown. You can only harvest what you sow. - You can only sow seeds on empty farmland, not on other crops - A good strategy is to sow seeds a farmland for the first three timesteps, and then harvest the crop sown on the first timestep, then harvest the crop sown on the second timestep, and sow seeds on the third timestep. - Ensure action bot has the necessary items (seeds) in their inventory - Do not assign multiple bots to perform the same action on the same crop plot simultaneously - Inventory may contain unneeded seeds or tools

**Workspace Coordinates:** - Center: (0,0,0) - Platform size: could be random size, surround by a single type of block, could be a oak, or dirt, ice, etc. - Platform corners (based on the top view image): * Top-left: (-3,0,-3) * Top-right: (3,0,3) * Bottom-right: (-3,0,3) * Bottom-left: (-3,0,-3) - Height: * -1: Farmland level (where the sow action can be performed) * 0: Crop level (where the harvest action can be performed)

**Orientation Strategy:** - Bot positions and orientations vary at each timestep, meaing bot's left might not be the same as the previous timestep, or the same as the other bots, or the same as the goal image - Recommended approach: First place blocks at known coordinates (like center 0,0,0 and one-block-to-the-left 0,0,-1) to establish reference points in bot's view - Use placed blocks and its positions in your action output to confirm your understanding of the coordinate system

**Response Format:** - Provide only action commands, each on a new line using new line separator - Do not include explanations, markdown, or code blocks - A bot can be idle (no action) for a timestep, simply not output such bot's action - Use "NULL" if all bots are idle **Example Response:**

```
farm_work(bot1, new Vec3(0,-1,1), 'sow', 'wheat_seeds')
farm_work(bot2, new Vec3(1,0,0), 'harvest')
```

Figure 42: Sample rollouts of GPT-4o in a farming task, part I.

**User Prompt**
Here are the observations at the current time step for each agent. Please harvest the crops when they are ready! You can only harvest what you sow. You can only sow seeds on empty plots, and you can only sow what you have in your inventory. When harvesting, do not specify the crop type. Here is the inventory status for each bot: bot1 has 7 xxx

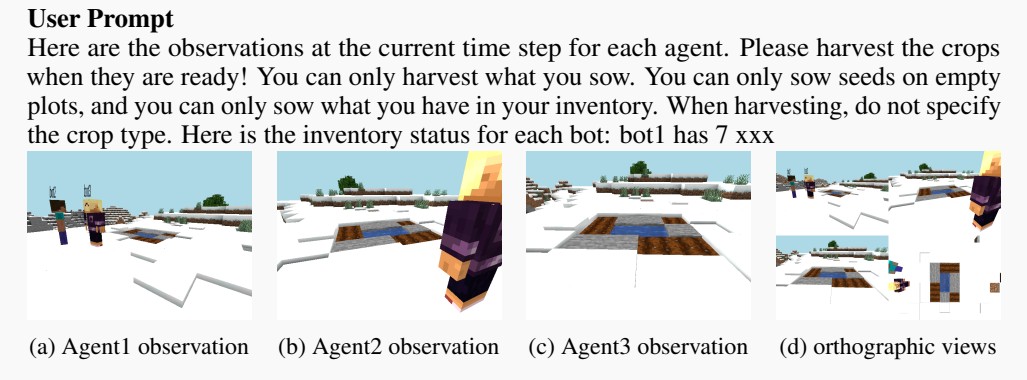

(a) Agent1 observation     (b) Agent2 observation     (c) Agent3 observation     (d) orthographic views

**Assistant**
farmwork(bot1, new Vec3(0, 0, -1), 'sow', 'potato')
farmwork(bot2, new Vec3(1, 0, 0), 'sow', 'potato')
farmwork(bot3, new Vec3(-1, 0, 1), 'sow', 'potato')

Figure 43: Sample rollouts of GPT-4o in a farming task, part II.

**User Prompt**
Here are the observations at the current time step for each agent. bot1 has ...

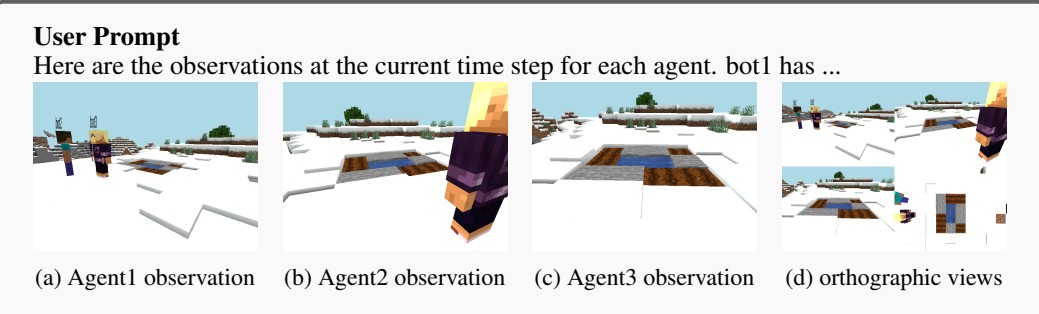

(a) Agent1 observation    (b) Agent2 observation    (c) Agent3 observation    (d) orthographic views

**Assistant**
farmwork(bot1, new Vec3(0, 0, 1), 'sow', 'potato'),
farmwork(bot2, new Vec3(0, 0, -1), 'sow', 'potato'),
farmwork(bot3, new Vec3(1, 0, 0), 'sow', 'potato')

**User Prompt**
Here are the observations at the current time step for each agent. bot1 has ...

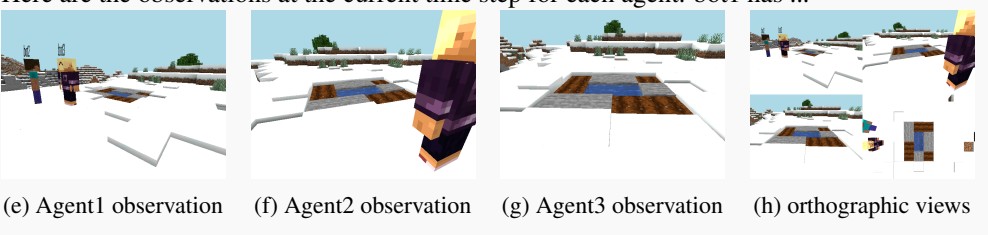

(e) Agent1 observation    (f) Agent2 observation    (g) Agent3 observation    (h) orthographic views

**Assistant**
farmwork(bot1, new Vec3(-1, 0, 0), 'sow', 'potato'),
farmwork(bot2, new Vec3(1, 0, 0), 'sow', 'potato'),
farmwork(bot3, new Vec3(0, 0, 1), 'sow', 'potato')

**Comment**
The model does not follow instruction, and keeps predicting the sowing actions without harvesting.

Figure 44: Sample rollouts of GPT-4o in a farming task, part III.

**System Prompt**
You are controlling up to 4 bots in a Minecraft world. Your mission is to control bots to smelt for specific goal items for a specific amount, by coordinating the bots' actions across multiple timesteps with observation feedback. To get final goal items, you need to obtain resouces like fuel, kill mobs or collect item to be smelted, put items and fuels in furnace, and take out the smelted item so that it is in any bot's inventory.
**Observations:** - **Images:** - Up to 4 first-person perspective images (one from each bot), all agent's view will reset to look at the middle of the platform for each timestep - The goal structure (last image): Contains three orthographic views of the target structure. * Top-left: Top view (looking down from above, shows x-z plane, x increasing to the right, z increasing to the top) Note if the structure has two layers, the top view will show the top layer. * Top-right: Front view (looking from front, shows x-y plane, x increasing to the right, y increasing to the top) * Bottom-left: Right side view (looking from right side, shows z-y plane, z increasing to the right, y increasing to the top) * Bottom-right: Pure black image (no information) - **Grid Information:** - The grid information that describes the blocks on the platform, as shown in the last image, is a 2D grid of blocks, with the center of the grid being (0,0,0). - **Current State:** - A text description of inventory information for each bot
**Action Format:** - "obtainBlock(botID, position)" - "putFuelFurnace(botID, 'item', position)" - "putItemFurnace(botID, 'item', position)" - "takeOutFurnace(botID, position)" - "killMob(botID, position)" with these parameters: - botID: 'bot1', 'bot2', 'bot3' (for agent number 1, 2, 3, depending on number of bots available); 'bot0', 'bot1', 'bot2', 'bot3' (if 4 bots are available) - item: A string, surround by single quote, the type of item to place (must be in bot's inventory) - position: new Vec3(x,y,z) where (based on the top view image): * x: vertical axis (-2 to 2) * y: height axis (0 to 1) * z: horizontal axis (-2 to 2) - No space in coordinates x,y,z - one action per bot per timestep
**Available Blocks:**

```
'acacia_log', 'acacia_planks', 'birch_log', 'birch_planks',
'chicken', 'coal_ore','cobblestone', 'cow', 'furnace',
'gold_ore', 'iron_ore', 'mooshroom', 'oak_log',
'oak_planks', 'pig', 'quartz_block', 'rabbit',
'red_sand', 'sandstone', 'sheep','spruce_log',
'spruce_planks', 'wet_sponge'
```

**Constraints:** - Each bot can only act one action per timestep. If you have two bots, you can only output two actions per timestep. – Bots must take out the item from the furnace to count as a successful smelting target acquired - Only one type of fuel can be used for one furnace - Bot can only place items from its own inventory - Bot can also obtain blocks needed for smelting. - Bot do not need to get those resources if they already in inventory. - Inventory may contain unneeded items - No overlapping items at the same position
**Workspace Coordinates:** - Center: (0,0,0) - Platform size: 5x5 units (x and z axes) - Platform corners (**based on the top view image**): * Top-left: (2,0,-2) * Top-right: (2,0,2) * Bottom-right: (-2,0,2) * Bottom-left: (-2,0,-2) - Height: * 0: Ground level (one block above the workspace platform, lowest height to place blocks) * 1: Above ground level
**Orientation Strategy:** - Bot positions and orientations vary at each timestep, meaing bot's left might not be the same as the previous timestep, or the same as the other bots, or the same as the goal image - Recommended approach: First place blocks at known coordinates (like center 0,0,0 and one-block-to-the-left 0,0,-1) to establish reference points in bot's view - Use placed blocks and its positions in your action output to confirm your understanding of the coordinate system
**Response Format:** - Provide only action commands, each on a new line using new line separator - Do not include explanations, markdown, or code blocks - A bot can be idle (no action) for a timestep, simply not output such bot's action - Use "NULL" if all bots are idle
**Example Response:**

```
killMob(bot1, new Vec3(1,0,1))
putFuelFurnace(bot2, 'oak_log', new Vec3(0,0,0))
putItemFurnace(bot3, 'porkchop', new Vec3(2,0,2))
```

Figure 45: Sample system prompt of GPT-4o in a smelt task, part I.

**System Prompt**
**\*\*Recepit:\*\***
Cooking Food:
1. To cook a 'cooked_beef', I need 'beef'. To get 'beef', I need to kill a 'cow' or a 'mushroom'.
2. To cook a 'cooked_porkchop', I need 'porkchop'. To get 'porkchop', I need to kill a 'pig'.
3. To cook a 'cooked_mutton', I need 'mutton'. To get 'mutton', I need to kill a 'sheep'.
4. To cook a 'cooked_chicken', I need 'chicken'. To get 'chicken', I need to kill a 'chicken'.
5. To cook a 'cooked_rabbit', I need 'rabbit'. To get 'rabbit', I need to kill a 'rabbit'.
6. To cook a 'cooked_cod', I need 'cod'.
7. To cook a 'cooked_salmon', I need 'salmon'.
8. To cook a 'baked_potato', I need a 'potato'.
Crafting Items:
1. To craft a 'gold_ingot', I need 'gold_ore'. To get 'gold_ore', I need to obtain 'gold_ore' blocks with a pickaxe.
2. To craft an 'iron_ingot', I need 'iron_ore'. To get 'iron_ore', I need to obtain 'iron_ore' blocks with a pickaxe.
3. To craft 'glass', I need 'red_sand'. To get 'red_sand', I need to obtain 'red_sand'.
4. To craft 'smooth_sandstone', I need 'sandstone'. To get 'sandstone', I need to obtain 'sandstone' with a pickaxe.
5. To craft 'stone', I need 'cobblestone'. To get 'cobblestone', I need to obtain 'cobblestone' with a pickaxe.
Fuel Sources:
1. To fuel the furnace, I can use 'coal'. To get 'coal', I need to obtain 'coal_ore'.
2. To fuel the furnace, I can use 'lava_bucket', 'coal_block', 'charcoal', .
3. To fuel the furnace, I can use 'oak_log', 'birch_log', 'acacia_log', 'spruce_log', 'oak_planks', 'birch_planks', 'acacia_planks', or 'spruce_planks'.

Figure 46: Sample system prompt of GPT-4o in a smelt task, part II.

**System Prompt**
You are controlling up to 4 bots in a Minecraft world. Your mission is to build a specific structure on a platform by coordinating the bots' actions across multiple timesteps with observation feedback.
**Observations:** - **Images:** - Up to 4 first-person perspective images (one from each bot), all agent's view will reset to look at the middle of the platform for each timestep - The goal structure (last image): Contains three orthographic views of the target structure. * Top-left: Top view (looking down from above, shows x-z plane, x increasing to the right, z increasing to the top) Note if the structure has two layers, the top view will show the top layer. * Top-right: Front view (looking from front, shows x-y plane, x increasing to the right, y increasing to the top) * Bottom-left: Right side view (looking from right side, shows z-y plane, z increasing to the right, y increasing to the top) * Bottom-right: Pure black image (no information) - **Current State:** - A text description of inventory information for each bot
**Action Format:** - Use the placeItem() function with these parameters: - botID: 'bot1', 'bot2', 'bot3' (for agent number 1, 2, 3, depending on number of bots available); 'bot0', 'bot1', 'bot2', 'bot3' (if 4 bots are available) - block: The type of block to place (must be in bot's inventory) - position: new Vec3(x,y,z) where (based on the top view image): * x: vertical axis (-2 to 2) * y: height axis (0 to 1) * z: horizontal axis (-2 to 2) - No space in coordinates x,y,z **Available Blocks:**

'oak_fence', 'birch_log', 'coal_ore', 'bricks', 'sandstone', 'stone',
'iron_ore', 'gold_ore', 'sponge', 'sea_lantern', 'dirt', 'grass_block',
'clay', 'oak_planks', 'emerald_block', 'pumpkin', 'orange_concrete',
'purple_wool', 'end_stone', 'bookshelf', 'acacia_fence', 'oak_log'

**Constraints:** - Each bot can only place blocks from its own inventory - Inventory may contain unneeded blocks - No overlapping blocks at the same position - No floating blocks (must build on existing structures) - Each bot can place at most one block per timestep
**Workspace Coordinates:** - Center: (0,0,0) - Platform size: 5x5 units (x and z axes) - Platform corners (**based on the top view image**): * Top-left: (2,0,-2) * Top-right: (2,0,2) * Bottom-right: (-2,0,2) * Bottom-left: (-2,0,-2) - Height: * 0: Ground level (one block above the workspace platform, lowest height to place blocks) * 1: Above ground level
**Orientation Strategy:** - Bot positions and orientations vary at each timestep, meaning bot's left might not be the same as the previous timestep, or the same as the other bots, or the same as the goal image - Recommended approach: First place blocks at known coordinates (like center 0,0,0 and one-block-to-the-left 0,0,-1) to establish reference points in bot's view - Use placed blocks and its positions in your action output to confirm your understanding of the coordinate system
**Response Format:** - Provide only action commands, each on a new line using new line as separator - Do not include explanations, markdown, or code blocks - A bot can be idle (no action) for a timestep, simply not output such bot's action - Use "NULL" if all bots are idle
**Example Response:**

placeItem(bot1, 'pumpkin', new Vec3(-1,0,-1))
placeItem(bot2, 'purple_wool', new Vec3(1,0,-1))
placeItem(bot3, 'coal_ore', new Vec3(0,0,-1))

Figure 47: Sample system (developer) prompt of o1 in a building task, o1 part I.

**User Prompt**
what is the block and its position in the image?

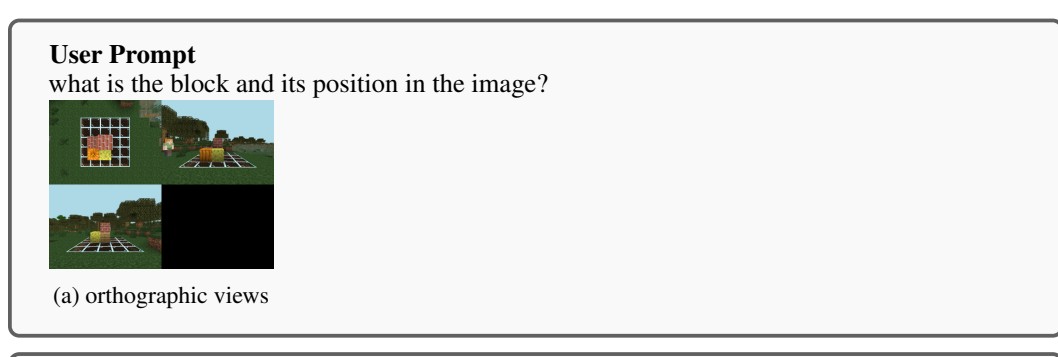

(a) orthographic views

**Assistant**
```
'oak_planks' at new Vec3(0,0,0)
'bricks' at new Vec3(0,0,-1)
'sponge' at new Vec3(-1,0,0)
'pumpkin' at new Vec3(-1,0,-1)
'bricks' at new Vec3(0,1,0)
```

Figure 48: Sample one shot prompt and answer example for special coordinate system, o1 part II.

**User Prompt**

Here are the observations at the current time step for each agent. bot1 has ...

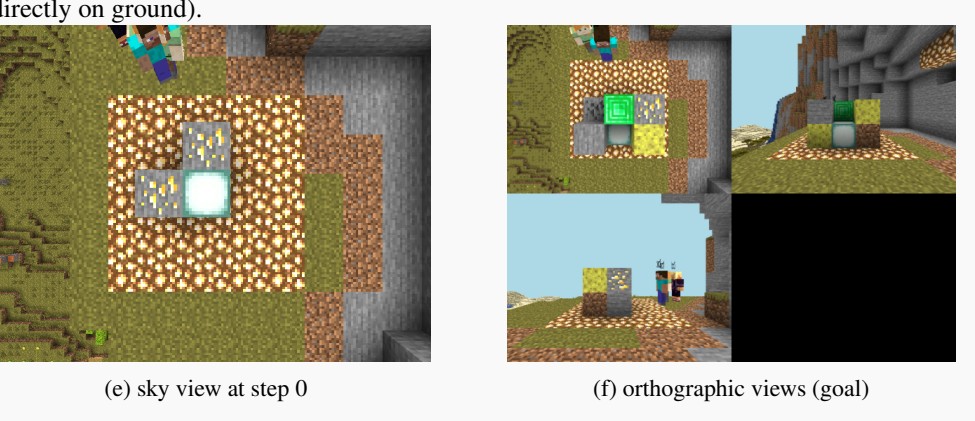

(a) Agent1 observation  (b) Agent2 observation  (c) Agent3 observation  (d) orthographic views

**Assistant**

```
placeItem(bot1, 'stone', new Vec3(-1,0,0))
placeItem(bot2, 'gold_ore', new Vec3(1,0,0))
placeItem(bot3, 'sea_lantern', new Vec3(0,0,0))
placeItem(bot1, 'gold_ore', new Vec3(0,0,-1))
placeItem(bot2, 'emerald_block', new Vec3(0,1,0))
placeItem(bot3, 'stone', new Vec3(0,1,-1))
```

**Comment**

The model is able to recognize the blocks in the goal image and associate blocks with their name.

The model does not follow instruction, generating two actions per agent, instead of the required one action per agent as specified in the system prompt. This result only random one out of two actions for each bot is being executed.

The model confused the spacial absolute space, miss placing the sea_lantern at (0,0,0), where it should be one block below the origin.

The model confused the spacial relative position, and fail to eductive from three orthographic views that the gold_ore is on the second layer (above another block), not the first layer (directly on ground).

(e) sky view at step 0                    (f) orthographic views (goal)

Figure 49: Step 1 of the sample building task, o1 part III.

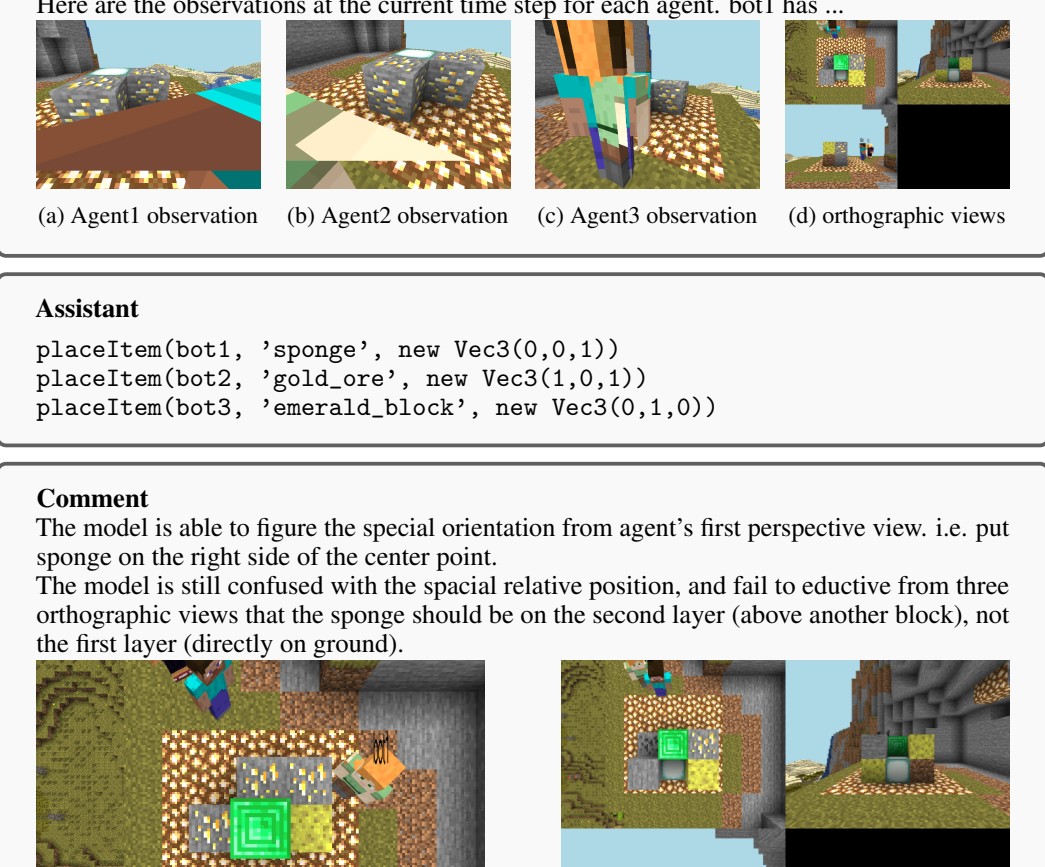

**User Prompt**

Here are the observations at the current time step for each agent. bot1 has ...

(a) Agent1 observation  (b) Agent2 observation  (c) Agent3 observation  (d) orthographic views

**Assistant**

```
placeItem(bot1, 'sponge', new Vec3(0,0,1))
placeItem(bot2, 'gold_ore', new Vec3(1,0,1))
placeItem(bot3, 'emerald_block', new Vec3(0,1,0))
```

**Comment**

The model is able to figure the special orientation from agent's first perspective view. i.e. put sponge on the right side of the center point.

The model is still confused with the spacial relative position, and fail to eductive from three orthographic views that the sponge should be on the second layer (above another block), not the first layer (directly on ground).

(e) sky view at step 0  (f) orthographic views (goal)

Figure 50: Step 2 of the sample building task, o1 part IV.

**System Prompt**
You are controlling up to 4 bots in a Minecraft world. Your mission is to build a specific structure on a platform by coordinating the bots' actions across multiple timesteps with observation feedback.
**Observations:** - **Images:** - Up to 4 first-person perspective images (one from each bot), all agent's view will reset to look at the middle of the platform for each timestep - The goal structure (last image): Contains three orthographic views of the target structure. * Top-left: Top view (looking down from above, shows x-z plane, x increasing to the right, z increasing to the top) Note if the structure has two layers, the top view will show the top layer. * Top-right: Front view (looking from front, shows x-y plane, x increasing to the right, y increasing to the top) * Bottom-left: Right side view (looking from right side, shows z-y plane, z increasing to the right, y increasing to the top) * Bottom-right: Pure black image (no information) - **Current State:** - A text description of inventory information for each bot
**Action Format:** - Use the placeItem() function with these parameters: - botID: 'bot1', 'bot2', 'bot3' (for agent number 1, 2, 3, depending on number of bots available); 'bot0', 'bot1', 'bot2', 'bot3' (if 4 bots are available) - block: The type of block to place (must be in bot's inventory) - position: new Vec3(x,y,z) where (based on the top view image): * x: vertical axis (-2 to 2) * y: height axis (0 to 1) * z: horizontal axis (-2 to 2) - No space in coordinates x,y,z **Available Blocks:**

'oak_fence', 'birch_log', 'coal_ore', 'bricks', 'sandstone', 'stone', 'iron_ore', 'gold_ore', 'sponge', 'sea_lantern', 'dirt', 'grass_block', 'clay', 'oak_planks', 'emerald_block', 'pumpkin', 'orange_concrete', 'purple_wool', 'end_stone', 'bookshelf', 'acacia_fence', 'oak_log'

**Constraints:** - Each bot can only place blocks from its own inventory - Inventory may contain unneeded blocks - No overlapping blocks at the same position - No floating blocks (must build on existing structures) - Each bot can place at most one block per timestep
**Workspace Coordinates:** - Center: (0,0,0) - Platform size: 5x5 units (x and z axes) - Platform corners (**based on the top view image**): * Top-left: (2,0,-2) * Top-right: (2,0,2) * Bottom-right: (-2,0,2) * Bottom-left: (-2,0,-2) - Height: * 0: Ground level (one block above the workspace platform, lowest height to place blocks) * 1: Above ground level
**Orientation Strategy:** - Bot positions and orientations vary at each timestep, meaning bot's left might not be the same as the previous timestep, or the same as the other bots, or the same as the goal image - Recommended approach: First place blocks at known coordinates (like center 0,0,0 and one-block-to-the-left 0,0,-1) to establish reference points in bot's view - Use placed blocks and its positions in your action output to confirm your understanding of the coordinate system
**Response Format:** - Provide only action commands, each on a new line using new line as separator - Do not include explanations, markdown, or code blocks - A bot can be idle (no action) for a timestep, simply not output such bot's action - Use "NULL" if all bots are idle
**Example Response:**

placeItem(bot1, 'pumpkin', new Vec3(-1,0,-1))
placeItem(bot2, 'purple_wool', new Vec3(1,0,-1))
placeItem(bot3, 'coal_ore', new Vec3(0,0,-1))

Figure 51: Sample system (developer) prompt of Claude 3.7 Sonnet in a building task, Claude part I.

**User Prompt**

what is the block and its position in the image?

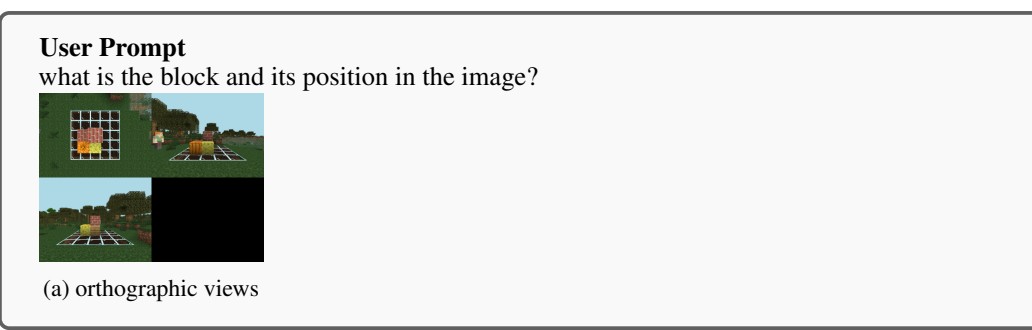

(a) orthographic views

**Assistant**

```
'oak_planks' at new Vec3(0,0,0)
'bricks' at new Vec3(0,0,-1)
'sponge' at new Vec3(-1,0,0)
'pumpkin' at new Vec3(-1,0,-1)
'bricks' at new Vec3(0,1,0)
```

Figure 52: Sample one shot prompt and answer example for special coordinate system, Claude part II.

**User Prompt**

Here are the observations at the current time step for each agent. bot1 has ...

(a) Agent1 observation     (b) Agent2 observation     (c) Agent3 observation     (d) orthographic views

---

**Assistant**

```
placeItem(bot1, 'sandstone', new Vec3(0,0,0))
placeItem(bot2, 'coal_ore', new Vec3(1,0,0))
placeItem(bot3, 'end_stone', new Vec3(-1,0,0))
```

---

**Comment**

The model is able to recognize the blocks in the goal image and associate blocks with their name.

The model can follow the instruction to generate required format for actions. Generated actions satisfy all requirements given in the system prompt.

The model can understand the spacial relative position, and able to deductive from three orthographic views that it needs to place blocks that on the first layer first.

The model is not able to figure the special orientation from agent's first perspective view. i.e. the first three blocks should be form a horizontal line but actually formed a vertical line.

(e) sky view at step 0        (f) orthographic views (goal)

Figure 53: Step 1 of the sample building task, Claude part III.

**User Prompt**

Here are the observations at the current time step for each agent. bot1 has ...

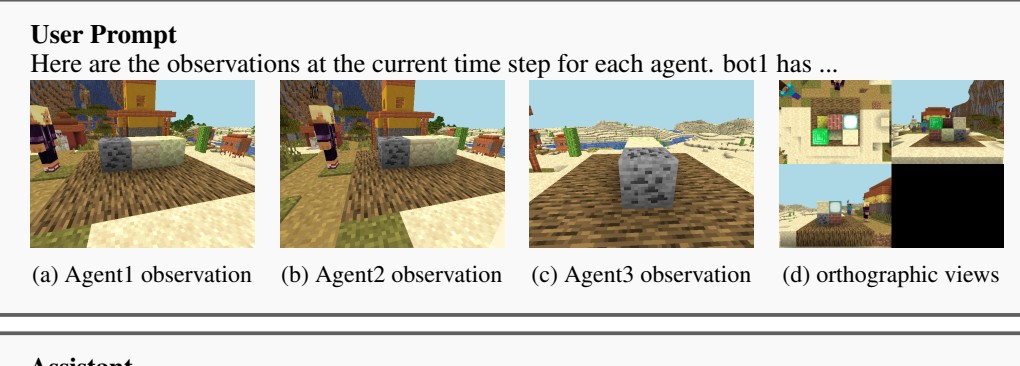

(a) Agent1 observation    (b) Agent2 observation    (c) Agent3 observation    (d) orthographic views

**Assistant**

```
placeItem(bot1, 'sea_lantern', new Vec3(-1,0,1))
placeItem(bot2, 'sandstone', new Vec3(0,0,1))
placeItem(bot3, 'emerald_block', new Vec3(-1,0,-1))
```

**Comment**

The model starts to be confused with the spacial relative position, and fail to eductive from three orthographic views that the sea_lantern should be on the second layer (above another block), not the first layer (directly on ground).

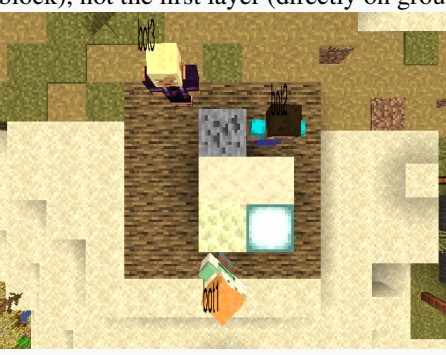

(e) sky view at step 0

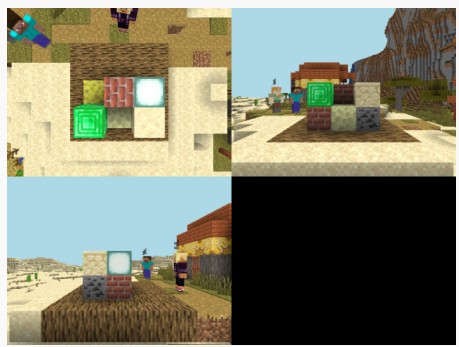

(f) orthographic views (goal)

Figure 54: Step 2 of the sample building task, Claude part IV.

**User Prompt**

Here are the observations at the current time step for each agent. bot1 has ...

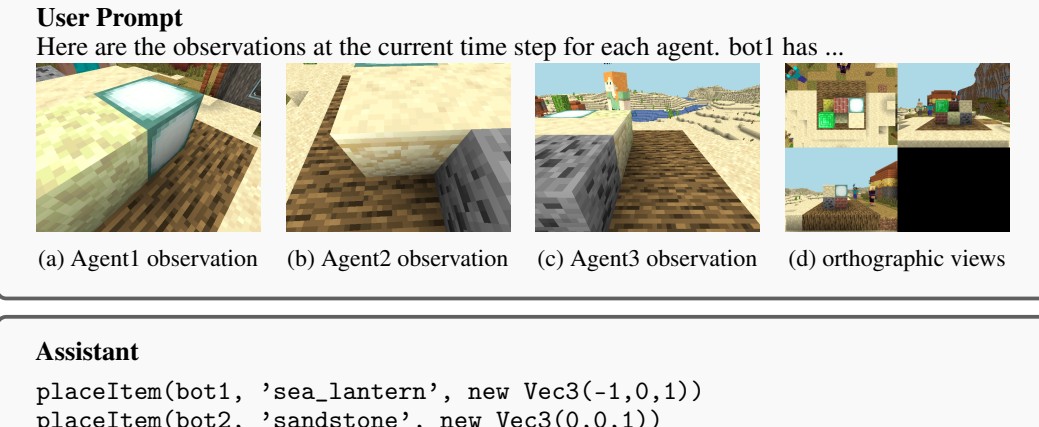

(a) Agent1 observation    (b) Agent2 observation    (c) Agent3 observation    (d) orthographic views

**Assistant**

```
placeItem(bot1, 'sea_lantern', new Vec3(-1,0,1))
placeItem(bot2, 'sandstone', new Vec3(0,0,1))
placeItem(bot3, 'emerald_block', new Vec3(-1,0,-1))
```

**Comment**

The model begins to randomly place disturbing blocks (those not on the goal image but in agents' inventories) on the ground.

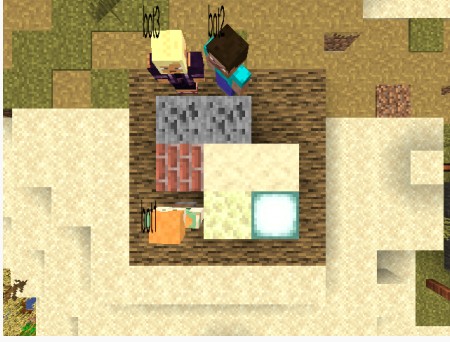
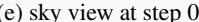

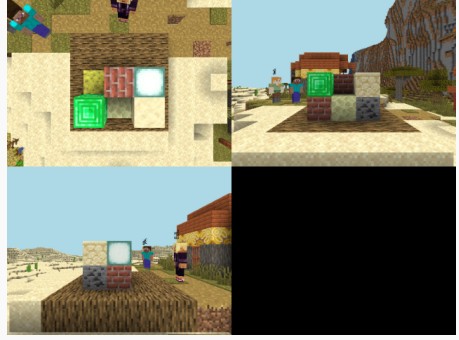

(e) sky view at step 0        (f) orthographic views (goal)

Figure 55: Step 3 of the sample building task, Claude part IV.

**User Prompt**

Here are the observations at the current time step for each agent. bot1 has ...

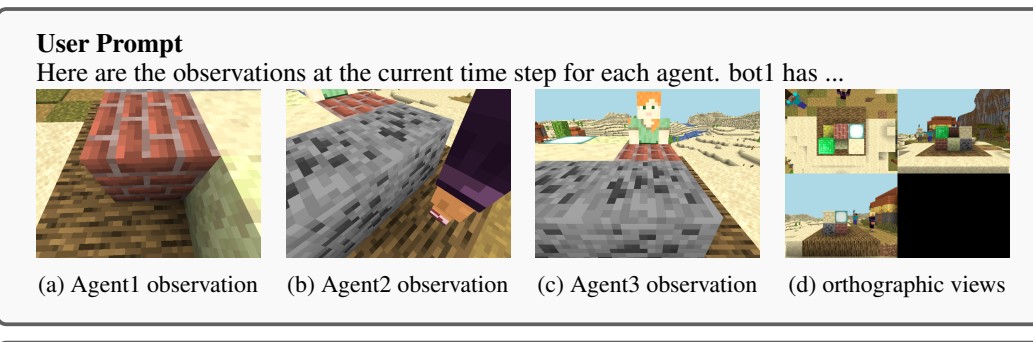

(a) Agent1 observation    (b) Agent2 observation    (c) Agent3 observation    (d) orthographic views

**Assistant**

```
placeItem(bot1, 'bricks', new Vec3(0,0,-1))
placeItem(bot2, 'coal_ore', new Vec3(1,0,-1))
placeItem(bot3, 'emerald_block', new Vec3(0,1,-1))
```

**Comment**

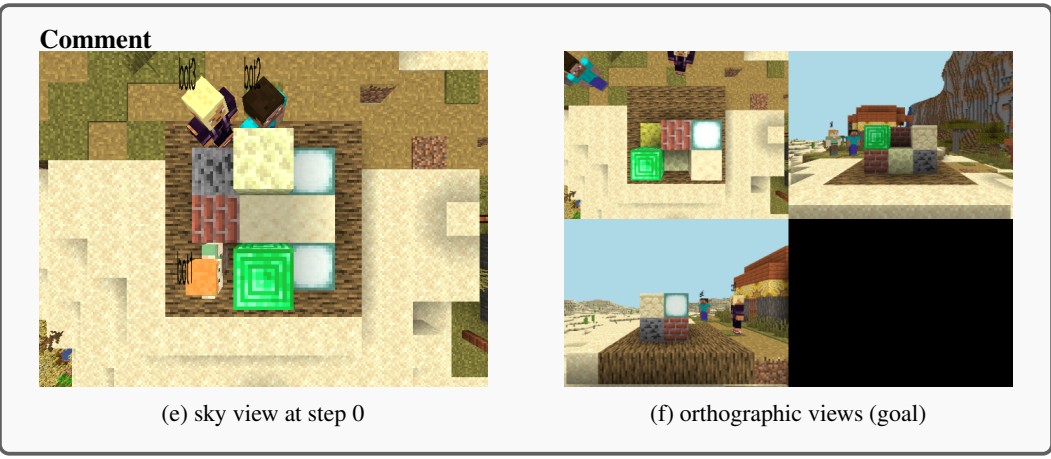

(e) sky view at step 0        (f) orthographic views (goal)

Figure 56: Step 4 of the sample building task, Claude part V.

**User Prompt**

Here are the observations at the current time step for each agent. bot1 has ...

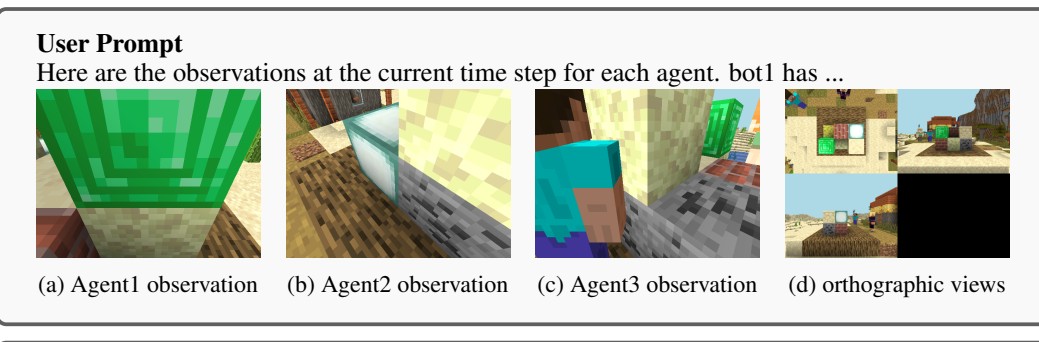

(a) Agent1 observation    (b) Agent2 observation    (c) Agent3 observation    (d) orthographic views

**Assistant**

```
placeItem(bot1, 'emerald_block', new Vec3(-1,1,0))
placeItem(bot2, 'sea_lantern', new Vec3(1,0,1))
placeItem(bot3, 'end_stone', new Vec3(1,1,0))
```

**Comment**

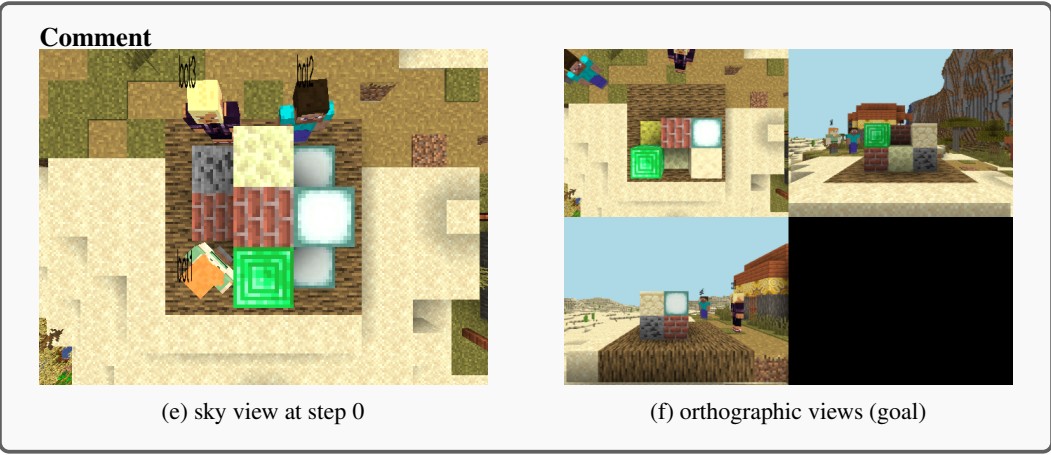

(e) sky view at step 0                    (f) orthographic views (goal)

Figure 57: Step 5 of the sample building task, Claude part VI.

**User Prompt**

Here are the observations at the current time step for each agent. bot1 has ...

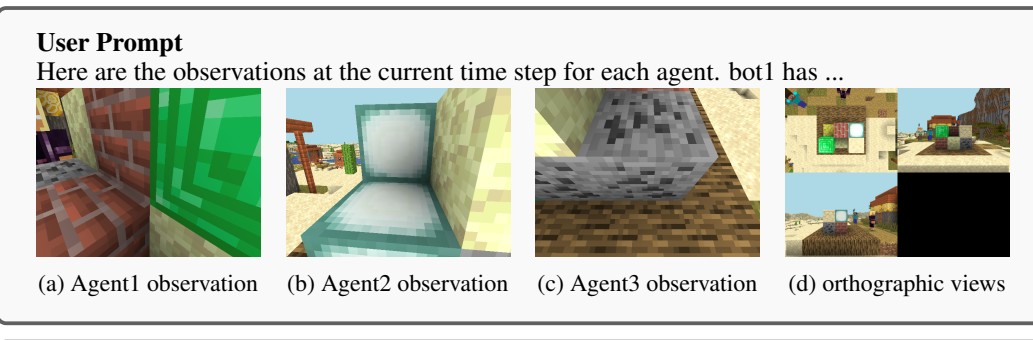

(a) Agent1 observation    (b) Agent2 observation    (c) Agent3 observation    (d) orthographic views

**Assistant**

```
placeItem(bot1, 'bricks', new Vec3(0,1,0))
placeItem(bot2, 'sea_lantern', new Vec3(0,1,1))
placeItem(bot3, 'end_stone', new Vec3(1,0,-1))
```

**Comment**

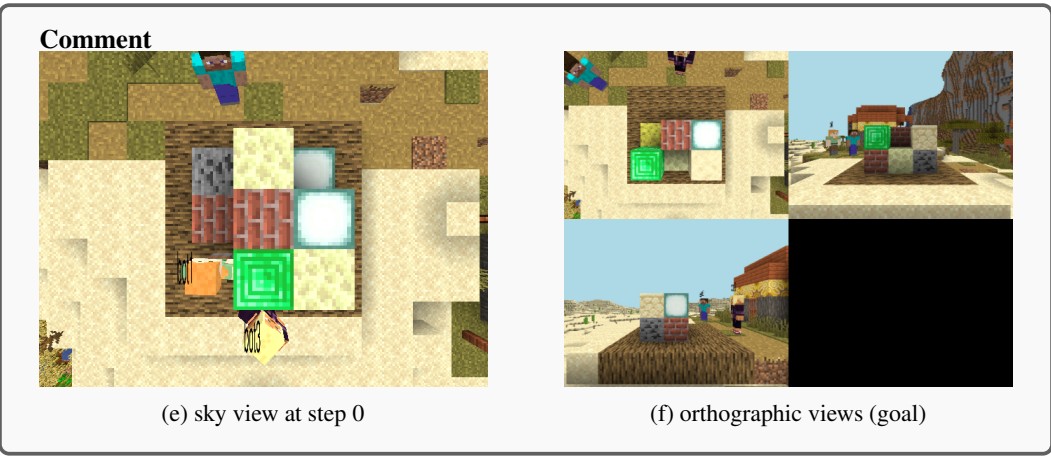

(e) sky view at step 0    (f) orthographic views (goal)

Figure 58: Step 6 of the sample building task, Claude part VII.

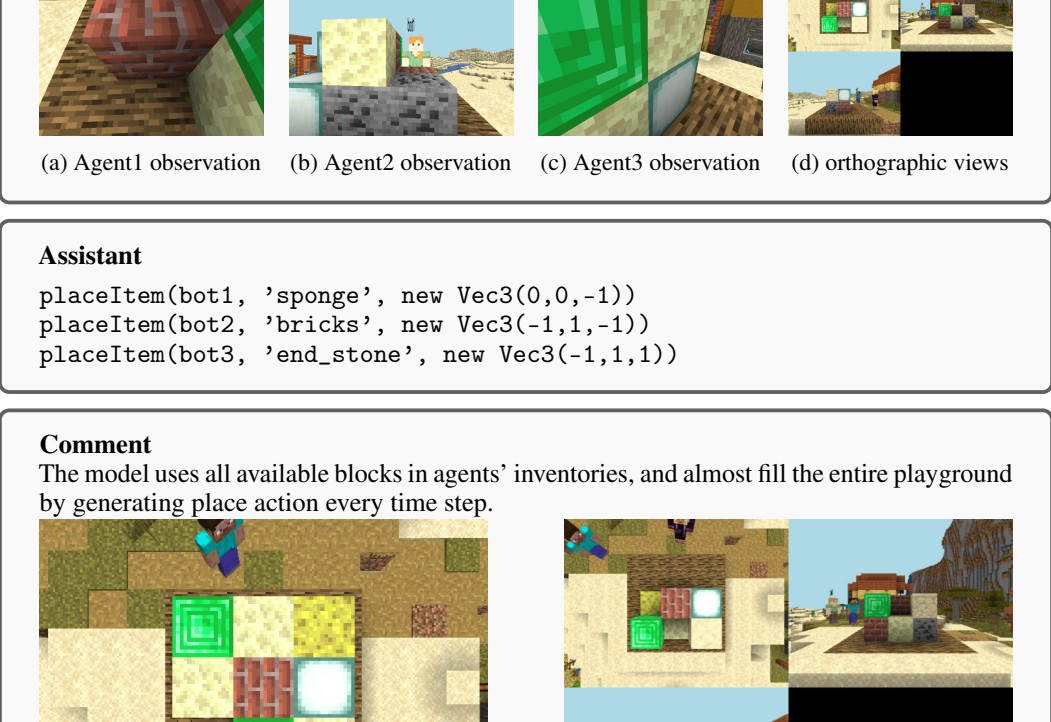

**User Prompt**
Here are the observations at the current time step for each agent. bot1 has ...

(a) Agent1 observation    (b) Agent2 observation    (c) Agent3 observation    (d) orthographic views

**Assistant**

```
placeItem(bot1, 'sponge', new Vec3(0,0,-1))
placeItem(bot2, 'bricks', new Vec3(-1,1,-1))
placeItem(bot3, 'end_stone', new Vec3(-1,1,1))
```

**Comment**
The model uses all available blocks in agents' inventories, and almost fill the entire playground by generating place action every time step.

(e) sky view at step 0            (f) orthographic views (goal)

Figure 59: Step 7 of the sample building task, Claude part VIII.

 # K    Dataset Statistics Tables

Table 14: Building Task Diversity Statistics

| Diversity | Type | Count | Percentage |
|---|---|---|---|
| **Action Sequences** | | | |
| | 3 | 7,777 | 51.85% |
| | 2 | 3,207 | 21.38% |
| | 4 | 3,091 | 20.61% |
| | 5 | 483 | 3.22% |
| | 6 | 440 | 2.93% |
| **Agents** | | | |
| | 3 | 7,505 | 50.03% |
| | 2 | 7,493 | 49.97% |
| **Scenes** | | | |
| | ice_on_water | 2,555 | 17.04% |
| | mountain_half | 2,553 | 17.03% |
| | village | 2,482 | 16.55% |
| | desert_village | 2,480 | 16.53% |
| | snow_mountain | 2,478 | 16.52% |
| | swamp | 2,450 | 16.34% |
| **Background Types** | | | |
| | stone | 1,530 | 10.20% |
| | pink_wool | 1,527 | 10.19% |
| | glowstone | 1,522 | 10.15% |
| | obsidian | 1,511 | 10.08% |
| | glass | 1,509 | 10.07% |
| | smooth_quartz | 1,499 | 10.00% |
| | hay_block | 1,494 | 9.96% |
| | gold_block | 1,473 | 9.82% |
| | oak_wood | 1,471 | 9.81% |
| | cyan_concrete | 1,462 | 9.75% |
| **Target Types** | | | |
| | bricks | 10,391 | 9.92% |
| | sponge | 5,438 | 5.19% |
| | coal_ore | 5,370 | 5.13% |
| | grass_block | 5,327 | 5.09% |
| | clay | 5,318 | 5.08% |
| | sea_lantern | 5,296 | 5.06% |
| | orange_concrete | 5,287 | 5.05% |
| | pumpkin | 5,269 | 5.03% |
| | purple_wool | 5,257 | 5.02% |
| | gold_ore | 5,247 | 5.01% |
| | oak_fence | 5,234 | 5.00% |
| | oak_planks | 5,216 | 4.98% |
| | birch_log | 5,184 | 4.95% |
| | stone | 5,182 | 4.95% |
| | sandstone | 5,176 | 4.94% |
| | emerald_block | 5,164 | 4.93% |
| | iron_ore | 5,160 | 4.93% |
| | dirt | 5,124 | 4.89% |
| | end_stone | 5,119 | 4.89% |
| **Target Counts** | | | |
| | 6 | 5,653 | 37.69% |
| | 7 | 2,625 | 17.50% |
| | 8 | 2,573 | 17.15% |
| | 5 | 2,122 | 14.15% |
| | 10 | 526 | 3.51% |
| | 12 | 515 | 3.43% |
| | 9 | 496 | 3.31% |
| | 11 | 488 | 3.25% |
| **Dimensional Shapes** | | | |
| | [3, 1, 2] | 3,859 | 25.73% |
| | [4, 1, 2] | 3,770 | 25.14% |
| | [2, 3, 2] | 3,695 | 24.63% |
| | [2, 2, 2] | 3,674 | 24.49% |

Table 15: Clearing Task Diversity Statistics

| Diversity | Type | Count | Percentage |
|---|---|---|---|
| **Action Sequences** | | | |
| | 4 | 4,027 | 27.51% |
| | 5 | 3,751 | 25.61% |
| | 6 | 3,270 | 22.32% |
| | 3 | 1,561 | 10.66% |
| | 7 | 1,396 | 9.53% |
| | 8 | 424 | 2.89% |
| | 9 | 133 | 0.91% |
| | 2 | 79 | 0.54% |
| **Agents** | | | |
| | 2 | 7,358 | 50.28% |
| | 3 | 7,283 | 49.72% |
| **Scenes** | | | |
| | desert_village | 3,012 | 20.56% |
| | snow_mountain | 2,948 | 20.13% |
| | swamp | 2,929 | 20.00% |
| | ice_on_water | 2,894 | 19.76% |
| | village | 2,858 | 19.54% |
| **Background Types** | | | |
| | smooth_quartz | 1,405 | 9.59% |
| | pink_wool | 1,357 | 9.27% |
| | gold_block | 1,353 | 9.24% |
| | oak_wood | 1,334 | 9.10% |
| | hay_block | 1,332 | 9.09% |
| | cyan_concrete | 1,332 | 9.09% |
| | grass_block | 1,328 | 9.06% |
| | glass | 1,325 | 9.04% |
| | glowstone | 1,309 | 8.93% |
| | stone | 1,302 | 8.89% |
| | obsidian | 1,264 | 8.63% |
| **Target Counts** | | | |
| | 6 | 4,310 | 29.43% |
| | 5 | 2,499 | 17.07% |
| | 4 | 2,436 | 16.64% |
| | 8 | 1,843 | 12.58% |
| | 7 | 1,803 | 12.31% |
| | 9 | 1,750 | 11.95% |
| **Target Types** | | | |
| | oak_fence | 5,879 | 6.45% |
| | grass_block | 5,836 | 6.40% |
| | clay | 5,816 | 6.38% |
| | oak_log | 5,772 | 6.33% |
| | sandstone | 5,748 | 6.30% |
| | acacia_fence | 5,744 | 6.30% |
| | birch_log | 5,732 | 6.28% |
| | bookshelf | 5,726 | 6.28% |
| | stone | 5,709 | 6.26% |
| | bricks | 5,695 | 6.25% |
| | crafting_table | 5,684 | 6.23% |
| | dirt | 5,671 | 6.22% |
| | cobweb | 5,605 | 6.15% |
| | iron_ore | 5,603 | 6.14% |
| | coal_ore | 5,555 | 6.09% |
| | anvil | 5,439 | 5.96% |
| **Dimensional Shapes** | | | |
| | 3 | 7,346 | 50.15% |
| | 2 | 7,295 | 49.84% |
| **Tools** | | | |
| | stone_pickaxe | 9,329 | 25.51% |
| | stone_sword | 9,180 | 25.10% |
| | stone_axe | 9,150 | 24.99% |
| | stone_shovel | 8,906 | 24.36% |

## Table 16: Farming Task Diversity Statistics

| Diversity | Type | Count | Percentage |
|---|---|---|---|
| **Action Sequences** | | | |
| | 4 | 7,458 | 50.33% |
| | 5 | 3,731 | 25.17% |
| | 3 | 3,264 | 22.02% |
| | 6 | 270 | 1.82% |
| | 2 | 81 | 0.55% |
| | 7 | 11 | 0.07% |
| **Agents** | | | |
| | 2 | 7,465 | 50.37% |
| | 3 | 7,350 | 49.63% |
| **Scenes** | | | |
| | snow_mountain | 3,732 | 25.18% |
| | swamp | 3,722 | 25.11% |
| | ice_on_water | 3,707 | 25.01% |
| | village | 3,654 | 24.69% |
| **Background Types** | | | |
| | stone | 2,892 | 19.51% |
| | obsidian | 1,549 | 10.46% |
| | hay_block | 1,527 | 10.30% |
| | oak_wood | 1,524 | 10.28% |
| | cyan_concrete | 1,492 | 10.06% |
| | glass | 1,465 | 9.88% |
| | smooth_quartz | 1,462 | 9.86% |
| | pink_wool | 1,455 | 9.81% |
| | dirt | 1,449 | 9.77% |
| **Target Types** | | | |
| | potato | 4,972 | 33.56% |
| | carrot | 4,955 | 33.45% |
| | wheat | 4,888 | 32.99% |
| **Target Counts** | | | |
| | 4 | 2,873 | 19.39% |
| | 3 | 2,269 | 15.31% |
| | 5 | 2,256 | 15.22% |
| | 6 | 2,151 | 14.51% |
| | 2 | 1,240 | 8.37% |
| | 8 | 1,112 | 7.50% |
| | 10 | 1,062 | 7.17% |
| | 7 | 933 | 6.29% |
| | 12 | 512 | 3.45% |
| | 14 | 407 | 2.75% |

## Table 17: Smelting Task Diversity Statistics

| Diversity | Type | Count | Percentage |
|---|---|---|---|
| **Action Sequences** | | | |
| | 5 | 3,261 | 30.20% |
| | 4 | 3,072 | 28.45% |
| | 6 | 2,041 | 18.89% |
| | 3 | 1,824 | 16.88% |
| | 2 | 358 | 3.31% |
| | 7 | 239 | 2.21% |
| | 8 | 8 | 0.07% |
| **Agents** | | | |
| | 3 | 5,480 | 50.75% |
| | 2 | 5,323 | 49.25% |
| **Scenes** | | | |
| | snow_mountain | 2,272 | 21.04% |
| | desert_villege | 2,257 | 20.92% |
| | swamp | 2,171 | 20.08% |
| | ice_on_water | 2,059 | 19.09% |
| | villege | 2,044 | 18.87% |
| **Background Types** | | | |
| | gold_block | 1,014 | 9.22% |
| | smooth_quartz | 1,010 | 9.19% |
| | cyan_concrete | 995 | 9.02% |
| | glowstone | 981 | 8.92% |
| | pink_wool | 990 | 8.99% |
| | glass | 978 | 8.89% |
| | oak_wood | 987 | 8.98% |
| | grass_block | 977 | 8.88% |
| | hay_block | 968 | 8.80% |
| | stone | 964 | 8.76% |
| | obsidian | 939 | 8.54% |
| **Furnace** | | | |
| | 1 | 5,772 | 53.45% |
| | 2 | 5,031 | 46.55% |
| **Fuel Types** | | | |
| | coal_block | 999 | 9.58% |
| | charcoal | 962 | 9.22% |
| | lava_bucket | 940 | 9.01% |
| | coal | 921 | 8.84% |
| | spruce_planks | 910 | 8.73% |
| | acacia_planks | 906 | 8.69% |
| | oak_planks | 861 | 8.26% |
| | birch_log | 893 | 8.57% |
| | acacia_log | 887 | 8.50% |
| | spruce_log | 845 | 8.10% |
| | oak_log | 840 | 8.05% |
| | birch_planks | 839 | 8.04% |
| **Goal Types** | | | |
| | food | 5,412 | 50.09% |
| | item | 5,391 | 49.91% |
| **Target Types** | | | |
| | glass | 1,144 | 10.26% |
| | gold_ingot | 1,094 | 9.81% |
| | stone | 1,077 | 9.66% |
| | smooth_sandstone | 1,040 | 9.32% |
| | iron_ingot | 1,036 | 9.29% |
| | cooked_salmon | 712 | 6.38% |
| | cooked_cod | 708 | 6.35% |
| | baked_potato | 758 | 6.80% |
| | cooked_mutton | 664 | 5.95% |
| | cooked_rabbit | 648 | 5.81% |
| | cooked_porkchop | 668 | 5.99% |
| | cooked_beef | 627 | 5.62% |
| | cooked_chicken | 627 | 5.62% |
| **Target Counts** | | | |
| | 2 | 3,999 | 37.01% |
| | 3 | 3,363 | 31.13% |
| | 1 | 1,909 | 17.68% |
| | 4 | 1,532 | 14.18% |
| **Tools** | | | |
| | iron_pickaxe | 18,633 | 29.69% |
| | iron_shovel | 13,676 | 21.78% |
| | iron_axe | 13,453 | 21.43% |
| | iron_sword | 13,448 | 21.42% |
| **Resource Types** | | | |
| | red_sand | 2,032 | 10.37% |
| | gold_ore | 1,999 | 10.20% |
| | cobblestone | 1,915 | 9.77% |
| | sandstone | 1,818 | 9.28% |
| | iron_ore | 1,780 | 9.08% |
| | coal_ore | 1,714 | 8.75% |
| | acacia_planks | 1,564 | 7.98% |
| | oak_planks | 1,503 | 7.67% |
| | birch_log | 1,486 | 7.58% |
| | spruce_log | 1,477 | 7.54% |
| | oak_log | 1,456 | 7.44% |
| | spruce_planks | 1,471 | 7.51% |
| | birch_planks | 1,344 | 6.86% |
| | sheep | 1,119 | 5.71% |
| | pig | 1,104 | 5.63% |
| | rabbit | 1,097 | 5.60% |
| | chicken | 1,081 | 5.52% |
| | cow | 700 | 3.57% |
| | mooshroom | 675 | 3.44% |

 # L    Datasheet

## Motivation

**For what purpose was the dataset created? Was there a specific task in mind? Was there a specific gap that needed to be filled? Please provide a description.**

The TeamCraft dataset was created to support development and evaluation for multi-modal multi-agent systems in MineCraft.

**Who created the dataset (e.g., which team, research group) and on behalf of which entity (e.g., company, institution, organization)?**

The dataset was created by the TeamCraft team.

**Who funded the creation of the dataset? If there is an associated grant, please provide the name of the grantor and the grant name and number.**

The dataset was funded by the TeamCraft team.

**Any other comments?**

None.

## Composition

**What do the instances that comprise the dataset represent (e.g., documents, photos, people, countries)? Are there multiple types of instances (e.g., movies, users, and ratings; people and interactions between them; nodes and edges)? Please provide a description.**

Each instance contains a ground-truth expert demonstration of a multi-agent team finishing a task in Minecraft, and the corresponding multi-modal prompts specifying the task.

**How many instances are there in total (of each type, if appropriate)?**

There are in total 57,207 instances.

**Does the dataset contain all possible instances or is it a sample (not necessarily random) of instances from a larger set? If the dataset is a sample, then what is the larger set? Is the sample representative of the larger set (e.g., geographic coverage)? If so, please describe how this representa-tiveness was validated/verified. If it is not representative of the larger set, please de-scribe why not (e.g., to cover a more diverse range of instances, because instances were withheld or unavailable).**

The dataset contain all possible instances.

**What data does each instance consist of? "Raw" data (e.g., unprocessed text or images) or features? In either case, please provide a description.**

Each instance consists of a multi-modal task specification, agents observations and expert tra-jectories. Each task specification contains one raw language instruction and three orthographic views images. Agents observations contain the first-person view RGB images and the inventory information.

**Is there a label or target associated with each instance? If so, please provide a description.**

N/A.

**Is any information missing from individual in-stances? If so, please provide a description, explaining why this information is missing (e.g., because it was unavailable). This does not include intentionally removed information, but might include, e.g., redacted text.**

We intentionally removed the expert demonstra-tion in the test set to prevent over-fitting.

**Are relationships between individual in-stances made explicit (e.g., users' movie rat-ings, social network links)? If so, please de-scribe how these relationships are made ex-plicit.**

Each instance in the dataset corresponds to an individual task variant that belongs to one of the four task types (i.e. building, clearing, farming, smelting). The task type is explicitly specified in the file name.

**Are there recommended data splits (e.g., training, development/validation, testing)? If so, please provide a description of these splits, explaining the rationale behind them.**

The dataset has been split into a training set (55,257 instances), a validation set (1,000 in-stances) and a test set (950 instances). The training set is designed for model training while

the validation set is for hyperparameter tuning and checkpoint selection. The test set is designed to evaluate the model's generalization capabilities across novel scenes, novel goal states and novel agent numbers.

Are there any errors, sources of noise, or redundancies in the dataset? If so, please provide a description.

None as we know.

Is the dataset self-contained, or does it link to or otherwise rely on external resources (e.g., websites, tweets, other datasets)? If it links to or relies on external resources, a) are there guarantees that they will exist, and remain constant, over time; b) are there official archival versions of the complete dataset (i.e., including the external resources as they existed at the time the dataset was created); c) are there any restrictions (e.g., licenses, fees) associated with any of the external resources that might apply to a dataset consumer? Please provide descriptions of all external resources and any restrictions associated with them, as well as links or other access points, as appropriate.

The dataset is entirely self-contained.

Does the dataset contain data that might be considered confidential (e.g., data that is protected by legal privilege or by doctor– patient confidentiality, data that includes the content of individuals' non-public communications)? If so, please provide a description.

None as we know.

Does the dataset contain data that, if viewed directly, might be offensive, insulting, threatening, or might otherwise cause anxiety? If so, please describe why.

None as we know.

Does the dataset identify any subpopulations (e.g., by age, gender)? If so, please describe how these subpopulations are identified and provide a description of their respective distributions within the dataset.

No.

Is it possible to identify individuals (i.e., one or more natural persons), either directly or indirectly (i.e., in combination with other data) from the dataset? If so, please describe how.

No.

Does the dataset contain data that might be considered sensitive in any way (e.g., data that reveals race or ethnic origins, sexual orientations, religious beliefs, political opinions or union memberships, or locations; financial or health data; biometric or genetic data; forms of government identification, such as social security numbers; criminal history)? If so, please provide a description.

None as we know.

Any other comments?

None.

## Collection Process

How was the data associated with each instance acquired? Was the data directly observable (e.g., raw text, movie ratings), reported by subjects (e.g., survey responses), or indirectly inferred/derived from other data (e.g., part-of-speech tags, model-based guesses for age or language)? If the data was reported by subjects or indirectly inferred/derived from other data, was the data validated/verified? If so, please describe how.

In each data instance, the expert trajectory was generated programmatically via a planning algorithm. The language instruction was created by language templates. The orthographic views images and agent observations were collected in MineCraft.

What mechanisms or procedures were used to collect the data (e.g., hardware apparatuses or sensors, manual human curation, software programs, software APIs)? How were these mechanisms or procedures validated?

The data is automatically generated by running the data collection scripts. The procedure is further verified by the team via manual inspection.

If the dataset is a sample from a larger set, what was the sampling strategy (e.g., deterministic, probabilistic with specific sampling probabilities)?

N/A.

Who was involved in the data collection process (e.g., students, crowdworkers, contractors) and how were they compensated (e.g., how much were crowdworkers paid)?

Only the TeamCraft team members are voluntarily involved in the data collection process.

Over what timeframe was the data collected? Does this timeframe match the creation timeframe of the data associated with the instances (e.g., recent crawl of old news articles)? If not, please describe the timeframe in which the data associated with the instances was created.

The data were collected between February 2024 and September 2024.

Were any ethical review processes conducted (e.g., by an institutional review board)? If so, please provide a description of these review processes, including the outcomes, as well as a link or other access point to any supporting documentation.

No.

Did you collect the data from the individuals in question directly, or obtain it via third parties or other sources (e.g., websites)?

N/A.

Were the individuals in question notified about the data collection? If so, please describe (or show with screenshots or other information) how notice was provided, and provide a link or other access point to, or otherwise reproduce, the exact language of the notification itself.

N/A.

Did the individuals in question consent to the collection and use of their data? If so, please describe (or show with screenshots or other information) how consent was requested and provided, and provide a link or other access point to, or otherwise reproduce, the exact language to which the individuals consented.

N/A.

If consent was obtained, were the consenting individuals provided with a mechanism to revoke their consent in the future or for certain uses? If so, please provide a description, as well as a link or other access point to the mechanism (if appropriate).

N/A.

Has an analysis of the potential impact of the dataset and its use on data subjects (e.g., a data protection impact analysis) been conducted? If so, please provide a description of this analysis, including the outcomes, as well as a link or other access point to any supporting documentation.

No.

Any other comments?

None.

## Preprocessing/cleaning/labeling

Was any preprocessing/cleaning/labeling of the data done (e.g., discretization or bucketing, tokenization, part-of-speech tagging, SIFT feature extraction, removal of instances, processing of missing values)? If so, please provide a description. If not, you may skip the remaining questions in this section.

Yes. In each data instance, the three orthographic views images rendered by MineCraft are manually concatenated as one image.

Was the "raw" data saved in addition to the preprocessed/cleaned/labeled data (e.g., to support unanticipated future uses)? If so, please provide a link or other access point to the "raw" data.

No.

Is the software that was used to preprocess/clean/label the data available? If so, please provide a link or other access point.

N/A.

Any other comments?

None.

## Uses

Has the dataset been used for any tasks already? If so, please provide a description.

The dataset is used to develop the TeamCraft-VLA model, as described in this paper.

**Is there a repository that links to any or all papers or systems that use the dataset? If so, please provide a link or other access point.**

No.

**What (other) tasks could the dataset be used for?**

This dataset can be used for the development and evaluation of multi-modal multi-agent systems in MineCraft.

**Is there anything about the composition of the dataset or the way it was collected and preprocessed/cleaned/labeled that might impact future uses? For example, is there anything that a dataset consumer might need to know to avoid uses that could result in unfair treatment of individuals or groups (e.g., stereotyping, quality of service issues) or other risks or harms (e.g., legal risks, financial harms)? If so, please provide a description. Is there anything a dataset consumer could do to mitigate these risks or harms?**

Unknown to the authors of the datasheet.

**Are there tasks for which the dataset should not be used? If so, please provide a description.**

Unknown to the authors of the datasheet.

**Any other comments?**

None.

## Distribution

**Will the dataset be distributed to third parties outside of the entity (e.g., company, institution, organization) on behalf of which the dataset was created? If so, please provide a description.**

Yes, the dataset is available on the Internet.

**How will the dataset will be distributed (e.g., tarball on website, API, GitHub)? Does the dataset have a digital object identifier (DOI)?**

The dataset will be available on Huggingface. It does not have a DOI.

**When will the dataset be distributed?**

The dataset will be available online by 12/01/2024.

**Will the dataset be distributed under a copyright or other intellectual property (IP) license, and/or under applicable terms of use (ToU)? If so, please describe this license and/or ToU, and provide a link or other access point to, or otherwise reproduce, any relevant licensing terms or ToU, as well as any fees associated with these restrictions.**

The dataset is under Apache 2.0 license.

**Have any third parties imposed IP-based or other restrictions on the data associated with the instances? If so, please describe these restrictions, and provide a link or other access point to, or otherwise reproduce, any relevant licensing terms, as well as any fees associated with these restrictions.**

None as we know.

**Do any export controls or other regulatory restrictions apply to the dataset or to individual instances? If so, please describe these restrictions, and provide a link or other access point to, or otherwise reproduce, any supporting documentation.**

Unknown to authors of the datasheet.

**Any other comments?**

None.

## Maintenance

**Who will be supporting/hosting/maintaining the dataset?**

The TeamCraft team will be maintaining the dataset.

**How can the owner/curator/manager of the dataset be contacted (e.g., email address)?**

Email: teamcraftbench@gmail.com

**Is there an erratum? If so, please provide a link or other access point.**

No.

Will the dataset be updated (e.g., to correct labeling errors, add new instances, delete instances)? If so, please describe how often, by whom, and how updates will be communicated to dataset consumers (e.g., mailing list, GitHub)?

No planned updates at the time of preparing this datasheet.

If the dataset relates to people, are there applicable limits on the retention of the data associated with the instances (e.g., were the individuals in question told that their data would be retained for a fixed period of time and then deleted)? If so, please describe these limits and explain how they will be enforced.

Unknown to authors of the datasheet.

Will older versions of the dataset continue to be supported/hosted/maintained? If so, please describe how. If not, please describe how its obsolescence will be communicated to dataset consumers.

N/A.

If others want to extend/augment/build on/-contribute to the dataset, is there a mechanism for them to do so? If so, please provide a description. Will these contributions be validated/verified? If so, please describe how. If not, why not? Is there a process for communicating/distributing these contributions to dataset consumers? If so, please provide a description.

Others may do so and should contact the original authors about incorporating fixes/extensions.

Any other comments?

None.

