# OpenReview forum: "TeamCraft: A Multi-Modal Benchmark for Collaborative Agents in Minecraft"
_NeurIPS.cc/2025/Datasets_and_Benchmarks_Track — Submitted to NeurIPS 2025 Datasets and Benchmarks Track_

### Official Review · Reviewer_M8XG · 2025-07-01

**Rating:** 4
**Confidence:** 5

**Summary:**

This paper presented TeamCraft, a benchmark for multi-modal multi-agent collaborative task planning in Minecraft. Through extensive experiments on this benchmark, the paper systematically reveals the significant limitations of current multi-agent VLA models in terms of generalization capability, particularly in collaborative planning, dynamic coordination, and adapting to new scenarios.

**Additional Feedback:**

For details, see the limitations discussion.

**Dataset Code Accessibility:**

Yes

**Dataset Code Comments:**

All data, code, and evaluation scripts are on GitHub and Huggingface.

**Ethical Comments:**

It does not include any personal data, human subjects, or sensitive biometric information.

**Ethical Considerations:**

No, there are no or only very minor ethics concerns

**Final Justification:**

After carefully reviewing your response, I have a clearer understanding of the proposed dataset. Your clarifications are consistent with my initial assessment, and I appreciate the effort made in addressing the concerns. I will maintain my score of 4 (Borderline Accept).

**Limitations Weaknesses:**

1. The conclusion on model scaling is not fully supported and appears inconsistent. The paper claims that "scaling up model size does not guarantee success", noting that the 7B model's performance approaches the 13B's as training data increases. However, a performance gap still exists in several experiments shown in Figure 5. Furthermore, the 7B-to-13B jump is not large enough to make a definitive claim about scaling laws. This conclusion also seems to conflict with the paper's other finding that larger proprietary models *did* significantly outperform the finetuned models on the complex, long-horizon Smelting task, suggesting that for certain tasks, a larger model size is indeed beneficial.
2. The benchmark does not sufficiently demonstrate the necessity of a multi-agent approach. While Table 7 compares 2-agent and 3-agent performance, the paper lacks a critical baseline: a single-agent system. Without this comparison, it is difficult to quantify the efficiency or accuracy gains achieved by adding more agents. This makes it unclear whether the tasks are merely "solvable" by multiple agents or truly "require" a multi-agent solution for optimal performance.
3. There is a lack of in-depth analysis of the relationship between task requirements and agent count. A crucial metric for a multi-agent benchmark is how task difficulty scales with the number of agents. The qualitative analysis reveals that when assigned a four-agent task, the model often "overlooked" the fourth agent. This raises a critical question: are the tasks inherently designed to be solvable by three agents, or is this simply a failure of the model's planning capability? If the former is true, it could limit the benchmark's significance for evaluating systems designed for a larger, more variable number of agents.
4. An interesting negative scaling phenomenon is observed but not fully explored. The paper notes that in the Grid-World setting, more training data led to worse generalization performance on some tasks, which the authors attribute to overfitting. However, in the primary multi-modal setting, performance improved with more data. It remains an open question whether this multi-modal setting would also suffer from performance degradation if the dataset were scaled even further. A deeper analysis of this discrepancy could provide valuable insights into the learning dynamics of state-based versus vision-based models.
5. The description of the expert demonstration planner lacks detail. The paper mentions that a planner optimizes a cost function to generate demonstrations, but the details of this optimization process are not provided. The claim that the resulting plans are optimal is not quantitatively validated, making it difficult for researchers to fully understand the properties and potential biases of the expert data they are using for training.

**Strengths Contributions:**

1. Addresses a critical gap in multi-agent research：Much of the existing research on multi-agent collaboration relies on simplified grid-world environments or text-only interactions, which lack the richness of real-world perception. This work introduces a benchmark that is fundamentally multi-modal, incorporating vision as a primary input. This allows for a more effective and realistic evaluation of multi-agent systems' performance in visually rich and task-diverse environments
2. The benchmark is not limited to simple navigation or movement. The four core task categories (Building, Clearing, Farming, and Smelting) are well-designed to test a range of complex collaborative skills. These include managing action dependencies , strategic tool use , dynamic state tracking of the environment and other agents' actions , and long-horizon planning for multi-step processes.
3. Thorough experimental evaluation with a focus on generalization.
4. Clear presentation and strong reproducibility.

---

> ### Author Rebuttal · Authors · 2025-07-31
>
> Dear Reviewer M8XG:
>
> Thank you for the time you took to provide your thorough, knowledgeable, and constructive assessment. We are inspired by your recognition that we have addressed the gap in multi-agent research with vision as a primary input, thorough evaluations, a clear presentation, and strong reproducibility.  We are particularly grateful for your acknowledgement that our task design effectively “tests a range of complex collaborative skills”.
>
> We address each of your concerns below:
>
> > The conclusion on model scaling is not fully supported and appears inconsistent. The paper claims that "scaling up model size does not guarantee success", noting that the 7B model's performance approaches the 13B's as training data increases. However, a performance gap still exists in several experiments shown in Figure 5.
>
> We agree that there is still a performance gap. Our main points here are (1) the gap is not significant, (2) the gap does not hold constant across multiple cases (especially those in task success rate with 100% data), the performances of several 7B models are very close to the performance of 13B, and even outperform 13B.
>
> > Furthermore, the 7B-to-13B jump is not large enough to make a definitive claim about scaling laws.
>
> We certainly agree that tests with more and larger models as additional baselines would definitely draw an even clearer pattern; however, we were limited by our budget, as excluding the overhead of preliminary dataset preparations, model iterations, and data verification, it took 8K GPU hours ($26K) to train and evaluate the reported model checkpoints and perform the reported ablations [1]. Running an experiment at this scale was a major undertaking for academics like ourselves. We allocated our budget to the most valuable aspects of our study, prioritizing scaling and methodological validation. We believe that our scaling trends and systematic generalization study with novel scenes, goals, materials, and numbers of agents has exposed the limitations of the latest multi-modal multi-agent models and has identified promising research directions.
>
> [1] Training: AWS p4d.24xlarge instances, 8\*A100 40G,  \\$32.77/h. 6,144 GPU hours, (\\$32.77 / 8) \* 6,144 = \\$25,167. Evaluation: Lambda Cloud A6000 instance, 1\*A6000, \\$0.8/h. 1,440 GPU hours, \\$0.8 \* 1,440 = \$1,152.
>
> > This conclusion also seems to conflict with the paper's other finding that larger proprietary models did significantly outperform the finetuned models on the complex, long-horizon Smelting task, suggesting that for certain tasks, a larger model size is indeed beneficial.
>
> Our scaling claim is drawn from controlled experiments where we varied only the parameter count of the same architecture, training recipe, and dataset. By contrast, the proprietary systems differ along many hidden axes, including but not limited to, total parameters, vision encoders, context length, instruction tuning, reinforcement learning, and curated private corpora. None of those aspects can we disentangle. Without access to those details a parameter-only comparison would be misleading, so we restricted our scaling conclusion to the fully specified, open-source model family.
>
>
> > The benchmark does not sufficiently demonstrate the necessity of a multi-agent approach. While Table 7 compares 2-agent and 3-agent performance, the paper lacks a critical baseline: a single-agent system. Without this comparison, it is difficult to quantify the efficiency or accuracy gains achieved by adding more agents. This makes it unclear whether the tasks are merely "solvable" by multiple agents or truly "require" a multi-agent solution for optimal performance.
>
> We defined our tasks such that they can be solved only in multi-agent settings. As specified in Section 3.3, success requires complementary inventories and tool specializations. For example, building tasks and farming tasks require heterogeneous bricks/seeds distributed across agents; breaking tasks and melting tasks require specialized tools to optimally perform tasks, such as killing chickens with a sword and splitting wooden logs with an axe.
>
> We tried a single-agent setting that merged all inventories and tools, such that it becomes possible to perform the task, but most of the test cases hit the episode timeout.
>
> **Action:** We will make these task constraints explicit in the main text. We appreciate your suggestion and will clarify the above rationale in our revision.
>
> > There is a lack of in-depth analysis of the relationship between task requirements and agent count. A crucial metric for a multi-agent benchmark is how task difficulty scales with the number of agents. The qualitative analysis reveals that when assigned a four-agent task, the model often "overlooked" the fourth agent. This raises a critical question: are the tasks inherently designed to be solvable by three agents, or is this simply a failure of the model's planning capability? If the former is true, it could limit the benchmark's significance for evaluating systems designed for a larger, more variable number of agents.
>
> The tasks are designed to scale with the number of agents, ensuring that the per-agent workload remains consistent. This design allows us to control for other variables and isolate the number of agents as the single independent variable. As mentioned earlier, certain tasks—such as building and farming—require heterogeneous resources (e.g., bricks or seeds) that are distributed across agents. Similarly, tasks like breaking and melting demand specialized tools, making full participation from all agents essential. Therefore, if only three agents are active in a four-agent task, the team will either be unable to complete the task or will do so inefficiently.
>
> **Action:** We will include in our revision a dedicated section explaining this design. Thank you again for your suggestion.
>
>
> > An interesting negative scaling phenomenon is observed but not fully explored. The paper notes that in the Grid-World setting, more training data led to worse generalization performance on some tasks, which the authors attribute to overfitting. However, in the primary multi-modal setting, performance improved with more data. It remains an open question whether this multi-modal setting would also suffer from performance degradation if the dataset were scaled even further. A deeper analysis of this discrepancy could provide valuable insights into the learning dynamics of state-based versus vision-based models.
>
> This is a very good question. We have two hypotheses:
>
> - First, the drop‑off shows up only on the Goal and Agent splits, not on the in‑distribution test set. That suggests potential over‑fitting.
>
> - Second, the multimodal network incorporates an additional vision encoder and vision‑projector that expands its effective parameter budget. At the current data scale, that extra capacity can still absorb new examples without forcing catastrophic interference, whereas the text‑only Grid model has already saturated and enters the regime where more data just means harder memorization instead of better generalisation.
>
> > The description of the expert demonstration planner lacks detail. The paper mentions that a planner optimizes a cost function to generate demonstrations, but the details of this optimization process are not provided. The claim that the resulting plans are optimal is not quantitatively validated, making it difficult for researchers to fully understand the properties and potential biases of the expert data they are using for training.
>
> Our demonstrations are high quality but not provably optimal. As stated in  Appendix F, we have optimized for a large, joint cost function, utilizing perfect knowledge of the task, including goal object positions, agents’ inventories, and each agent’s efficiency in performing actions.
> The cost function is also hand-turned for different tasks.
>
> Meanwhile, perfect optimality is neither necessary nor even desirable for imitation learning. Human tele‑operation data are also sub‑optimal, and overly flawless trajectories can hurt robustness because the policy never sees recoveries from mistakes. This observation echoes findings in π‑0 [2] that mildly sub‑optimal demonstrations improve a model’s ability to self‑correct. By tuning the cost function rather than insisting on global optimality, we strike the same balance: plans are efficient enough to provide a strong signal yet imperfect enough to expose recoverable failure states.
>
> [2]  π‑0: A Vision–Language–Action Flow Model for General Robot Control

---

> > ### Comment · Reviewer_M8XG · 2025-08-09
> >
> > Thank you for your rebuttal. After carefully reviewing your response, I have a clearer understanding of the proposed dataset. Your clarifications are consistent with my initial assessment, and I appreciate the effort made in addressing the concerns. I will maintain my score of 4 (Borderline Accept).

---

### Official Review · Reviewer_h7of · 2025-07-01

**Rating:** 4
**Confidence:** 4

**Summary:**

This paper introduces TeamCraft, a new multi-modal, multi-agent benchmark built on Minecraft. Additionally, the authors propose TeamCraft-VLA as a baseline method trained within this benchmark.

**Dataset Code Accessibility:**

Yes

**Ethical Considerations:**

No, there are no or only very minor ethics concerns

**Limitations Weaknesses:**

- **Limited novelty**: Several existing benchmarks are already based on Minecraft, such as MineDojo, MindAgent, and the less explicitly mentioned Mineland. Notably, MindAgent and Mineland also support multi-agent scenarios. Compared to these, the main contribution of this work appears to lie in providing additional demonstrations and in explicitly considering decentralized testing settings.
- **Task diversity is somewhat limited**: The four main task types proposed in this paper are mostly constrained to operations like adding or removing items within a given area. However, Minecraft offers a much broader range of possible tasks, such as crafting, navigation, and combat—many of which are included in benchmarks like MineDojo and Mineland.
- **Insufficient baseline comparisons**: In the proprietary VLA's vanilla setup, the comparison is made only against GPT-4o. As a benchmark, the work would benefit from including more baselines, such as other open-source and closed-source VLMs. It would also be valuable to evaluate models that have been specifically fine-tuned for Minecraft tasks, whether VLMs or VLAs. Including such comparisons would provide a more comprehensive assessment

**Strengths Contributions:**

- TeamCraft offers a diverse set of generalization tasks and includes a rich collection of pre-collected demonstrations. It also considers both centralized and decentralized training scenarios for the VLA model.
- The paper is well-written and easy to follow, with detailed examples that clearly illustrate the proposed tasks.

---

> ### Author Rebuttal · Authors · 2025-07-31
>
> Dear Reviewer h7of:
>
> Thank you for your time and valuable comments. We are pleased by your acknowledgement that  our paper is “clearly written and well organized”, as well as your praise of our appendix. We really appreciate your approval of our efforts and contributions in this work.
>
> We address your concerns point-by-point below:
>
> > Limited novelty: Several existing benchmarks are already based on Minecraft, such as MineDojo, MindAgent, and the less explicitly mentioned Mineland. Notably, MindAgent and Mineland also support multi-agent scenarios. Compared to these, the main contribution of this work appears to lie in providing additional demonstrations and in explicitly considering decentralized testing settings.
>
> Minecraft has indeed become a popular testbed. However, to our knowledge we are **one of the first multi-modal multi-agent benchmarks** that combines them at scale within a visually rich and dynamic environment, with generalization across real‑world‑like backgrounds, diverse object categories, complex action pipelines, and shifting task dynamics.
>
> - **MineDojo:** As Table 1 illustrates, MineDojo offers only voxel‑based observations and is restricted to a single‑agent setup. TeamCraft, by contrast, supplies first‑person RGB views for multiple collaborating agents, richer multimodal task specifications delivered via vision‑and‑language prompts, and evaluation under both centralized and decentralized control regimes.
>
> - **MindAgent:** Compared to MindAgent, TeamCraft features multi-modal task specification, visual observation, wide variety of generalization, a large dataset of  expert demonstrations, and extensive analysis of the decentralized settings (Table 1).
>
> - **MineLand:** The biggest difference is Mineland’s heavy focus as a simulation platform, not a standardized benchmark. It ships no standardized task suite, splits, and as of yet has not released its dataset. Additionally, we offer (1) greater task varieties, (2) extensive multi-modal task specification (Appendix F.1, Figure 29 – 36), (3) agent control regimes in both centralized and decentralized settings, (4) challenges in spatial understanding, (5) a procedurally-generated large-scale dataset, and (6) extensive analysis in multiple perspectives.
>
>   The two projects also differ fundamentally in their treatment of perception. MineLand supplies each agent with complete voxel‑grid state, inventories for all agents, and nearby entity information. Because an agent already has full knowledge of the world, the pixel view is optional and the setting does not stress vision‑language integration. By contrast, TeamCraft deliberately enforces partial observability (Sections 3.1, 3.4): an agent receives only its first‑person RGB stream and its own inventory, mirroring real‑world robotics where the private states of teammates are hidden. This makes our work “fundamentally multi-modal, incorporating vision as a primary input” (Reviewer M8XG),  and allows us to evaluate both centralized models, and decentralized models that must infer teammates’ intentions from limited cues, in a regime that MineLand’s “perfect information” assumption cannot support.
>
> > Task diversity is somewhat limited: The four main task types proposed in this paper are mostly constrained to operations like adding or removing items within a given area. However, Minecraft offers a much broader range of possible tasks, such as crafting, navigation, and combat—many of which are included in benchmarks like MineDojo and Mineland.
>
> TeamCraft is “not limited to simple navigation or movement”, and introduces “a range of complex collaborative skills” (Reviewer M8XG) including crafting, navigation, and combat, as is  detailed in Section 3.3. and Appendix B, which explains that nearly every atomic action carries a Location argument, so an agent must first navigate to the proper coordinates before manipulating blocks or entities.
>
> For example, in the Smelting task, agents locate and kill sheep, pigs, or cows, collect the raw meat, forage or mine fuel, move to a furnace, wait for the cooking process to finish, and finally retrieve the cooked items. This pipeline demands spatial reasoning, resource scheduling, and on‑the‑fly combat.
>
> In addition, we further allow the user to expand the task variety. To make such extensions straightforward, we have already included in our public codebase a step‑by‑step guide (Repo_url/docs/customize_task.md) that shows users how to script new tasks in our benchmark.
>
> > Insufficient baseline comparisons: In the proprietary VLA's vanilla setup, the comparison is made only against GPT-4o. As a benchmark, the work would benefit from including more baselines, such as other open-source and closed-source VLMs. It would also be valuable to evaluate models that have been specifically fine-tuned for Minecraft tasks, whether VLMs or VLAs. Including such comparisons would provide a more comprehensive assessment.
>
> As shown in Figure 6, we have conducted numerous tests and evaluations, including GPT-4o, o4-mini, Claude 3.7, and Gemini 2.5 Pro under the grid setting. We agree that additional baselines would enrich our study, however, a practical constraint limited us:
>
> - Excluding the overhead of preliminary dataset preparations, model iterations, and data verification, it took 8K GPU hours ($26K) to train and evaluate the reported model checkpoints and perform the reported ablations [1]. Running an experiment at this scale was a major undertaking for academics like ourselves. We allocated our budget to the most valuable aspects of our study, prioritizing scaling and methodological validation. We believe that our scaling trends and systematic generalization study with novel scenes, goals, materials, and numbers of agents has exposed the limitations of the latest multi-modal multi-agent models and identified promising research directions.
>
> [1] Training: AWS p4d.24xlarge instances, 8\*A100 40G,  \\$32.77/h. 6,144 GPU hours, (\\$32.77 / 8) \* 6,144 = \\$25,167. Evaluation: Lambda Cloud A6000 instance, 1\*A6000, \\$0.8/h. 1,440 GPU hours, \\$0.8 \* 1,440 = \$1,152.

---

### Official Review · Reviewer_BA9X · 2025-07-03

**Rating:** 5
**Confidence:** 3

**Summary:**

This paper introduces TeamCraft, a multi-agent benchmark environment built in Minecraft, featuring both text-based state representations and visual observations. The benchmark is designed to evaluate collaborative task planning and execution across multiple agents. The authors also propose TeamCraft-VLA, a model that generates discrete task plans for multiple agents based on multimodal (vision and language) observations. The paper presents experimental results examining the performance of TeamCraft-VLA under varying model capacities and across both centralized and decentralized planning settings. Additionally, the authors compare their model against several proprietary VLMs. The findings highlight the strong task planning capabilities of certain VLMs, even without task-specific fine-tuning, and demonstrate competitive performance of the proposed TeamCraft-VLA model in multi-agent planning scenarios.

**Dataset Code Accessibility:**

Yes

**Ethical Considerations:**

No, there are no or only very minor ethics concerns

**Final Justification:**

All my concerns have been addressed and clarified. I'm therefore recommending acceptance.

**Limitations Weaknesses:**

- The biggest concern is with the lack of evaluation against other VLAs. While it’s true that multi-agent VLAs are still rare, the decentralized TeamCraft-VLA is basically the same as a single-agent setup. So it would still be useful to compare against existing single-agent VLAs just to establish a baseline. Also, in Figure 5, TeamCraft-VLA barely performs under the decentralized setting, and in Figure 6, it underperforms compared to several VLMs that weren’t even fine-tuned. That makes it hard to see what the model itself is really contributing to the paper.
- There’s also a missing comparison to Voyager, which uses LLMs for planning and was evaluated in Minecraft too. Since that’s a pretty well-known work in this space, the lack of discussion around it is a noticeable omission.
- The comparison table doesn’t include VillagerBench, another recent multi-agent benchmark in Minecraft (from VillagerAgent: A Graph-Based Multi-Agent Framework). That’s an important point of reference and should be included to give readers a clearer picture of where TeamCraft fits in.
- The paper also misses recent multi-agent RL benchmarks like BenchMARL and MultiAgentBench (which came out in March 2025 and includes a Minecraft subset). These are relevant and would help situate TeamCraft better in the broader multi-agent learning landscape.
- Lastly, there’s some terminology confusion. In my understanding, models like GPT-4o, Claude, and Gemini 2.5 are VLMs, not VLAs. They were not trained to generate actions directly, so it’s a bit misleading to group them under the same umbrella as VLA models.

**Strengths Contributions:**

- The paper is clearly written and well organized, with extensive and helpful supplementary material that enhances clarity and reproducibility.
- The codebase is open-source and well documented
- The authors propose a multi-agent VLA model, evaluated on the TeamCraft benchmark. The model demonstrates reasonable performance, providing a decent baseline for future multi-agent VLA research.

---

> ### Author Rebuttal · Authors · 2025-07-31
>
> Dear Reviewer BA9X:
>
> Thank you for reviewing our paper and acknowledging that it is “clearly written and well organized”, as well as for your praise of our  appendix. We appreciate your approval of our effort and contributions.
>
> We address your concerns below and indicate the edits that we will make in our revision:
>
> > The biggest concern is with the lack of evaluation against other VLAs. While it’s true that multi-agent VLAs are still rare, the decentralized TeamCraft-VLA is basically the same as a single-agent setup. So it would still be useful to compare against existing single-agent VLAs just to establish a baseline.
>
> **As shown in Figure 6, we have conducted numerous tests and evaluations, including GPT-4o, o4-mini, Claude 3.7, and Gemini 2.5 Pro.** We agree that additional single‑agent VLAs could offer context; however, two factors have limited us:
>
> - Methodological: We have defined tasks such that they can only be solved in multi-agent settings. As specified in Section 3.3, success requires complementary inventories and tool specializations. For example, building tasks and farming tasks require heterogeneous bricks/seeds distributed across agents; breaking tasks and melting tasks require specialized tools to optimally perform them, such as killing chickens with a sword and breaking wooden logs with an axe. In Appendix I.4, we compared multi-agent effectiveness under two and three agents settings.
>
> - Practical: Excluding the overhead of preliminary dataset preparations, model iterations, and data verification, it took 8K GPU hours ($26K) to train and evaluate the reported model checkpoints and perform the reported ablations[1]. For academics like ourselves, running experiments at this scale was a major undertaking. We allocated our budget to the most valuable aspects of our study, prioritizing scaling and methodological validation. We believe that our scaling trends and systematic generalization study with novel scenes, goals, materials, and numbers of agents have exposed the limitations of the latest multi-modal multi-agent models and identified promising research directions.
>
>
> [1] Training: AWS p4d.24xlarge instances, 8\*A100 40G,  \\$32.77/h. 6,144 GPU hours, (\\$32.77 / 8) \* 6,144 = \\$25,167. Evaluation: Lambda Cloud A6000 instance, 1\*A6000, \\$0.8/h. 1,440 GPU hours, \\$0.8 \* 1,440 = \$1,152.
>
> > Also, in Figure 5, TeamCraft-VLA barely performs under the decentralized setting, and in Figure 6, it underperforms compared to several VLMs that weren’t even fine-tuned. That makes it hard to see what the model itself is really contributing to the paper.
>
> We do not position the TeamCraft-VLA model itself as the primary novelty, and we do not claim architectural originality. Decentralized TeamCraft‑VLA is provided as a strong, transparent reference baseline in order to expose where current VLA systems break: understanding of spatial 3D images, coordination under partial observability, intent inference without communication, and scaling to unseen goals/scenes/agent counts. The objective of our model and our systematic studies is to reveal concrete gaps and thereby inspire follow‑up research on communication priors, decentralized planning, and scalable coordination mechanisms, as discussed in Section 5.1, L333–L337. Several publications [2,3,4] have found our insights valuable and have cited our work.
>
> [2] Why do Multi-Agent LLM Systems Fail?
>
> [3] Large Language Models for Planning: A Comprehensive and Systematic Survey
>
> [4] From LLM Reasoning to Autonomous AI Agents: A Comprehensive Review
>
> > There’s also a missing comparison to Voyager, which uses LLMs for planning and was evaluated in Minecraft too. Since that’s a pretty well-known work in this space, the lack of discussion around it is a noticeable omission.
>
> We appreciate your suggestion to position our work more clearly with respect to Voyager, a seminal effort in the LLM‑as‑agent literature (cited in L78), and we will do so in our revision. Voyager is a single‑agent system research prototype, not a reusable benchmark: it operates from text‑only world state, lacks pixel‑level perception, and provides no facility for multi‑agent coordination. By contrast, TeamCraft is the first benchmark that simultaneously offers (i) visual observations, (ii) multiple collaborating agents, and (iii) tasks spanning diverse scenes, object categories, and crafting pipelines.
>
> > The comparison table doesn’t include VillagerBench, another recent multi-agent benchmark in Minecraft (from VillagerAgent: A Graph-Based Multi-Agent Framework). That’s an important point of reference and should be included to give readers a clearer picture of where TeamCraft fits in.
>
> VillagerBench re‑uses the MineAgent simulator skeleton and adds three multi-agent tasks. Both MineAgent and VillagerBench do not support visual observation, and operate only in centralized settings. Both  also only offer tasks in one setting and do not include task generalization and datasets based on those generalizations. Because VillagerBench’s capabilities are essentially a superset of MineAgent, we cited it together with MineAgent in Section 2 (L80) and, due to space limitations, kept only the MineAgent row in Table 1.
>
>
> > The paper also misses recent multi-agent RL benchmarks like BenchMARL and MultiAgentBench (which came out in March 2025 and includes a Minecraft subset). These are relevant and would help situate TeamCraft better in the broader multi-agent learning landscape.
>
> Thank you for bringing up these two very recent efforts. As we submitted the initial version of our paper on Arxiv in Dec 2024, both papers were essentially concurrent with our work, and we will cite them in our revision.
>
> MultiAgentBench (Mar 2025) is a suite that aggregates several existing multi‑agent tasks. Its Minecraft component re‑uses the voxel-based MineAgent environment. Because that subset inherits MineAgent’s observation modality (no raw vision) and offers only a few more tasks, we did not originally separate it from MineAgent in Table 1.
>
> BenchMARL (Nov 2024) is an excellent collection that standardizes evaluation across MARL algorithms, but the included environments, VMAS (2022), SMAC v2 (2022), MPE (2017), SISL (2017), MeltingPot (2021), are comparatively low‑fidelity, featuring 2‑D or voxel‑based observations and short horizons.
>
> TeamCraft therefore targets a different research gap: multi-modal multi-agent collaboration in complex tasks and wide generalization. We view the above two suites as complementary, as BenchMARL and MultiAgentBench are ideal for fast MARL algorithm benchmarking, whereas TeamCraft stresses perception‑heavy planning and division‑of‑labor reasoning.
>
> **Action:** We will 1) expand the Related‑Work section with a paragraph comparing their scope and modalities to TeamCraft, and 2) provide pointers on how TeamCraft can be incorporated into future releases of these suites to create a unified evaluation spectrum.
>
>
> > Lastly, there’s some terminology confusion. In my understanding, models like GPT-4o, Claude, and Gemini 2.5 are VLMs, not VLAs. They were not trained to generate actions directly, so it’s a bit misleading to group them under the same umbrella as VLA models.
>
> **Action:** Thank you, we agree, and will re-label them as VLM planners or vision-language models throughout. Our term “VLA” will be reserved for models trained to output structured action tokens. We will update Section titles, Figure labels, and captions accordingly.

---

> > ### Comment · Reviewer_BA9X · 2025-08-07
> >
> > Thank you for the clarification and additional details. I appreciate you bringing up the point about training cost, and I completely understand. All my other concerns have been adequately responded to and addressed. With the proposed revision, I believe TeamCraft would be a great addition to the community. I'm therefore raising my score and would recommend acceptance.

---

### Official Review · Reviewer_CYq5 · 2025-07-03

**Rating:** 4
**Confidence:** 3

**Summary:**

This paper introduces a multimodal benchmark **TeamCraft** to support multi-modal multi-agent learning, especially in complex visual environments. This benchmark utilizes Minecraft as the experimental platform to simulate multi-agent task collaboration, involving sequences of interleaved language and image prompts and agents' actions. It covers a variety of complex and interactive multi-agent collaborative tasks, such as building, clearing, farming, and smelting.

Additionally, this work proposes TeamCraft-VLA, a multimodal VLA model for multi-agent collaborations. Extensive experiments are conducted to compare TeamCraft-VLA with other proprietary VLA models (GPT-4o, o4-mini, Claude 3.7, and Gemini-2.5-Pro), leading to many interesting observations.

**Dataset Code Accessibility:**

Yes

**Dataset Code Comments:**

Both the code (hosted on GitHub) and data (hosted on Hugging Face) are accessible and well-organized.

**Ethical Comments:**

The benchmark primarily uses Minecraft for data collection, and to the best of my knowledge, there are no significant data security or privacy concerns. Nevertheless, the authors should include a dedicated section discussing the potential social impacts of their work.

**Ethical Considerations:**

No, there are no or only very minor ethics concerns

**Final Justification:**

Most of my concerns have been addressed, so I raise my score to borderline accept.

**Limitations Weaknesses:**

1. The description and implementation details (L209-L215) of TeamCraft-VLA are insufficiently elaborated. How was this model trained? And what is the motivation behind its design and introduction?

2. Based on Table 1, it appears that existing benchmarks already focus on multimodal prompting (e.g., VIMA-Bench) and multi-agent collaboration (e.g., Neural MMO 2.0). The significance and unique advantages of the proposed benchmark should be more clearly articulated in comparison to these existing benchmarks.

3. Although Minecraft offers a powerful and user-friendly experimental platform to support simulated multi-agent collaboration, there remains a significant domain gap between the game environment and the real world. This work does not appear to include any experimental analysis or discussion addressing this domain gap.

**Strengths Contributions:**

1. Multi-agent collaboration is an important problem with significant implications for embodied intelligence, autonomous systems, and human-computer interaction. This benchmark is expected to promote the learning of multimodal multi-agent collaboration capabilities.

2. The experimental analysis is thorough and systematic, encompassing a total of 15 ablation studies that vary across dataset sizes, control settings, experimental configurations, and the sizes of VLA models.

3. The paper is well-written and provides clear and comprehensive details regarding the construction of the benchmark.

---

> ### Author Rebuttal · Authors · 2025-07-31
>
> Dear  Reviewer CYq5:
>
> Thank you for reviewing our paper and providing your valuable comments. We greatly appreciate your recognition of the importance of our benchmark to “promote the learning of multimodal multi-agent collaboration capabilities” as well as your acknowledgement of our thorough experimental analysis and writing clarity.
>
> Below we address each of your concerns and indicate the actions we will take in revising our paper:
>
> > The description and implementation details (L209-L215) of TeamCraft-VLA are insufficiently elaborated. How was this model trained?
>
> We “followed Liu et al. (2024)" (L212) to train our model. The full TeamCraft-VLA implementation and training details (language model, visual encoder, hyperparameters) are specified  in Appendix H (L1024–L1046) and in our released code.  Space constraints in the main paper compelled us to compress the section describing the model.
>
> **Action:** We will move key details from the appendix to the main text and include a reference to Appendix H.
>
> > And what is the motivation behind its design and introduction?
>
> We expounded our motivation in the Abstract and the Introduction section (L46–L54).
>
> Most existing VLAs are single‑agent or confined to tightly controlled tabletop-style setups with plain or uniform backgrounds. We developed TeamCraft-VLA to probe a model’s ability to coordinate and control multiple agents as a team in both centralized and decentralized settings. Our overarching objective was to assess whether the model generalizes to novel goals, unseen scenes, complex action sequences, and varying numbers of agents, in a **visually-rich**, complex, and dynamic environment.
>
> > Based on Table 1, it appears that existing benchmarks already focus on multimodal prompting (e.g., VIMA-Bench) and multi-agent collaboration (e.g., Neural MMO 2.0). The significance and unique advantages of the proposed benchmark should be more clearly articulated in comparison to these existing benchmarks.
>
> As is documented in Table 1, VIMA-Bench only supports single agents and Neural MMO 2.0 lacks visual inputs altogether (no visual observation, spatial understanding, multi-modal task specification). TeamCraft is purposefully designed to bridge this gap and catalyze research on multi-modal multi‑agent collaboration.
>
> We agree that both multimodal prompting and multi‑agent collaboration have previously been studied; however, to our knowledge we are **one of the first multi-modal multi-agent benchmarks** that combines them at scale within a visually rich and dynamic environment, with generalization across real‑world‑like backgrounds, diverse object categories, complex action pipelines, and shifting task dynamics.
>
>
> > Although Minecraft offers a powerful and user-friendly experimental platform to support simulated multi-agent collaboration, there remains a significant domain gap between the game environment and the real world. This work does not appear to include any experimental analysis or discussion addressing this domain gap.
>
> Although Minecraft is constructed entirely of blocks, TeamCraft includes a diverse range of target objects and visually distinct scenes, tools, and resources (Section 3.5 & Appendix E). Leveraging Minecraft’s rich taxonomy of block varieties, we create diverse environments by combining different block types to form unique landscapes, operational areas, animals, and objects analogous to the real world. These elements span various scales, from distant scenery to interactive elements within human sight, significantly enhancing the overall visual diversity.
>
> Moreover, the environment’s interaction logic with persistent state, resource constraints, and long-horizon objectives consistently reflects the structure of real-world tasks.
>
> > Nevertheless, the authors should include a dedicated section discussing the potential social impacts of their work.
>
> TeamCraft’s broader social impact is presented as an open, fully reproducible testbed for vision-based teamwork that promises to accelerate research on reliable multi-agent coordination in visually complex settings. By publishing transparent evaluation code, and expert demonstrations to a synthetic domain, we have aimed to maximize the scientific and educational value of TeamCraft.
>
> **Action:** Thank you for your suggestion. We will add a dedicated section addressing potential social impacts.

---

> > ### Comment · Reviewer_CYq5 · 2025-08-06
> >
> > Thank the authors for their detailed and constructive responses. Most of my concerns have been addressed, so I will increase my rating. However, the significance of combining "multi-modal" and "multi-agent" into one benchmark should be clarified.

---

### Note · Authors · 2025-08-13

We thank all reviewers for their careful evaluations and the time invested in our submission. Your constructive feedback helped us clarify our presentation and sharpen our positioning. We are pleased to see that all substantive concerns have been addressed with our rebuttal.

In our revision, we will incorporate the suggested clarifications and refinements. We hope the final version serves the community as a clear, reproducible resource that advances research on multi-modal, multi-agent collaboration at scale in visually rich, dynamic environments. With diverse objects, complex actions, and shifting task dynamics, our benchmark offers a promising path toward robust agent collaboration that mirrors how human teams accomplish complex objectives.

---

### Decision · Program_Chairs · 2025-09-18

**Decision:**

Reject

**Comment:**

The paper introduces TeamCraft, a multimodal multi-agent benchmark in Minecraft, along with a baseline model, TeamCraft-VLA, and is praised across reviews for being clearly written, reproducible, and filling an important gap in multimodal multi-agent learning. Strengths include the benchmark’s thoughtful design with diverse collaborative tasks, support for both centralized and decentralized settings, extensive ablation studies, strong reproducibility with open-source code, and thorough experiments that highlight limitations of current models. However, reviewers point out several weaknesses: the novelty and unique advantages of TeamCraft are not fully distinguished from prior Minecraft-based benchmarks (e.g., MineDojo, MindAgent, Mineland, Voyager, VillagerBench, BenchMARL, MultiAgentBench), and task diversity is somewhat limited compared to what Minecraft enables. Baseline comparisons are insufficient, with missing evaluations against both single-agent VLAs and specialized Minecraft models, making it difficult to judge the true contribution of TeamCraft-VLA, which also underperforms against proprietary VLMs in some settings. Other concerns include limited details on TeamCraft-VLA’s design and expert planner, lack of analysis of domain gaps to the real world, unclear justification of scaling claims, and insufficient evidence that the tasks truly require multi-agent coordination. Overall, the paper provides a valuable and timely benchmark with strong empirical work, but its novelty, baselines, and deeper analysis of design choices need strengthening.